

# Crystals and double quiver algebras from Jeffrey-Kirwan residues

Jiakang Bao[1][⋆] and Masahito Yamazaki[1,2,3][†]

**1** Kavli Institute for the Physics and Mathematics of the Universe,
University of Tokyo, Kashiwa, Chiba 277-8583, Japan
**2** Trans-Scale Quantum Science Institute, University of Tokyo, Tokyo 113-0033, Japan
**3** Graduate School of Physics, University of Tokyo, Tokyo 113-0033, Japan

⋆ jiakang.bao@ipmu.jp , † masahito.yamazaki@ipmu.jp

## Abstract

We construct statistical mechanical models of crystal melting describing the flavoured Witten indices of $\mathcal{N} \geq 2$ supersymmetric quiver gauge theories. Our results can be derived from the Jeffrey-Kirwan (JK) residue formulas, and generalize the previous results for quivers corresponding to toric Calabi-Yau threefolds and fourfolds to a large class of quivers satisfying the no-overlap condition, including those corresponding to some non-toric Calabi-Yau manifolds. We construct new quiver algebras which we call the double quiver Yangians/algebras, as well as their representations in terms of the aforementioned crystals. For theories with four supercharges, we compare the double quiver algebras with the existing quiver Yangians/BPS algebras, which we show can also be constructed from the JK residues. For theories with two supercharges, the double quiver algebras provide an algebraic description of the BPS states, including the information of the fixed points and their relative coefficients in the full partition functions.

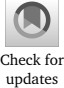

# 1 Introduction

The celebrated Witten index [1] and their cousins have always been important quantities in the study of supersymmetric quantum field theories. For supersymmetric theories on a two-torus, the elliptic genera [2] have attracted extensive studies in both physics and mathematics as they are topological invariants under smooth deformations of the theories. Under dimensional reduction, we obtain the Witten index that enumerates the Bogomol'nyi-Parasad-Sommerfield (BPS) states. The study of the BPS subspaces allows us to extract exact results for the theories.

A further dimensional reduction of the supersymmetric quantum mechanics leads us to the partition functions of the 0d matrix models, which can serve as useful models of the higher-dimensional counterparts.

Using the technique of supersymmetric localization, a powerful tool for computing these functionals has been developed in [3–6]. The path integrals can then be translated into the Jeffery-Kirwan (JK) residues [7] of the one-loop determinants. In this paper, we shall focus on the supersymmetric quiver gauge theories of the following two types. One of them would have $\mathcal{N} = 1$ supersymmetry in 4d, and the other would be 2d $\mathcal{N} = (0, 2)$ quivers. Therefore, upon dimensional reductions, we have the following partition functions:

- elliptic genera for 2d $\mathcal{N} = (2, 2)$ and 2d $\mathcal{N} = (0, 2)$ theories,

- Witten indices for 1d $\mathcal{N} = 4$ and 1d $\mathcal{N} = 2$ theories,

- matrix model partition functions for 0d $\mathcal{N} = 4$ and 0d $\mathcal{N} = 2$ theories.

Henceforth, we shall also refer to them as the cases with $\mathcal{N} = 4$ (four supercharges) and $\mathcal{N} = 2$ (two supercharges), respectively.

A salient family of these quivers can be constructed from D-branes probing toric Calabi-Yau (CY) threefolds or fourfolds [8–20]. The BPS counting problem is then formulated as the statistical mechanical model known as the crystal melting model [21–39]. The fixed points are encoded by the melting of the crystal where the atoms (resp. chemical bonds) correspond to the gauge nodes (resp. arrows) in the quiver. In the cases of toric $CY_3$, with this combinatorial structure at hand, the quiver Yangians/BPS algebras[1] $Y$ can be constructed as in [40–44] (see also [45] for a concise review).

These algebras have many prominent features with rich connections to different areas in physics and mathematics. As Yangian(-type) algebras with the coproduct structure [44,46,47], they have intimate relations with integrable models via the Bethe/gauge correspondence [48–52]. This is also reflected by the common elliptic/trigonometric/rational hierarchy [43] on both the gauge theory and the integrability sides. There are also hints on the extensions of these algebras to theories on Riemann surfaces of higher genera associated with generalized cohomologies [43, 53, 54]. Mathematically, counting BPS states is the study of Donaldson-Thomas (DT) invariants. It is believed that the quiver Yangian is the double of the cohomological Hall algebra (CoHA) [55–59] in the sense that the CoHA gives the positive/negative half of the quiver Yangian. Physically, they should also be related to the gauge origami systems and hence the quiver W-algebras [60–65]. The quivers that arise from toric $CY_3$ without compact divisors have also been widely studied under names such as affine Yangians, quantum toroidal algebras, quantum elliptic algebras, etc. In connections with the vertex operator algebras (VOAs), they are expected to play the role of the "universal algebras" of the affine $\mathcal{W}$-algebras [66–73].

The above-mentioned results on crystal melting and BPS algebras raises several questions. First, akin to the calculations of the JK residues, in the crystal representations of the quiver BPS algebras, the simple pole structure also plays a crucial role. Therefore, one might wonder if the quiver BPS algebras could be constructed from the JK residue formula. However, in the one-loop determinants, there are often more poles compared to the ones appearing in the quiver Yangians. Therefore, we need to separate the admissible poles from the inadmissible ones in the JK residue formula (at least for the cyclic chambers, as we will explain). Moreover, from the perspective of the crystals, the JK residues of the partition functions only add atoms while the pole structures from the algebras contain both the addable and the removable atoms for a given configuration. As we will see, these two discrepancies are closely related.

---

[1]For convenience, the quiver Yangians refer to the same algebras as the quiver BPS algebras in this paper (although strictly speaking, they are the rational quiver BPS algebras).

The investigations of the counting and the algebras are well-known in the context of toric geometry. Then one could ask how general these discussions can be. Given a generic quiver, do we still have the crystal melting description and does the quiver Yangian remain to be the BPS algebra? Given the speculation in the previous paragraph, this seems to be determined by how we may apply the JK residue formula.

So far, we have been mainly considering the theories with four supercharges. For theories with two supercharges, we would soon meet some obstructions. Although the fixed points still have the structure of crystal melting, the full information is not completely encoded by the crystals [38]. More concretely, the coefficients are now functions of the fugacities instead of just $\pm 1$ in the partition functions. Even if we just want to find some quiver Yangians that admit the crystals as representations and temporarily neglect the extra information, the knowledge we have for toric $CY_3$ does not completely work. In particular, a similar analysis of the pole structure in the factors that define the quiver Yangians would lead to some expressions that depend on the crystal states [36] and make it difficult to extract the algebra relations. From the JK residue formula, we can see that the expressions of the one-loop determinants are different for $\mathcal{N} = 4$ and $\mathcal{N} = 2$. Nevertheless, we still have the formula for the $\mathcal{N} = 2$ cases. Again, we might hope that the JK residue could help us construct the algebras not only for the $\mathcal{N} = 4$ cases but also for the $\mathcal{N} = 2$ ones.

In this paper, we will address these questions. After recalling the expressions of the one-loop determinants in the JK residue formula in §2, we will consider the quivers satisfying certain conditions where we can apply the JK residue formula to compute the partition functions in §3. This again allows us to arrange the fixed points as some combinatorial structure which we shall still call crystals for simplicity, and some examples are given in §4. For these quivers, we will construct new algebras, the double quiver Yangians/algebras $\widetilde{Y}$, from the JK residue formula in §5. They are expected to recover the (refined) BPS indices in the cyclic chambers, and we shall also comment on the crystals and the algebras for more general chambers. Then in §6, we will see how the quiver Yangians could be obtained from the JK residue formula. In particular, this tells us to what extent the standard representations of the quiver Yangians could still serve as the counting of the BPS states of the theories. In §7, we will apply the same method to construct the double quiver algebras for theories with two supercharges. They would not only have the crystals as their representations but also contain the full information of the BPS states in their actions on the states. We will discuss some examples in §8 as an illustration of the construction of the algebras. Some basic aspects of the quiver Yangians are reviewed in Appendix A.

## 2 Summary of the Jeffrey-Kirwan residue formula

As studied in [3–6], the supersymmetric partition functions of interest in this paper can be computed as the JK residues [7]:[2]

$$\mathcal{Z}(\epsilon) = \frac{1}{|\mathcal{W}|} \sum_{u^* \in \mathsf{M}^*_{\text{sing}}} \text{JK-Res}_{u=u^*}(\mathsf{Q}(u^*), \eta) \, Z_{\text{1-loop}}(\epsilon, u) \,. \tag{1}$$

Here, we consider a theory with gauge group $G$, whose Lie algebra we denote as $\mathfrak{g}$ and Weyl group as $\mathcal{W}$. The complexified Cartan subalgebra $\mathfrak{h}_{\mathbb{C}}$ of the gauge symmetry is parameterized as $u = \{u_i\}_{i=1}^N$, with $N$ being the rank of $G$. Similarly, the Cartan subalgebra for the flavour

---

[2]While this formula is written in terms of the JK residues, the physical derivations of the JK residue formula rely on manipulations of the path integral, and hence it is not obvious if the results agree with the mathematical Jeffrey-Kirwan localization of virtual invariants. This question was recently addressed in [74], which proves the equivalence of the two results for theories with four supercharges. See also [75–78].

symmetry is parametrized by $\epsilon = \{\epsilon_i\}_{i=1}^{F}$, with $F$ being the rank of the flavour symmetry group. The flavour chemical potentials $\epsilon$ will hereafter be called the equivariant parameters since they represent the equivariant torus action on the moduli space. The one-loop determinant $Z_{1\text{-loop}}$ denotes the integrand of the residue, and JK-Res denotes the Jeffrey-Kirwan residue. Now, let us explain the ingredients in more detail.

**The space $\mathfrak{M}$** The space $\mathfrak{M}$ is defined to be $\mathfrak{h}_{\mathbb{C}}/Q^{\vee}$, where $\mathfrak{h}$ is the Cartan subalgebra and $Q^{\vee}$ is the coroot lattice. In the one-loop determinant, each multiplet gives rise to a hyperplane $H_i = \{Q_i(u) + \cdots = 0\} \subset \mathfrak{M}$ with covector $Q_i \in \mathfrak{h}^*$. For example, a chiral multiplet in a representation R of the gauge symmetry leads to $\rho(u) - \epsilon_{\chi} = 0$ where $Q_i = \rho$ is the weight of the representation R. The union of the hyperplanes is $\mathfrak{M}_{\text{sing}} = \bigcup_i H_i$.

To compute the residues, we need to consider $Q(u^*)$, the set of $Q_i$ meeting at $u^* \in \mathfrak{M}^*_{\text{sing}}$. Here, $\mathfrak{M}^*_{\text{sing}}$ is the set of isolated points where at least $N$ linearly independent hyperplanes meet. If the singularities are non-degenerate (namely, the number of hyperplanes meeting at $u^*$ is equal to the total rank $N$ of the gauge group), then the residue computations can be performed straightforwardly. However, the singularities could be degenerate in generic cases.[3] Moreover, the arrangement of the hyperplanes could even be non-projective: by projectivity, we mean that the (co)vectors $Q_i$ all belong to a half-space. In general, a systematic procedure to resolve this issue is not known to the best of our knowledge.[4] In this paper, we will focus on the cases where this can be done by uplifting with enough equivariant parameters, as discussed below in the main context.

The covector $\eta \in \mathfrak{h}^*$ picks out the allowed sets of hyperplanes in the JK residue. This is given by the positivity condition:

$$\eta \in \text{Cone}(Q_{i_j}) := \left\{ \sum_{j=1}^{N} a_j Q_{i_j} \,\middle|\, a_j \geq 0 \right\}. \tag{2}$$

We shall refer to the hyperplanes/poles satisfying this positivity condition as admissible hyperplanes/poles (and inadmissible otherwise). In particular, for cyclic chambers where the crystal structures are well-known (at least for those arising from toric singularities), the choice is given by $\eta = (1, 1, \ldots, 1)$.

**The Jeffrey-Kirwan residue** The JK residue [7] (see also [79]) is defined by

$$\text{JK-Res} \frac{dQ_{i_1}(u)}{Q_{i_1}(u)} \wedge \cdots \wedge \frac{dQ_{i_N}(u)}{Q_{i_N}(u)} := \begin{cases} \text{sgn}(\det(Q_{i_1}, \ldots, Q_{i_N})), & \eta \in \text{Cone}(Q_{i_j}), \\ 0, & \text{otherwise.} \end{cases} \tag{3}$$

This can be rewritten as

$$\text{JK-Res} \frac{du_1 \wedge \cdots \wedge du_N}{Q_{i_1}(u) \ldots Q_{i_N}(u)} = \begin{cases} \dfrac{1}{|\det(Q_{i_1}, \ldots, Q_{i_N})|}, & \eta \in \text{Cone}(Q_{i_j}), \\ 0, & \text{otherwise.} \end{cases} \tag{4}$$

There is an equivalent constructive definition of the JK residue as a sum of iterated residues [80]. For each $u^*$ with $(Q_{i_1}, \ldots, Q_{i_N})$, we consider a flag

$$\mathcal{F} = [\mathcal{F}_0 = \{0\} \subset \mathcal{F}_1 \subset \ldots \mathcal{F}_N], \qquad \dim \mathcal{F}_j = j, \tag{5}$$

---

[3]In such cases, the strategy is often to use the constructive definition discussed below.
[4]One way to deal with this is to slightly deform the pole $u^*$ so that it would split into multiple projective ones. Some examples can be found in [3,4].

where the vector space $\mathcal{F}_j$ at level $j$ is spanned by $\{Q_{i_1}, \ldots, Q_{i_j}\}$. For the set $\mathcal{FL}(Q(\boldsymbol{u}^*))$ of flags, we choose the subset

$$\mathcal{FL}^+(Q(\boldsymbol{u}^*)) := \left\{ \mathcal{F} \in \mathcal{FL}(Q(\boldsymbol{u}^*)) \,\middle|\, \eta \in \mathrm{Cone}\left(\kappa_1^{\mathcal{F}}, \ldots, \kappa_N^{\mathcal{F}}\right) \right\}, \tag{6}$$

with

$$\kappa_j^{\mathcal{F}} := \sum_{Q_i \in Q(\boldsymbol{u}^*) \cap \mathcal{F}_j} Q_i. \tag{7}$$

The JK residue can be obtained by

$$\mathrm{JK\text{-}Res}(Q(\boldsymbol{u}^*), \eta) = \sum_{\mathcal{F} \in \mathcal{FL}^+} \mathrm{sgn}\left(\det\left(\kappa_1^{\mathcal{F}}, \ldots, \kappa_N^{\mathcal{F}}\right)\right) \mathrm{JK\text{-}Res}_{\mathcal{F}}. \tag{8}$$

In this expression, the iterated residue $\mathrm{JK\text{-}Res}_{\mathcal{F}}$ is defined as follows. Given an $N$-form $\omega = \omega_{1,\ldots,N}\, \mathrm{d}u_1 \wedge \cdots \wedge \mathrm{d}u_N$, choose new coordinates

$$\widetilde{u}_j = Q_{i_j} \cdot u, \qquad j = 1, \ldots, N, \tag{9}$$

such that $\omega = \widetilde{\omega}_{1,\ldots,N}\, \mathrm{d}\widetilde{u}_1 \wedge \cdots \wedge \mathrm{d}\widetilde{u}_N$. The contribution to the JK residue from a flag is then

$$\mathrm{JK\text{-}Res}_{\mathcal{F}}\, \omega = \mathrm{Res}_{\widetilde{u}_r = \widetilde{u}_r^*} \ldots \mathrm{Res}_{\widetilde{u}_1 = \widetilde{u}_1^*} \widetilde{\omega}_{1,\ldots,N} = J\left(\frac{\partial \widetilde{u}_i}{\partial u_j}\right) \mathrm{Res}_{u_r = u_r^*} \ldots \mathrm{Res}_{u_1 = u_1^*} \omega_{1,\ldots,N}, \tag{10}$$

where $J$ denotes the Jacobian.

**Quiver gauge theories**  In this paper, we assume for concreteness that

- A quiver gauge theory is defined from a finite quiver $Q$, i.e., a finite oriented graph. We will denote the set of vertices of $Q$ by $Q_0$.

- The quiver nodes $a \in Q_0$ has unitary gauge groups $\mathrm{U}(N_a)$ for some non-negative integer $N_a$.

- We also have framing (i.e. flavour) node, extending the quiver $Q$ to $^\sharp Q$. In practice we will concentrate on examples with a single framing node.

- We assume that any framing node has a U(1) gauge symmetry. This means that all the chemical potentials are associated with U(1) symmetries.

- The quiver arrows represent the bifundamental matters, or the (anti-)fundamental matters if one of the nodes is a framing node.

- The $F$-term or $J$-/$E$-term relations are polynomial relations involving the fields.

It is straightforward to see from our discussions below that some of our results generalize straightforwardly even when some of our assumptions are lifted. Our assumptions above, however, already includes a huge number of examples.

Under these assumptions, the matter contents transform under the (bi)fundamental or adjoint representations $\mathbb{R}$ (of the corresponding gauge nodes that the edges connect). In particular, the root system $\Phi$ coincides with the set of non-zero weights given by the adjoint representation. In the following, $N$ denotes the ranks of gauge group $G$.

**The one-loop determinant**   It is useful to introduce the function[5]

$$\zeta(z) = \begin{cases} \dfrac{i\theta_1(\tau,z)}{\eta(\tau)}, & \text{elliptic,} \\ 2i\sin(\pi z), & \text{trigonometric,} \\ z, & \text{rational,} \end{cases} \tag{11}$$

as well as the elliptic functions

$$\eta(\tau) = q^{1/24}\prod_{k=1}^{\infty}\left(1-q^k\right), \qquad \theta_1(\tau,z) = -iq^{1/8}y^{1/2}\prod\left(1-q^k\right)\left(1-yq^k\right)\left(1-y^{-1}q^{k-1}\right), \tag{12}$$

with $q = e^{2\pi i\tau}$ and $y = e^{2\pi iz}$. The three choices of $\zeta$ corresponds to the elliptic genera, Witten indices and matrix model partition functions respectively.

**Theories with four supercharges**   Let us write

$$\xi_{\mathcal{N}=4} := \begin{cases} \left(-\dfrac{\eta(\tau)^3}{i\theta_1(\tau,\epsilon)}\right)^N \prod_{i=1}^N d(2\pi iu_i), & \text{elliptic,} \\ \left(-\dfrac{1}{2i\sin(\pi\epsilon)}\right)^N \prod_{i=1}^N d(2\pi iu_i), & \text{trigonometric,} \\ \left(-\dfrac{1}{\epsilon}\right)^N \prod_{i=1}^N du_i, & \text{rational.} \end{cases} \tag{13}$$

The integrand $Z_{\text{1-loop}}$ factorizes into the contributions from the vector/chiral multiplet contributions:

$$Z_{\text{1-loop}}(u) = \prod_V Z_V(\epsilon,u)\prod_\chi Z_\chi(\epsilon,u), \tag{14}$$

where we have the following:

- Vector multiplet:

$$Z_V(\epsilon,u) = \xi_{\mathcal{N}=4}\prod_{\alpha\in\Phi(G)}\frac{-\zeta(\alpha(u))}{\zeta(\alpha(u)+\epsilon)} = \xi_{\mathcal{N}=4}\prod_{i\neq j}^N\frac{-\zeta(u_i-u_j)}{\zeta(u_i-u_j+\epsilon)}. \tag{15}$$

- Chiral multiplet with U(1) symmetry charge $\epsilon_\chi$:

$$Z_\chi(\epsilon,u) = \prod_{\rho\in R}\frac{-\zeta(\rho(u)+\epsilon-\epsilon_\chi)}{\zeta(\rho(u)-\epsilon_\chi)} \tag{16}$$

$$= \begin{cases} \displaystyle\prod_{i=1}^{N_s}\prod_{j=1}^{N_t}\frac{-\zeta\left(u_j^{(t)}-u_i^{(s)}+\epsilon-\epsilon_\chi\right)}{\zeta\left(u_j^{(t)}-u_i^{(s)}-\epsilon_\chi\right)}, & \text{(bi)fundamental from } s \text{ to } t\,, \\ \left(\dfrac{-\zeta(\epsilon-\epsilon_\chi)}{\zeta(-\epsilon_\chi)}\right)^N \displaystyle\prod_{i\neq j}^N\frac{-\zeta(u_i-u_j+\epsilon-\epsilon_\chi)}{\zeta(u_i-u_j-\epsilon_\chi)}, & \text{adjoint.} \end{cases}$$

---

[5]The trigonometric version of $\zeta(z)$ can be obtained by taking the limit $q\to 0$ of the elliptic expression. Therefore, there should be an extra factor $q^{1/12}$ in the trigonometric $\zeta(z)$. Nevertheless, we shall always consider the cases where such extra $q^{1/12}$ get cancelled in the full expression of the one-loop determinant (which would be automatic for the theories with four supercharges). As a result, the extra factor $2\pi i$ when taking the rational limit from the trigonometric version can also be omitted. This is the case for $\xi_{\mathcal{N}=2}$ defined below as well.

In these expressions, $\epsilon$ denotes the fugacity for an R-symmetry that rotates the two extra supercharges for the $\mathcal{N}=4$ supersymmetry compared with their $\mathcal{N}=2$ counterparts. In the literature of toric geometry, we often choose a parametrization of the equivariant parameters $\epsilon_k$ such that $\epsilon = \sum_k \epsilon_k$; the index with $\epsilon = 0$ (resp. $\epsilon \neq 0$) defines the unrefined (resp. refined) indices.

For theories with four supercharges, to make the one-loop determinant non-trivial, we first need to keep $\epsilon \neq 0$ in the JK residue computations [38], even if one is eventually interested in the $\epsilon = 0$ case; otherwise the one-loop determinant trivializes if we set $\epsilon = 0$ inside the integrand of the JK residue.

**Theories with two supercharges**   Let us write

$$
\xi_{\mathcal{N}=2} := \begin{cases} \eta(\tau)^{2N} \prod_{i=1}^{N} \mathrm{d}(2\pi\mathrm{i}u_i), & \text{elliptic,} \\[2mm] \prod_{i=1}^{N} \mathrm{d}(2\pi\mathrm{i}u_i), & \text{trigonometric,} \\[2mm] \prod_{i=1}^{N} \mathrm{d}u_i, & \text{rational.} \end{cases}
\tag{17}
$$

The integrand $Z_{\text{1-loop}}$ factorizes into the contributions from the vector/chiral/Fermi multiplet contributions:

$$
Z_{\text{1-loop}}(u) = \prod_V Z_V(\epsilon, u) \prod_\chi Z_\chi(\epsilon, u) \prod_\Lambda Z_\Lambda(\epsilon, u),
\tag{18}
$$

where we have the following:

- Vector multiplet:

$$
Z_V(\epsilon, u) = \xi_{\mathcal{N}=2} \prod_{\alpha \in \Phi(G)} (-\zeta(\alpha(u))) = \xi_{\mathcal{N}=2} \prod_{i \neq j}^{N} (-\zeta(u_i - u_j)).
\tag{19}
$$

- Chiral multiplet:

$$
\begin{aligned}
Z_\chi(\epsilon, u) &= \prod_{\rho \in \mathrm{R}} \frac{1}{\zeta(\rho(u) - \epsilon_\chi)} \\
&= \begin{cases} \displaystyle\prod_{i=1}^{N_s} \prod_{j=1}^{N_t} \frac{1}{\zeta\left(u_j^{(t)} - u_i^{(s)} - \epsilon_\chi\right)}, & \text{(bi)fundamental from } s \text{ to } t, \\[4mm] \displaystyle\frac{1}{\zeta(-\epsilon_\chi)^N} \prod_{i \neq j}^{N} \frac{1}{\zeta(u_i - u_j - \epsilon_\chi)}, & \text{adjoint.} \end{cases}
\end{aligned}
\tag{20}
$$

- Fermi multiplet:

$$
Z_\Lambda(\epsilon, u) = \prod_{\rho \in \mathrm{R}} (-\zeta(\rho(u) - \epsilon_\Lambda))
\tag{21}
$$

$$
= \begin{cases} \displaystyle\prod_{i=1}^{N_s} \prod_{j=1}^{N_t} \left(-\zeta\left(u_j^{(t)} - u_i^{(s)} - \epsilon_\Lambda\right)\right), & \text{(bi)fundamental from } s \text{ to } t, \\[4mm] (-\zeta(-\epsilon_\Lambda))^N \displaystyle\prod_{i \neq j}^{N} \left(-\zeta(u_i - u_j - \epsilon_\Lambda)\right), & \text{adjoint,} \end{cases}
$$

In the quivers, the Fermi multiplets, whose charges are denoted by $\epsilon_\Lambda$, are not oriented, but we need to choose a "head" and a "tail" of the edge. Readers are referred to [38] on how this "orientation" can be chosen (at least for the toric cases). Once the orientation is determined for one Fermi multiplet, the others are automatically fixed. Moreover, different choices always give the same partition function.

Let us also remark that the quivers in this paper are always framed. In other words, there are round and square nodes denoting the gauge and flavour nodes respectively. When we have an edge connecting a flavour node and a gauge node, it transforms as the fundamental or anti-fundamental under the gauge group. Its contribution to the one-loop determinant is simply given by $Z_{\text{matter}}$, where the corresponding chemical potential is set to 0, and we shall use $v_i$ to denote its weight.[6] For instance, in an $\mathcal{N} = 4$ theory, a chiral of weight $v_1$ pointing from a U(1) flavour node to a U($N$) gauge node (labelled by $a$) has the contribution

$$Z_\chi = \prod_{i=1}^{N} \frac{-\zeta\left(u_i^{(a)} + \epsilon - v_1\right)}{\zeta\left(u_i^{(a)} - v_1\right)}. \tag{22}$$

## 3 Crystals from JK residues

For toric $\mathcal{N} = 4$ quivers in the cyclic chambers, the BPS counting problem has an underlying combinatorial structure known as the crystal melting [21–23, 31, 81]. In particular, molten crystals correspond one-to-one with the torus fixed points of the moduli space, and hence, counting the BPS states can be translated into counting the crystal configurations. Moreover, this encapsulates the general ranks of the gauge groups via the melting of the crystals, connecting different BPS states/fixed points in the moduli space.

This section aims to generalize this discussion to a much larger class of quiver gauge theories. We will find that we encounter many interesting subtleties which are not present in the toric examples.

### 3.1 Definition of crystals

Let us now spell out the types of quiver gauge theories we consider and what the crystals mean for these quivers. We shall define the crystal as a graph, where a vertex is called an atom and an arrow is called a chemical bond between atoms.

To explain the basic idea, suppose that there are only simple poles in the one-loop determinant. When computing the partition function at level $N$ (where $N$ is the rank of the total gauge group), the non-vanishing indices are completely determined by the poles in the JK residue, which for bifundamental matters are determined by the hyperplanes $\{u_j^{(a)} - u_i^{(b)} - \cdots = 0\}$. Similar to the toric cases studied in [33, 38], these contributions have a partial ordering as can be seen from the constructive definition of the JK residue formula. In other words, the partial ordering is exactly given by the one for the corresponding flags $\mathcal{F}$ in (5) and (6). As a result, an index at level $N$ can be obtained from some index at level $N-1$ with an extra admissible pole $(u_N^{(a)} - u_i^{(b)} - \dots)^{-1}$, and this process can be performed inductively. Roughly speaking, a crystal is a collection of such poles, and this inductive process implements how the crystal grows.

Suppose moreover that we choose the dimension vector $\boldsymbol{N} = (N_a)_{a \in Q_0}$ and fix the covector $\eta_{\boldsymbol{N}}$, a vector with $N = \sum_a N_a$ components. In the following, we will consider not only fixed

---

[6]Alternatively, we can assign some chemical potential, say $u_f$, to a flavour node. Since this is not integrated over in the contour integral, we can always absorb it into the edge weights $v_i$.

values of $N$, but rather vary them over the whole of $\mathbb{Z}_{\geq 0}^{|Q_0|}$. Correspondingly, we need to fix an infinite-size covector $\eta$, whose truncations generate the aforementioned $\eta_N$. We will fix this covector throughout the rest of this discussion.

Let us denote by $\mathfrak{U}(N, \eta)$ the set of $\boldsymbol{u}^* = \left(u_i^{(a)*}\right) \in \mathfrak{M}_{\mathrm{sing}}$ such that:

- The JK residue at $\boldsymbol{u} = \boldsymbol{u}^*$ is non-zero and contributes non-trivially (i.e. is admissible) under the covector $\eta$.

By definition, each $\boldsymbol{u}^* \in \mathfrak{U}(N, \eta)$ is an isolated point where at least $\sum\limits_{a \in Q_0} N_a$ hyperplanes meet. Let $\mathfrak{H}(\boldsymbol{u}^*)$ be the set of all hyperplanes meeting at $\boldsymbol{u}^*$.

In our discussion, since we consider the quiver gauge theories with gauge and flavour nodes, a hyperplane in $\mathfrak{H}(\boldsymbol{u}^*)$ takes the form

$$\left\{ u_j^{(a)} - u_i^{(b)} - \cdots = 0 \right\}, \qquad (a, b \in Q_0, \quad i \in \{1, \ldots, N_a\}, \quad j \in \{1, \ldots, N_b\}), \tag{23}$$

if it is associated with a bifundamental/adjoint matter or a vector multiplet, or

$$\left\{ u_j^{(a)} - \cdots = 0 \right\}, \qquad (a \in Q_0, \quad i \in \{1, \ldots, N_a\}), \tag{24}$$

for an (anti-)fundamental matter — in these expressions, the ellipsis denotes the $u$-independent linear combinations of the residual fugacities $\epsilon_k$'s. At the intersection of the hyperplanes, we find that each $u_i^{(a)*}$ is a linear combination of these $\epsilon_k$'s:

$$u_i^{(a)*} \in \bigoplus_{k=1}^{F} \mathbb{Z}\,\epsilon_k, \qquad \text{for each } a \in Q_0, \quad i \in \{1, \ldots N_a\}. \tag{25}$$

We collect all of the $u_i^{(a)*}$'s over $i$ and $a$ to define

$$\mathscr{A}(\boldsymbol{u}^*) := \left\{ u_i^{(a)*} \,\middle|\, a \in Q_0; i = 1, \ldots, N_a \right\} \subset \bigoplus_{k=1}^{F} \mathbb{Z}\,\epsilon_k, \tag{26}$$

inside the lattice $\bigoplus\limits_{k=1}^{F} \mathbb{Z}\,\epsilon_k$. We moreover collect these sets as

$$\mathscr{A}(N, \eta) := \bigcup_{\boldsymbol{u}^* \in \mathfrak{U}(N, \eta)} \mathscr{A}(\boldsymbol{u}^*), \tag{27}$$

which is called the *set of atoms at level $N$*. Instead of specifying the level by a collection of integers $(N_a)_{a \in Q_0}$, we can specify the level by a single integer $N = \sum\limits_{a \in Q_0} N_a$ to define the *set of atoms at level $N$*:

$$\mathscr{A}(N, \eta) := \bigcup_{N : \sum_a N_a = N} \mathscr{A}(N, \eta). \tag{28}$$

We further define the *full set of atoms* as

$$\mathscr{A}(\eta) := \bigcup_{N \in \mathbb{Z}_{\geq 0}^{|Q_0|}} \mathscr{A}(N, \eta) = \bigcup_{N \in \mathbb{Z}_{\geq 0}} \mathscr{A}(N, \eta). \tag{29}$$

In other words, $\mathscr{A}(\eta)$ is defined by a collection of $u_i^{(a)}$'s which arises as one of the components of the singular point $\boldsymbol{u}^*$, for some dimension vector $N$. This is in general either a finite or an infinite set, depending on the choice of $\eta$ and the gauge theory.

Note that we have

$$\mathscr{A}(\boldsymbol{M}, \eta) \subseteq \mathscr{A}(\boldsymbol{N}, \eta), \qquad \text{when} \qquad \boldsymbol{M} \le \boldsymbol{N}, \tag{30}$$

where the partial ordering $\boldsymbol{M} \le \boldsymbol{N}$ is defined as

$$M_a \le N_a, \quad \text{for all} \qquad a \in Q_0. \tag{31}$$

This follows by definition since if we want to evaluate the residues of $N$ variables, we can first evaluate the residues for the first $M$ variables and take the singularities only involving the first $M$ variables. The inclusion (30) means that the definition should rather be regarded as a direct limit with respect to the partial ordering:

$$\mathscr{A}(\eta) = \varinjlim \mathscr{A}(\boldsymbol{N}, \eta). \tag{32}$$

Now the *crystal* $\mathscr{C}(\eta) = (\mathscr{A}(\eta), \mathscr{I}(\eta))$ is defined as an oriented *weighted* graph with vertices $\mathscr{A}(\eta)$ and arrows $\mathscr{I}(\eta)$ given as follows:

- The collection of vertices is given by the full set of atoms $\mathscr{A}(\eta)$. A vertex will be called an *atom*,[7] which we denote as $\mathfrak{a}, \mathfrak{b}, \cdots$.

- Suppose that we have two atoms $\mathfrak{a}$ and $\mathfrak{b}$. We draw an arrow $I$ from $\mathfrak{b}$ to $\mathfrak{a}$ if the following two conditions are satisfied:

    - First, there exists $\boldsymbol{N} \in \mathbb{Z}_{\ge 0}^{|Q_0|}$ and $\boldsymbol{u}^* \in \mathfrak{U}(\boldsymbol{N}, \eta)$ such that $\mathfrak{a}, \mathfrak{b} \in \mathscr{A}(\boldsymbol{u}^*)$.

Note that without this condition, $\mathfrak{a}$ and $\mathfrak{b}$ are in general contained in $\mathscr{A}(\boldsymbol{M}, \eta)$ for $\boldsymbol{M}$ and $\boldsymbol{u}^*$ of different values.[8] Under the first condition, we can write $\mathfrak{a} = u_i^{(a)*}, \mathfrak{b} = u_j^{(b)*}$ for some $a, b \in Q_0$ and $a \in \{1, \ldots, N_a\}, b \in \{1, \ldots, N_b\}$. We can then state the second condition:

    - Second, there exists a hyperplane of the form

$$\left\{ u_j^{(a)} - u_i^{(b)} - \cdots = 0 \right\} \tag{33}$$

    inside $\mathfrak{H}(\boldsymbol{u}^*)$.

In the atomic terminology, the arrows are also called the chemical bonds.[9]

Note that the graph thus defined is a weighted graph, i.e., each vertex/edge of the graph has an associated point in the lattice $\bigoplus_{k=1}^{F} \mathbb{Z}\epsilon_k$ of equivariant parameters. The weight $\epsilon(\mathfrak{a})$ of a vertex $\mathfrak{a}$ is by definition determined by a point in the lattice, and that of an arrow $I$ connecting two vertices from $\mathfrak{a}$ to $\mathfrak{b}$ are given by the differences of the weights for the two vertices: $\epsilon_I = \epsilon(\mathfrak{b}) - \epsilon(\mathfrak{a})$.

## 3.2 No-overlap condition

For the cases considered in this paper, we also need an extra condition called the no-overlap condition. With the definition of the crystal above, we can now formulate the no-overlap condition.

---

[7]Given more general quivers and/or general chambers, the structures for counting are dubbed various names, such as poset representations in [54] and glasses in [82]. Here, for convenience, we shall simply borrow the terminologies from the toric cases and refer to the states as "crystals" with "atoms" of different "colours" corresponding to different nodes in the quiver.

[8]We can always choose a common $\boldsymbol{M}$ by choosing $\boldsymbol{M}$ to be sufficiently large in the partial ordering. It is not guaranteed, however, if we can choose the same $\boldsymbol{u}^*$.

[9]When we plot the crystals below, we shall always omit the chemical bonds from the framing nodes, and draw the initial atoms in a different colour.

Suppose that we are given an atom $\mathfrak{a}$ in $\mathscr{A}(\eta)$ such that $\mathfrak{a} \in \mathscr{A}(N, \eta)$ for some $N$. In general, this can have multiple realizations inside $\mathscr{A}(N, \eta)$, so that we have

$$\mathfrak{a} = u_i^{(a)*} = u_j^{(b)*}, \tag{34}$$

for a different pair $(i, a), (j, b)$ (in other words, either $i \neq j$, or $a \neq b$ if $i = j$), but with the same $\boldsymbol{u}^* \in \mathscr{A}(N, \eta)$. If this ever happens for some $N$, we say that the no-overlap condition is violated; otherwise, we say that the no-overlap condition is satisfied.

Recall that the points of $\mathscr{A}(N, \eta)$ are the values of the Cartan elements in the evaluations of the JK residues for the dimension vector $N$ — the no-overlap condition dictates that the Cartan elements never come back to the same point while evaluating the residue, inside the lattice $\bigoplus_{k=1}^{F} \mathbb{Z}\epsilon_k$, for any choice of $N$. This is equivalent to the condition that any $\mathscr{A}(\boldsymbol{u}^*)$ contains $\sum_{a \in Q_0} N_a$ different elements for any $\boldsymbol{u}^* \in \mathfrak{U}(N, \eta)$ for any dimension vector.[10]

In the following, we impose the no-overlap condition as one of the crucial ingredients in our discussions:

- The crystal satisfies the no-overlap condition.

This condition ensures that there are only simple poles in the one-loop determinants, and hence the validity of the JK residue formula.

Notice that here the crystal is constructed from the computation of the partition functions via the JK residue formula. As shown in [38], the JK residue formula recovers the crystals for toric quivers, both to Calabi-Yau threefolds and fourfolds. Therefore, our definition here is consistent with the crystal for the toric cases. In the following, we will often suppress the dependence on $\eta$ from the notations and denote e.g. $\mathscr{A}$ and $\mathscr{C}$.

### 3.3 Molecules and melting rules

Given the crystal $\mathscr{C} = (\mathscr{A}, \mathscr{I})$, we can define the *molecule* as a finite subset $\mathscr{M} \subset \mathscr{A}$ such that the following condition (melting rule) is satisfied:

Suppose that $\mathfrak{a}, \mathfrak{b} \in \mathscr{A}$ with an arrow $I \in \mathscr{I}$ connecting from $\mathfrak{a}$ to $\mathfrak{b}$.
If $\mathfrak{b} \in \mathscr{M}$, then $\mathfrak{a} \in \mathscr{M}$. $\tag{35}$

In the literature of the quiver algebras (and in particular the quiver Yangians), a molecule here is often called a *crystal state*, since it will span the weight space for the crystal-melting representation of the algebra, and it is a natural terminology in this context. In this paper, we use the words "molecule" and "crystal state" interchangeably, and relatedly a molecule (i.e., a crystal state) will be denoted as $\mathscr{C}$ in later sections, to make comparisons with the existing literature easier. Note that the complement $\mathscr{C} \setminus \mathscr{M}$ is called a *molten crystal*.

Given a molecule, we can enlarge the crystal by "adding an atom". Suppose that we have a molecule $\mathscr{M}$, as well as an arrow connecting $\mathfrak{a} \in \mathscr{M}$ to $\mathfrak{b} \notin \mathscr{M}$. We can then define a new molecule by $\mathscr{M} \sqcup \{\mathfrak{b}\}$. We can repeat this process and add multiple atoms to the molecule.

Conversely, suppose that a molecule $\mathscr{M}$ contains an atom $\mathfrak{a}$ such that there exists no atom $\mathfrak{b} \in \mathscr{M}$ with an arrow pointing from $\mathfrak{a}$ to $\mathfrak{b}$. We can then simply remove $\mathfrak{a}$ from the molecule

---

[10]More intuitively, the no-overlap condition states that the atoms are not allowed to overlap in the crystal, as will be clearer when we discuss examples below. Note, however, that in our technical definition such overlapped atoms are already identified as points in the lattice when we defined $\mathscr{A}$.

$\mathscr{M}$ to define a smaller molecule $\mathscr{M}\setminus\{\mathfrak{a}\}$. We can again repeat this process to remove multiple atoms from the molecule. We can of course more generally consider a more complicated process where atoms are both added and removed at multiple steps. Eventually, one expects that we can create any molecule from nothing so that a molecule can be considered as a composite of a collection of atoms.

Note that the growth of the crystal corresponds to a collection of arrows in the quiver diagram. This follows from the fact that there is an equivalent constructive definition of the JK residue as reviewed in §2. The partition function at level $N+1$ can be computed from the one at level $N$:

$$Z_{\text{1-loop}}(u_1,\ldots,u_{N+1}) = Z_{\text{1-loop}}(u_1,\ldots,u_N)\Delta Z(u_1,\ldots,u_{N+1}), \tag{36}$$

and the poles in $\Delta Z$ are given by the chiral multiplets corresponding to the arrows in the quiver. Therefore, we can still have the melting rule which requires all its precedents to be present for an atom to appear in the crystal.

In practice, we will discuss examples of molecules which are expressed as $\mathscr{A}(\boldsymbol{N},\eta)$ for a dimension vector $\boldsymbol{N}$. Since the BPS indices themselves are defined with respect to the fixed dimension vector $\boldsymbol{N}$, it is natural that the molten crystal configurations for fixed levels $\boldsymbol{N}$ are the ones directly relevant for the BPS state counting problem.

Of course, it is non-trivial to verify that such a set satisfies the melting rule, and indeed, whether this is satisfied or not depends crucially on the values of $\eta$. This is what we discuss next.

After taking the JK residue for the partition function $\mathcal{Z}$, we need to make sure that the poles at both level $N$ and level $N+1$ are both admissible: the crystal $\mathscr{C}+\mathfrak{a}$ at level $N+1$ being allowed by $\eta_{N+1}$ does not necessarily guarantee that the crystal state $\mathscr{C}$ at level $N$ is allowed by $\eta_N$ (where the subscript $N$ of $\eta$ denotes the truncation of the infinite $\eta$ onto the first $N$ elements). Schematically, we can have the situation

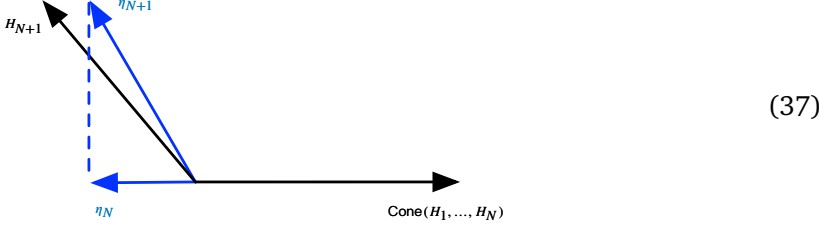

$$\tag{37}$$

where the horizontal black arrow indicates the cone generated by the hyperplanes $H_1,\ldots,H_N$. As we can see, $\eta_{N+1}$ lies in the cone generated by $H_1,\ldots,H_{N+1}$ while its projection $\eta_N$ is not in $\text{Cone}(H_1,\ldots,H_N)$. In this case, we can have the crystal state $\mathscr{C}+\mathfrak{a}$, but not $\mathscr{C}$, thus violating the melting rule. In this case, we may still grow the crystal, but the minimal step of such growth may involve multiple atoms.

To avoid such situations, one should require that the minimal step of melting is always one single atom. Then given a cone formed by the admissible hyperplanes, the projections of $\eta$ onto lower dimensions would always lie in the lower-dimensional cones:

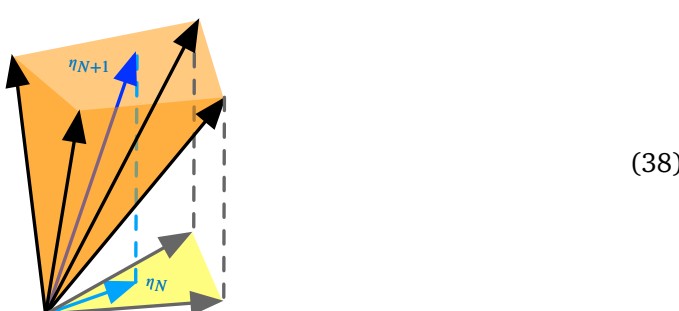

$$\tag{38}$$

We shall refer to a chamber (in the space of covectors $\eta$) whose corresponding $\eta$ satisfies this condition as a cyclic chamber.

For the quivers satisfying our conditions above, the choice $\eta = (1, 1, \ldots, 1)$ always gives cyclic chambers.[11] This can be seen as follows. Let us denote $\delta(k_1, -k_2)$ as the vector with the element 1 (resp. $-1$) at the $k_1^{\text{th}}$ (resp. $k_2^{\text{th}}$) entry and 0 elsewhere. Without loss of generality, we may assume that the hyperplanes are ordered such that $H_i = \delta(i, -(i-1))$ with $H_1 = (1, 0, \ldots, 0)$ being the only hyperplane with no negative entries. Suppose that at level $N + 1$, we have

$$\eta_{N+1} = (1, \ldots, 1) = \sum_{i=1}^{N+1} a_i H_i, \qquad a_i \geq 0, \tag{39}$$

with $a_i = N - i + 2$. If we project out the last hyperplane $H_{N+1}$, the covector $\eta_N$ still lives in the cone generated by the first $N$ hyperplanes since

$$\eta_N = \sum_{i=1}^{N} a_i' H_i, \tag{40}$$

has the solution $a_i' = N - i + 1$.

This process can also be reversed to obtain the level-$(N+1)$ admissibility from that for the level $N$. Suppose that we have $\eta_N = \sum_{i=1}^{N} b_i H_i$ with $b_i \geq 0$. We can then express $\eta_{N+1}$ as

$$\eta_{N+1} = \eta_N + \sum_{i=1}^{N+1} H_i = \sum_{i=1}^{N+1} b_i' H_i, \tag{41}$$

where $b_i' = b_i + 1$ $(i = 1, \ldots, N)$ and $b_{N+1}' = 1$.

The choice $\eta = (1, \ldots, 1)$ further ensures that no poles originating from the vector multiplets would contribute to the JK residue as discussed in [38] — the discussions in the main context of that paper was for toric Calabi-Yau fourfolds while the cases for toric Calabi-Yau threefolds were argued in the appendix. This means that the crystal structure can be understood entirely from the structure of the quiver arrows and the relations satisfied by them, as we will discuss next.

## 3.4 Crystals from path algebras — Modules of truncated Jacobi algebras

Given a toric quiver and the cyclic chamber, crystals can be constructed as the modules of the Jacobi algebra [22] associated with the quiver with superpotentials. Let us next discuss the generalization of this statement.

Suppose that we have a molecule $\mathcal{M}$. Then its complement $\mathcal{C} \setminus \mathcal{M}$ defines a module of the path algebra $\mathbb{C}Q$ of the quiver $Q$. Here, the path algebra is an algebra whose elements are the $\mathbb{C}$-linear combinations of (in general open) finite paths of the quiver, and their products are defined by the concatenations of the paths (which are defined to be zero when one cannot concatenate two paths). Indeed, given a molten crystal $\mathcal{C} \setminus \mathcal{M}$ one can define a formal vector space

$$M = \sum_{\mathfrak{a} \in \mathcal{C} \setminus \mathcal{M}} \mathbb{C}\mathfrak{a}. \tag{42}$$

---

[11]For any general vectors, $\eta = (1, 1, \ldots, 1)$ does not have to satisfy this condition. For instance, $(1, 1, 1)$ lies in the cone generated by $(1, 0, 0)$, $(0, -1, 0)$ and $(0, 2, 1)$. However, $(1, 1)$ is certainly not in the cone generated by $(1, 0)$ and $(0, -1)$.

An arrow $I$ inside the path algebra has an associated hyperplane and hence an arrow in the crystal as explained above — we denote the latter by the same symbol $I$ for notational simplicity. Assuming we choose $\eta = (1, \ldots, 1)$, we can then define an action of $I$ on $\mathfrak{a} \in M$ as $I \cdot \mathfrak{a}$ as explained above, and this is linearly extended to the whole of $M$.

While the path algebra $\mathbb{C}Q$ is defined solely from the quiver, physics setup dictates that the module should be compatible with the $F$-term (for $\mathcal{N} = 4$ supersymmetry) or the $J$-/$E$-term (for $\mathcal{N} = 2$ supersymmetry) relations. This means that we have a module of the *Jacobi algebra* $\mathcal{J}$, which is the truncation of the path algebra defined as

$$
\mathcal{J} := \begin{cases} \mathbb{C}Q/\langle F\text{-terms}\rangle = \mathbb{C}Q/\langle \partial W\rangle, & \mathcal{N} = 4 \text{ supersymmetry,} \\ \mathbb{C}Q/\langle J\text{-terms and } E\text{-terms}\rangle, & \mathcal{N} = 2 \text{ supersymmetry,} \end{cases} \tag{43}
$$

where $W$ is the superpotential of an $\mathcal{N} = 4$ theory.

It turns out that we can in general further truncate this algebra. To explain this, let us consider the $\mathcal{N} = 4$ case with superpotential $W$, which we assume to be a polynomial in terms of the bifundamental (and (anti)fundamental) chiral multiplets. The derivatives of the superpotential generate the relations of the form

$$
\sum_{i=1}^{n} P_i = 0, \tag{44}
$$

where each $P_i$ is a product of the chiral multiplets (times the complex coefficients), represented on the quivers by concatenations of the arrows. Of course, this equation defines one of the relations in the Jacobi algebra.

When we consider theories associated with toric geometries, the relations (44) always involve two monomial terms so that we have (monomial) = (monomial): this is essentially one of the definitions of toric geometry. In general, however, we have more than two monomials in a relation. For example, we may have a relation of the form

$$
X_1 X_2 = X_3 X_4 X_5 + X_6 X_7, \tag{45}
$$

where each $X_i$ represents an arrow of the quiver (and the associated chiral multiplet).

Now, recall that our crystal is defined in the lattice of the equivariant parameters and that the fugacities should be compatible with any relations of the theory, i.e., the $F$-terms and $J$-/$E$-terms. This means that all the terms of the relation (44) should have the same equivariant weights, and should be represented as the same arrow inside the lattice. In other words, we are effectively imposing the stronger relations

$$
P_1 = P_2 = \cdots = P_n. \tag{46}
$$

For example, (45) is replaced by a stronger relation

$$
X_1 X_2 = X_3 X_4 X_5 = X_6 X_7. \tag{47}
$$

This is because the one-loop determinant only sees the weights of the edges coming from the superpotential, and the signs as well as the coefficients are not important.[12]

---

[12]For toric CY cases, the relations of the crystal coincide with the $F$-term or $J$-/$E$-term relations as they are always of the form $P_1 - P_2 = 0$. In general, the two sets of relations could be different. As a simplest example, suppose one $F$-term or $J$-/$E$-term relation is given by $p_1 p_2 + q_1 q_2 = 0$ for some edges $p_i, q_i$. In other words, we have $p_1 p_2 = -q_1 q_2$. If this was a relation of the crystal, then the growth of the crystal would stop at the tails of $p_2$ and $q_2$. This is because if we consider $(p_1 p_2 + q_1 q_2)r$ for some $r$ whose head is the same as the tail of $p_2$, then $q_1 q_2 r = 0$ and hence $p_1 p_2 r = 0$. Of course, the tails of $p_2$ and $q_2$ are different. On the other hand, the relation in the crystal, $p_1 p_2 = q_1 q_2$, would make the two tails coincide. The two relations $p_1 p_2 = \pm q_1 q_2$ are certainly not the same as together they would give $p_1 p_2 = q_1 q_2 = 0$.

We will call (46) as the *enhanced F-term (or J-/E-term) relations*, and define the *truncated Jacobi algebra* $\mathfrak{J}^\sharp$ as the quotient of the path algebra $\mathbb{C}Q$ by the enhanced $F$-term ($J$-/$E$-term) relations:

$$\mathcal{J}^\sharp = \mathbb{C}Q/\langle(46)\rangle. \tag{48}$$

We have $\mathcal{J}^\sharp = \mathcal{J}$ for the toric cases, but in general, $\mathcal{J}^\sharp$ is a truncation of $\mathcal{J}$ as suggested by the terminology.

We have now concluded that a molten crystal (or its complement) is associated with a module of the truncated Jacobi algebra. This statement is beneficial for the analysis of concrete examples in later sections.

## 3.5 Subtleties on equivariant parameters

To satisfy the no-overlap condition, it is crucial to turn on enough equivariant parameters. An extreme situation is a case where all the symmetries of the theory are broken so that there are no $\epsilon_k$'s; the lattice $\bigoplus_{k=1}^{F} \mathbb{Z}\epsilon_k$ then collapses to a point, and everything becomes too degenerate.

The equivariant parameters assign a parameter $\epsilon_I$ to each arrow of the quiver diagram — this includes both chiral multiplets and Fermi multiplets for theories with two supercharges. These parameters are required to be compatible with the $F$-term or $J/E$-term constraints; This in practice means that all the terms $P_i$ in (46) have the same equivariant weights.

Let us for simplicity first concentrate on the theories with four supercharges in the rest of this subsection. The constraint above can then be written as

$$\sum_{I \in P} \epsilon_I = 0, \tag{49}$$

for each monomial term $P$ in the superpotential $W$ (or the $J$-/$E$-interaction in the cases with two supercharges), where the sum is over all the arrows $I$ which appears inside $P$. Such a constraint was called the loop constraint in [40] since a monomial in the superpotential is represented by a loop in a periodic uplift of the quiver diagram.

One expects, however, that there are still redundancies in the parameterization since we can still change the values of the $\epsilon_I$'s by gauge transformations:

$$\epsilon_I \rightarrow \epsilon_I + \varepsilon_a \mathrm{sgn}_a(I), \tag{50}$$

for some parameters $\varepsilon_a$, where $I \in a$ stands for the edges that are connected to the node $a$ and $\mathrm{sgn}_a(I) = \pm 1$ indicates whether $I$ starts from or ends at $a$. To eliminate such gauge redundancies we need to impose gauge-fixing conditions, and we can for example impose the vertex constraints [40]:

$$\sum_{I \in a} \mathrm{sgn}_a(I)\epsilon_I = 0. \tag{51}$$

Indeed, we obtain the correct number of counting only after imposing both the loop and vertex constraints. For example, for toric Calabi-Yau threefolds, we obtain two parameters representing the isometries of the geometry.

One should be careful, however, in imposing the vertex constraints (51), or any other gauge-fixing conditions. Indeed, there are examples (see for example §4.3) where we can satisfy the no-overlap condition only if we do not impose the vertex constraints; while we expect physically that any gauge transformation should not change the Witten index, the one-loop determinants can have poles of higher multiplicities inside the one-loop determinants when we impose the vertex constraints. The correct procedure is then to impose only the loop

constraints, evaluate the residue, and if necessary impose the vertex constraints only after the computation.

The discussion for the cases with two supercharges is similar but with some twists associated with the presence of Fermi multiplets, which play the role of enforcing relations. When the equivariant parameters of the chiral multiplets and the Fermis multiplets are all independent, we may encounter higher-order poles, and relatedly, overlapping atoms; only with the non-trivial $J$-/$E$-term the equivariant parameters for the chiral multiplets and those for the Fermi multiplets are related with each other, leading to cancellations of some poles and hence the no-overlap condition. Recall that we also need to choose orientations of the Fermi multiplets in the one-loop determinant. The indices should, however, be the same if we require the no-overlap condition.

Let us illustrate the importance of the $J$-/$E$-term relations with an example. It is known that the $\mathbb{C}^4$ case has the fixed point labelled by the solid partitions [33]. Let us consider a configuration with three atoms present in the molecule, say along $u_1 = v_1$, $u_2 = v_1 + \epsilon_1$, $u_3 = v_1 + \epsilon_2$. The quiver diagram and the parametrization of the edges can be found in §8.3.1 below. Now, suppose that we would like to add the atom at position $\epsilon_1 + \epsilon_2$ to the molecule. The one-loop determinant at level 4 reads

$$Z_{\text{1-loop}} = \frac{u_4 - u_1 - \epsilon_1 - \epsilon_2}{(u_4 - u_2 - \epsilon_2)(u_4 - u_3 - \epsilon_1)} \times \dots, \tag{52}$$

where the ellipsis denotes the terms that are irrelevant in this illustration. The two terms in the denominator correspond to the two chiral multiplets with equivariant weights $\epsilon_{1,2}$, and the term in the numerator comes from the Fermi with weight $\epsilon_1 + \epsilon_2$. As we can see, this is a simple pole for adding this atom as one of the factors in the denominator is cancelled by this factor in the numerator. If there were no $J$-/$E$-term relations, the weights of these edges would be completely independent. This would give a double pole:

$$Z_{\text{1-loop}} = \frac{u_4 - u_1 - \epsilon_\Lambda}{(u_4 - u_2 - \epsilon_2)(u_4 - u_3 - \epsilon_1)} \times \dots = \frac{u_4 - v_1 - \epsilon_\Lambda}{(u_4 - v_1 - \epsilon_1 - \epsilon_2)^2} \times \dots, \tag{53}$$

where $\epsilon_\Lambda \neq \epsilon_1 + \epsilon_2$. As we can see, for $\mathcal{N} = 4$, adding superpotential could "collapse" a crystal to the one with fewer parameters while for $\mathcal{N} = 2$, removing the $J$-/$E$-interactions could "collapse" the crystal.

## 3.6 An example

To illustrate the above discussions, let us take the simplest non-trivial example which would be the Jordan quiver:[13]

$$\tag{54}$$

with the superpotential $W = 0$. Then the contributions from the vector multiplets and the chiral multiplets are

$$Z_V(\epsilon_k, u) = \xi_{\mathcal{N}=4} \prod_{i \neq j}^N \frac{-\zeta(u_i - u_j)}{\zeta(u_i - u_j + \epsilon)}, \tag{55}$$

and

$$Z_\chi(\epsilon_k, u) = \frac{-\zeta(\epsilon - \epsilon_1)}{\zeta(-\epsilon_1)} \prod_{i \neq j}^N \frac{-\zeta(u_i - u_j + \epsilon - \epsilon_1)}{\zeta(u_i - u_j - \epsilon_1)}, \tag{56}$$

---

[13]This is the quiver associated with $\mathbb{C}$ which is toric.

for the gauge group of rank $N$. We also have the contribution from the framing:

$$Z_{\mathfrak{f}} = \prod_{i=1}^{N} \frac{-\zeta(u_i + \epsilon - v_1)}{\zeta(u_i - v_1)} \,. \tag{57}$$

Let us study examples of $\mathscr{A}(N, \eta)$ for $N = 1, 2, 3$, under the choice $\eta = (1, 1, \ldots, 1)$. In our general discussion, the analysis of $\mathscr{A}(N, \eta)$, or rather its subset $\mathscr{A}(\boldsymbol{N}, \eta)$ (with $\sum_a N_a = N$) requires listing $\boldsymbol{u}^* \in \mathfrak{U}(\boldsymbol{N}, \eta)$. In practice, however, it is easier to list the corresponding set of hyperplanes $\mathfrak{H}(\boldsymbol{u}^*)$ (which of course determines $\boldsymbol{u}^*$ as their intersection points). This is what we do below.

Let us thus list some collections of hyperplanes at the first several levels.

- At level $N = 0$, it is trivially $Z = 1$.

- At level $N = 1$, there is only one hyperplane:

$$\mathfrak{H} = \{u_1 - v_1 = 0\} \,. \tag{58}$$

Hence, the set of atoms,

$$\mathscr{A} = \{u_1^* = v_1\} \,, \tag{59}$$

has only one element, and there is only one possible configuration of the molten crystal.

- At level $N = 2$, we have the following sets of the hyperplanes:

$$\begin{aligned}
\mathfrak{H}_1 &= \{u_1 - v_1 = 0, \ u_2 - u_1 - \epsilon_1 = 0\} \,, \\
\mathfrak{H}_2 &= \{u_2 - v_1 = 0, \ u_1 - u_2 - \epsilon_1 = 0\} \,, \\
\mathfrak{H}_3 &= \{u_1 - v_1 = 0, \ u_2 - v_1 = 0\} \,, \\
\mathfrak{H}_4 &= \{u_1 - v_1 = 0, \ u_1 - u_2 - \epsilon_1 = 0\} \,, \\
\mathfrak{H}_5 &= \{u_2 - v_1 = 0, \ u_2 - u_1 - \epsilon_1 = 0\} \,, \\
\mathfrak{H}_6 &= \{u_1 - u_2 - \epsilon_1 = 0, \ u_2 - u_1 - \epsilon_1 = 0\} \,.
\end{aligned} \tag{60}$$

However, the one in the second line has residue zero, and the sets in the third and fourth lines are ruled out by the covector $\eta$. Therefore, only the first line would contribute. As the two sets of atoms are exactly the same:

$$\mathscr{A}_1 = \mathscr{A}_2 = \{v_1, v_1 + \epsilon_1\} \,, \tag{61}$$

there is only one possible molten crystal with two atoms at $v_1$ and $v_1 + \epsilon_1$, along with a chemical bond pointing from the former atom to the latter.

- At level $N = 3$, there is still only one admissible set

$$\mathfrak{H} = \{u_1 - v_1 = 0, u_2 - u_1 - \epsilon_1 = 0, u_3 - u_2 - \epsilon_1 = 0\} \,, \tag{62}$$

with a non-vanishing JK residue, up to permutations/Weyl group actions. The set of atoms is $\mathscr{A} = \{v_1, v_1 + \epsilon_1, v_1 + 2\epsilon_1\}$. The chemical bonds are given by $v_1 \to v_1 + \epsilon_1$ and $v_1 + \epsilon_1 \to v_1 + 2\epsilon_1$.

The crystal is simply

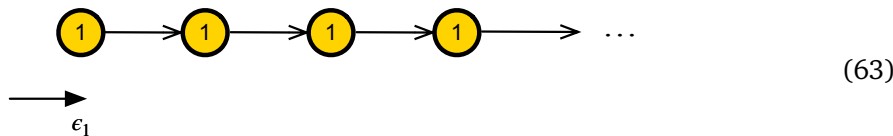

$$\tag{63}$$

As we can see, the crystal in this case is a one-dimensional chain, where the distance between any two neighbouring atoms is equal to $\epsilon_1$. It is straightforward to write down the partition function, which reads

$$\mathcal{Z} = 1 - p + p^2 - p^3 + p^4 - \cdots = \frac{1}{1+p}\,, \tag{64}$$

where $p$ is the dummy variable corresponding to the gauge node. Here, the partition function is of the form

$$\mathcal{Z} = \sum_{N_a} \Omega(\{N_a\}, \{\epsilon_k\}) \boldsymbol{p}^{\boldsymbol{N}}\,, \tag{65}$$

where $\Omega(\{N_a\}, \{\epsilon_k\})$ is the BPS degeneracy and $\boldsymbol{p}^{\boldsymbol{N}} = \prod_a p_a^{N_a}$ with dummy variables $p_a$.

## 4 More examples

Let us now discuss more examples. In general, there can be different edges from the framing node to different gauge nodes which may be related by wall crossings (of the second kind). Together with different choices of $\eta$, we can go to different chambers. For all but one example below, we shall consider the simplest situations as illustrations,[14] where $\eta = (1, \ldots, 1)$. For toric quivers, the computations of the indices and the crystal melting models have been extensively studied in the literature for both CY threefolds and CY fourfolds. Hence, we will not expound such cases in detail here (except for an example with a non-cyclic chamber).

### 4.1 No superpotentials for $\mathcal{N} = 4$

We shall first consider theories with four supercharges where there are no superpotential constraints. Therefore, the number of independent parameters is equal to the number of arrows in the quiver.

**Example 1** Let us take the quiver

$$\tag{66}$$

with $W = 0$. The partition function reads

$$\mathcal{Z} = 1 - p_1 - p_1 p_2 + p_1^2 p_2 + p_1^2 p_2^2 - \ldots\,, \tag{67}$$

where $p_a$ is the variable for the $a^{\text{th}}$ gauge node in the quiver so that $p_a^m$ indicates that there are $m$ atoms of colour $a$ in the molecule. The crystal is

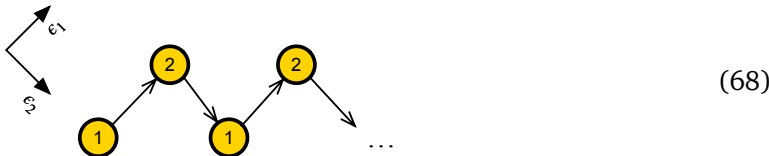

$$\tag{68}$$

Although there are two independent parameters, we have a one-dimensional zigzag crystal in $\mathbb{R}^2$.

---

[14]We shall always take the case with only one chiral from the framing node. The cases with different framings follow exactly the same discussions since they are also some matter contributions in the one-loop determinant.

**Example 2** Let us consider a different quiver

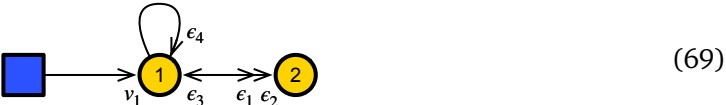

$$(69)$$

with $W = 0$. The partition function reads

$$\mathcal{Z} = 1 - p_1 + p_1^2 - 2p_1 p_2 - p_1^3 + 4p_1^2 p_2 - p_1 p_2^2 + p_1^4 - 8p_1^3 p_2 + 10p_1^2 p_2^2 - \dots \qquad (70)$$

There are four independent parameters $\epsilon_{1,2,3,4}$, and we get a four-dimensional crystal.

## 4.2 Trivial partition functions

Now, we would like to consider the cases when there are not any equivariant parameters. Then the counting would become trivial, and we would expect no BPS states.

Examples of this type can be easily constructed by choosing certain (inhomogeneous) superpotentials. For instance, we can take the same quiver as in (54), but with the superpotential

$$W = X^3 + X^4, \qquad (71)$$

where $X$ denotes the single adjoint loop. Suppose that $X$ has weight $\epsilon_1$. Then the above superpotential implies $3\epsilon_1 = 4\epsilon_1$, i.e., $\epsilon_1 = 0$. As a result, there is no free parameter in this case. Since the $F$-term relation gives $3X^2 + 4X^3 = 0$, a single path/chemical bond does not necessarily vanish. This would then violate the no-overlap condition as the atoms to be added would always be at the same point due to $\epsilon_1 = 0$. Indeed, if we uplift this case with one extra parameter (which would of course violate the superpotential constraints), a double pole would appear at level 2, namely $(u_2 - u_1 + \epsilon_1)^2$.

Nevertheless, in the trivial cases, we can circumvent such issue even without any uplift. Recall that $\epsilon = \sum_k \epsilon_k$ is the sum of all the independent equivariant parameters in the one-loop determinant. The standard process for theories with four supercharges is to take the unrefinement $\epsilon \to 0$ after the evaluation of the residues, so that the integrand would be non-trivial and we can get the refined indices if we do not take this limit. However, in the situation here, the parameter $\epsilon$ is forced to be zero in the one-loop determinant since there are no free parameters. Therefore, we simply have $Z_{1\text{-loop}} = 1$, and hence $\mathcal{Z} = 1$. The crystal is also trivial.

## 4.3 Affine $C_2$ theory

The next example would be a quiver of affine Dynkin type. The one associated to $C_2^{(1)}$ is [83]

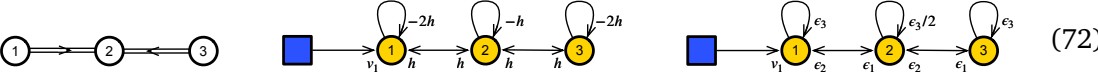

$$(72)$$

The leftmost figure shows the Dynkin diagram, and we can see from the quiver in the middle that there is only one independent parameter due to the superpotential

$$W = X_{12}X_{21}X_{11} - X_{21}X_{12}X_{22}^2 + X_{23}X_{32}X_{22}^2 - X_{32}X_{23}X_{33}. \qquad (73)$$

The $F$-term relations are then

$$X_{11}: \quad X_{12}X_{21} = 0, \tag{74a}$$

$$X_{12}: \quad X_{21}X_{11} = X_{22}^2 X_{21}, \tag{74b}$$

$$X_{21}: \quad X_{11}X_{12} = X_{12}X_{22}^2, \tag{74c}$$

$$X_{22}: \quad X_{22}X_{21}X_{12} + X_{21}X_{12}X_{22} = X_{22}X_{23}X_{32} + X_{23}X_{32}X_{22}, \tag{74d}$$

$$X_{23}: \quad X_{32}X_{22}^2 = X_{33}X_{32}, \tag{74e}$$

$$X_{32}: \quad X_{22}^2 X_{23} = X_{23}X_{33}, \tag{74f}$$

$$X_{33}: \quad X_{32}X_{23} = 0, \tag{74g}$$

where the column $X_{ab}$ on the left indicates the $F$-term relation $\partial W/\partial X_{ab} = 0$. To compute the partition function, we first uplift this with an extra parameter whose parametrization is given as the above rightmost figure,[15] and then take the limit $\epsilon_3 \to -\epsilon_1 - \epsilon_2$. The first few terms in $\mathcal{Z}$, along with the corresponding molecules, are

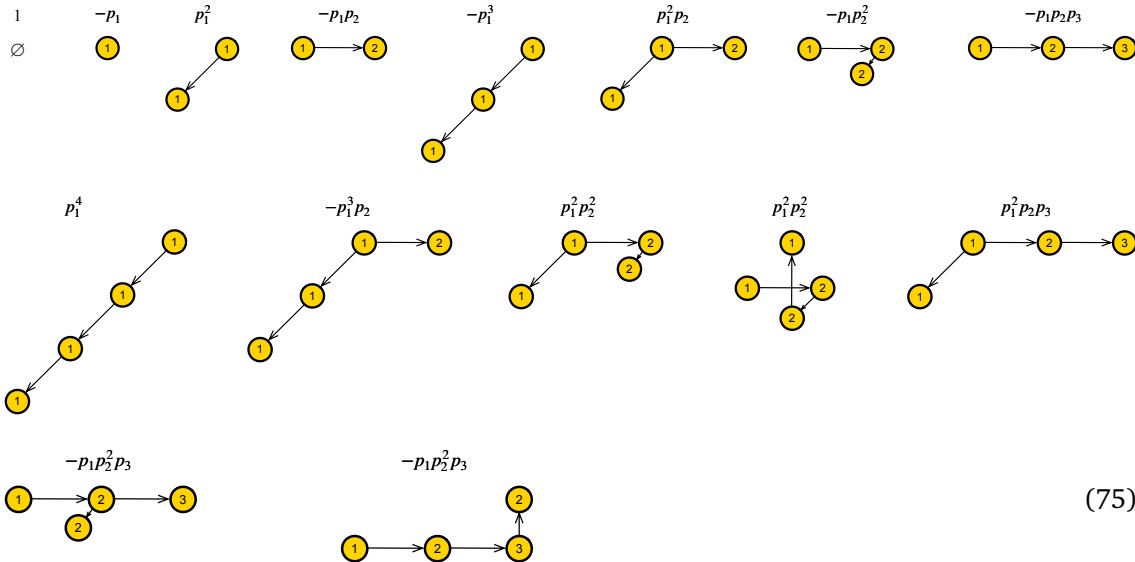

$$\tag{75}$$

As there is one single parameter, the crystal is two-dimensional:

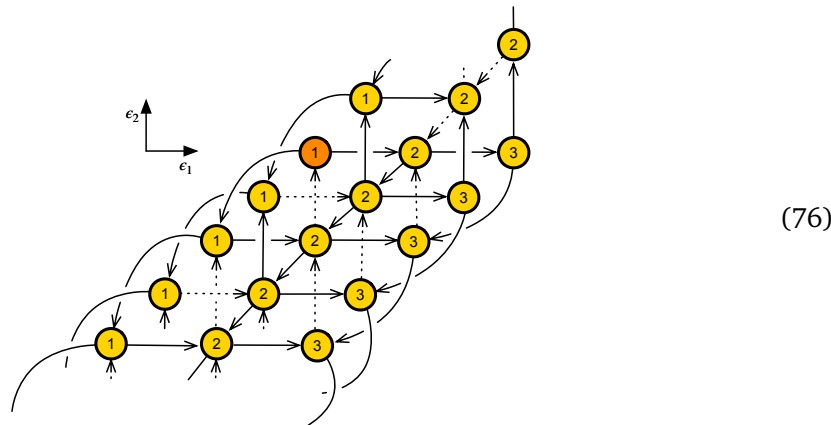

$$\tag{76}$$

where the initial atom is in orange. The dashed arrows correspond to arrows in the quiver but are not present as chemical bonds in the crystal. This is because they point to existing atoms in the crystal that do not have any corresponding poles in the one-loop determinants.

---

[15]When there is no uplift, the JK residue formula would contain only one parameter, say $\epsilon_1'$, such that $\epsilon_3' = -\epsilon_1'$. Therefore, in the uplift with an extra parameter, we would have $\epsilon_{1,2}$, and taking $\epsilon_3 \to -\epsilon_1 - \epsilon_2$ after evaluating the residues is the standard process for theories with $\mathcal{N} = 4$.

## 4.4 Affine $G_2$ theory

As another example of affine Dynkin type, let us consider the $G_2^{(1)}$ case [83]:

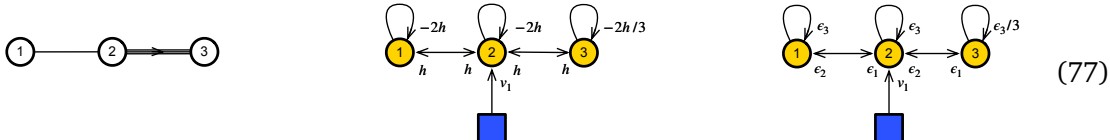

$$(77)$$

Here, we have taken the framing to be connected to node 2. The leftmost figure shows the Dynkin diagram, and we can see from the quiver in the middle that there is only one independent parameter due to the superpotential

$$W = X_{12}X_{21}X_{11} - X_{21}X_{12}X_{22} + X_{23}X_{32}X_{22} - X_{32}X_{23}X_{33}^3 \,. \tag{78}$$

The $F$-term relations are then

$$
\begin{align}
X_{11}: \quad & X_{12}X_{21} = 0 \,, \tag{79a} \\
X_{12}: \quad & X_{21}X_{11} = X_{22}X_{21} \,, \tag{79b} \\
X_{21}: \quad & X_{11}X_{12} = X_{12}X_{22} \,, \tag{79c} \\
X_{22}: \quad & X_{21}X_{12} = X_{23}X_{32} \,, \tag{79d} \\
X_{23}: \quad & X_{32}X_{22} = X_{33}^3 X_{32} \,, \tag{79e} \\
X_{32}: \quad & X_{22}X_{23} = X_{23}X_{33}^3 \,, \tag{79f} \\
X_{33}: \quad & X_{32}X_{23}X_{33}^2 + X_{33}X_{32}X_{23}X_{33} + X_{33}^2 X_{32}X_{23} = 0 \,. \tag{79g}
\end{align}
$$

To compute the partition function, we first uplift this with an extra parameter whose parametrization is given as the above rightmost figure, and then take the limit $\epsilon_3 \to -\epsilon_1 - \epsilon_2$. The first few terms in the index $\mathcal{Z}$, along with the corresponding molecules, are

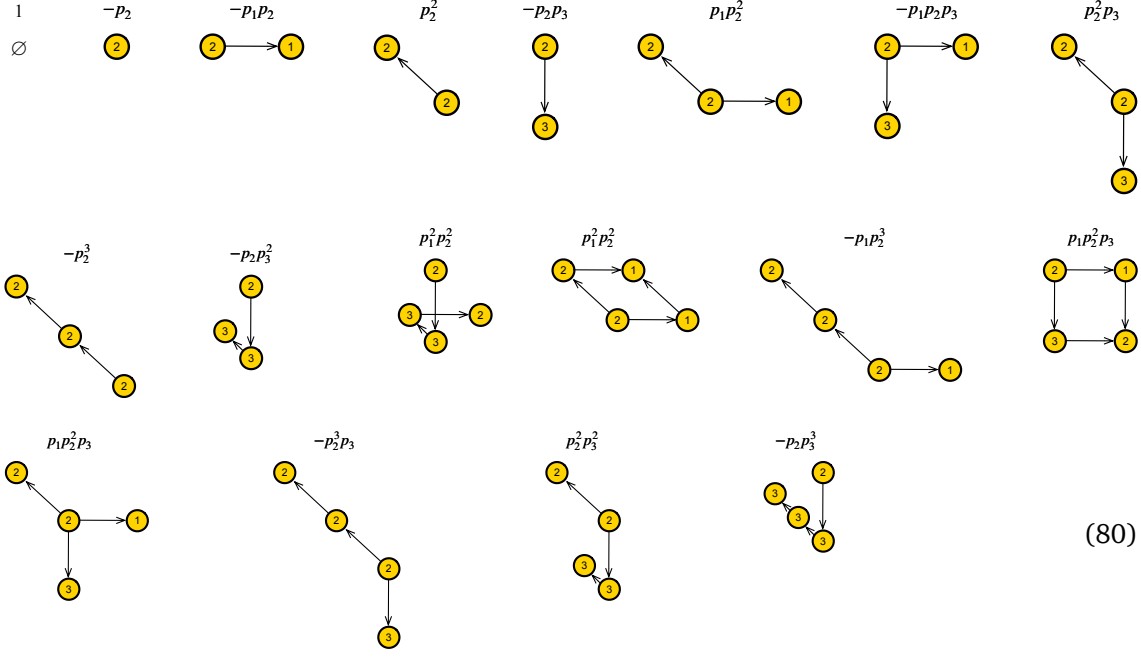

$$(80)$$

As there is one single parameter, the crystal is two-dimensional:

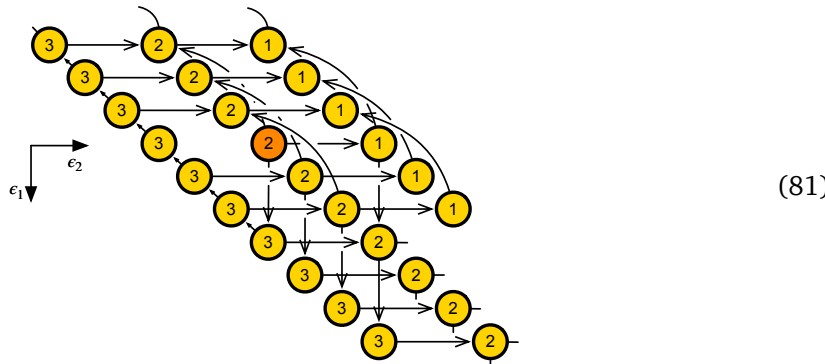

(81)

where the initial atom is in orange. We have omitted all the dashed arrows to avoid clutter of the arrows.

## 4.5 Affine $B(0,1)$ theory

As another example, let us consider the following super affine Dynkin diagram and the quiver [84]

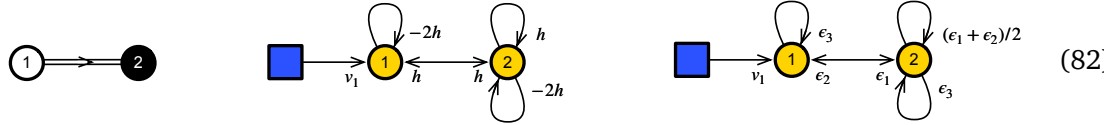

(82)

The black node in the Dynkin diagram indicates that it is a non-isotropic odd node. There is one independent parameter due to the superpotential

$$W = X_{12}X_{21}X_{11} - X_{21}X_{12}X_{22,2} + X_{22,1}^2 X_{22}.$$  (83)

The $F$-term relations are then

$$X_{11}: \quad X_{12}X_{21} = 0,$$  (84a)

$$X_{12}: \quad X_{21}X_{11} = X_{22,2}X_{21},$$  (84b)

$$X_{21}: \quad X_{11}X_{12} = X_{12}X_{22,2},$$  (84c)

$$X_{22,1}: \quad X_{22,1}X_{22,2} + X_{22,2}X_{21,2} = 0,$$  (84d)

$$X_{22,2}: \quad X_{21}X_{12} = X_{22,2}^2.$$  (84e)

To compute the partition function, we first uplift this with an extra parameter whose parametrization is given as the above rightmost figure, and then take the limit $\epsilon_3 \to -\epsilon_1 - \epsilon_2$. The first few terms in $\mathcal{Z}$, along with the corresponding molecules, are

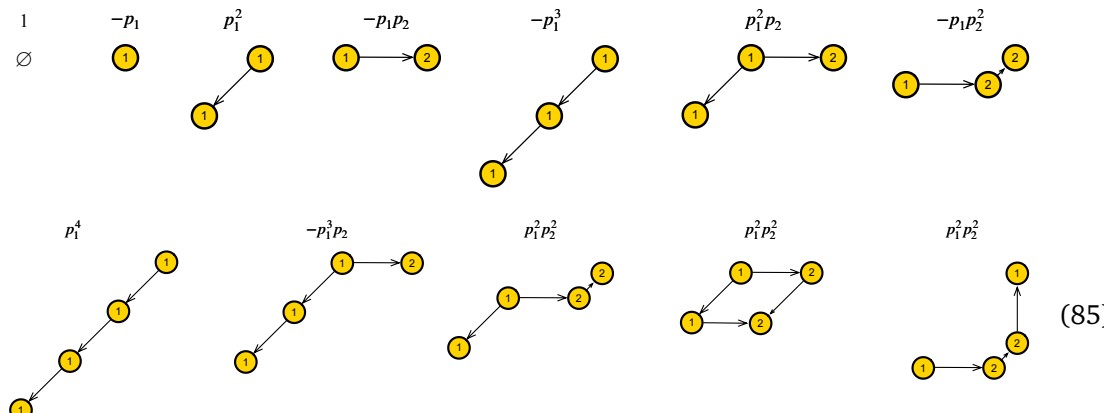

(85)

As there is one single parameter, the crystal is two-dimensional:

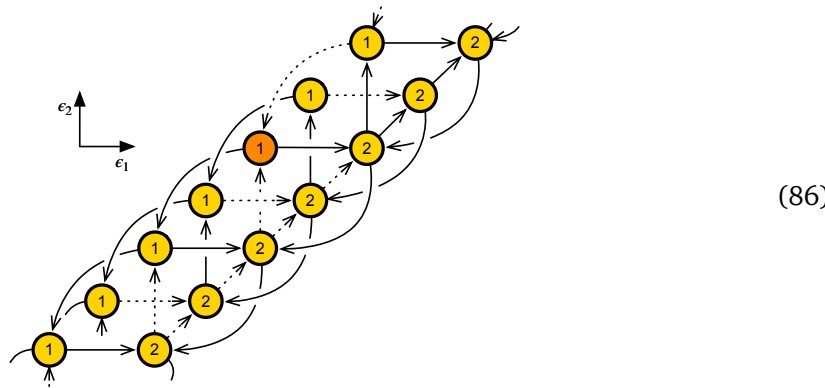

$$(86)$$

where the initial atom is in orange.

## 4.6 General $\eta$

In this subsection only, we shall consider general choices of the covector $\eta$. This would take us to different chambers, possibly non-cyclic. As a result, the minimal step of melting may not be one atom as discussed towards the end of §5.2.

Consider the conifold with the quiver

$$(87)$$

There are two independent parameters due to the superpotential

$$W = X_{12,1}X_{21,1}X_{12,2}X_{21,2} - X_{12,1}X_{21,2}X_{12,2}X_{21,1}. \tag{88}$$

The $F$-term relations are then

$$X_{12,1}: \quad X_{21,1}X_{12,2}X_{21,2} = X_{21,2}X_{12,2}X_{21,1}, \tag{89a}$$

$$X_{12,2}: \quad X_{21,2}X_{12,2}X_{21,1} = X_{21,1}X_{12,1}X_{21,2}, \tag{89b}$$

$$X_{21,1}: \quad X_{12,2}X_{21,2}X_{12,1} = X_{12,1}X_{21,2}X_{12,2}, \tag{89c}$$

$$X_{21,2}: \quad X_{12,1}X_{21,1}X_{12,2} = X_{12,2}X_{21,1}X_{12,1}. \tag{89d}$$

As an illustration, let us take $\eta = (-1/10, 1, 1, \ldots, 1)$. At level 1, since $\eta_1 = (-1/10)$ and the only hyperplane is $u_1 - v_1 = 0$, there are no admissible poles. Therefore, there is no single-atom crystal, and the melting would directly skip to configurations with two atoms. This illustrates the discussions in §3.3.

Moreover, at level 2, we find that an integer index is not guaranteed from the formula. With $\eta_2 = (-1/10, 1)$, there are two admissible cones:

$$\{(1,0),(-1,1)\} \quad \text{and} \quad \{(0,1),(-1,1)\}. \tag{90}$$

For the configuration with $p_1 p_2$, the contributions only come from the first cone, and the index is $-2$. For $p_1^2$, there are no contributions from the first cone as the pole $u_2^{(1)} - u_1^{(1)} + \epsilon_1 + \epsilon_2 + \epsilon_3 = 0$ from the vector multiplet is cancelled by the factor $u_2^{(1)} - v_1 + \epsilon_1 + \epsilon_2 + \epsilon_3 = 0$ in the numerator. However, for the second cone, there is a non-trivial contribution from the vector multiplet. In fact, the hyperplanes

$$u_2^{(1)} - v_1 = 0, \qquad u_2^{(1)} - u_1^{(1)} + \epsilon_1 + \epsilon_2 + \epsilon_3 = 0, \tag{91}$$

together with the factor $1/|\mathcal{W}| = 1/2$, would give $-1/2$.

Now, this twisted partition function obtained from the formula does not quite coincide with the Witten indices which should be integers.[16] In this paper, we shall not consider such issues. See also [85–87] for some related discussions.

## 4.7 Overlapping atoms: $D_4^{(1)}$ theory

Let us also look at an example where there are violations of the no-overlap condition. As a result, the partition functions cannot obtained using the method here although we expect the indices to be correct at low levels before the double poles appear.

The Dynkin diagram and the quiver associated to $D_4^{(1)}$ are [83]

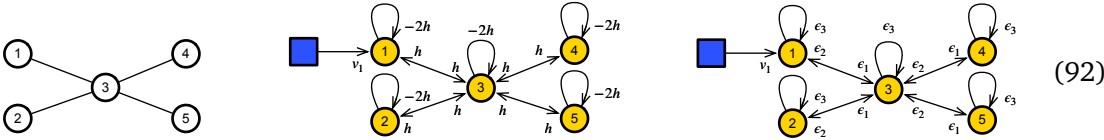

$$(92)$$

There is one independent parameter due to the superpotential

$$W = \sum_{\substack{a<b \\ a \text{ and } b \text{ connected}}} X_{ab}X_{ba}X_{aa} - X_{ba}X_{ab}X_{bb} \,. \qquad (93)$$

In fact, at low levels, one may check that the uplift in the above figure does recover the correct indices as given in [88, 89]. However, at level 5, the configuration

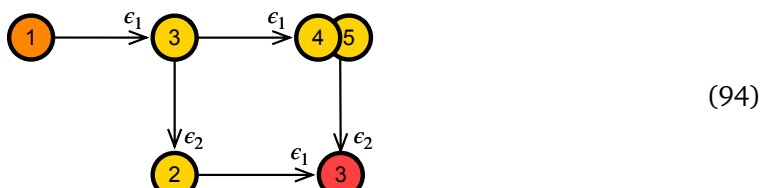

$$(94)$$

gives rise to a double pole $\left(u_2^{(3)} - v_1 - \epsilon_1 - \epsilon_2\right)^2$ for the red atom to be added. This is because each preceding atom (of colour 2, 4, or 5) carries one such pole while there is only one such factor in the numerator. In other words, any two of these three atoms would give a simple pole for the atom in red to be added to the crystal (and this also fits into the counting at level 4). However, when the three atoms are all in the crystal, we would have a double pole. One may also try some uplift with more parameters, but it would still lead to double poles at certain levels.

## 4.8 Theories with two supercharges

Now, we shall consider some examples with two supercharges. For these cases, the indices are non-trivial functions of the equivariant parameters/fugacities instead of just numbers.

---

[16]This does not mean that all the non-cyclic chambers for any theories would give fractional numbers from the formula. For instance, in the case with an arrow from the framing node (resp. node 1) to node 1 (resp. the framing node) of (generic) weight $v_1$ (resp. $v_2$) for the conifold quiver, the partition function reads $\mathcal{Z} = 1 + q_1 + 2q_1q_2 + 4q_1^2q_2 + q_1q_2^2 + \ldots$ from the JK residue computations. This is because the asymptotic flat directions have been lifted by some more involved framings.

**Example 1** Consider the quiver

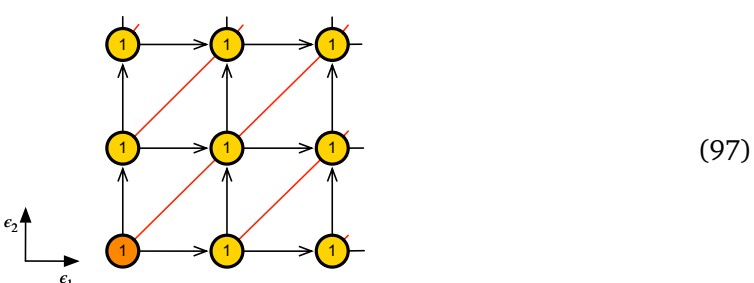

$$(95)$$

with the relation $\epsilon_3 = -\epsilon_1 - \epsilon_2$. This is in fact the dimensional reduction of (54) with the $J$-/$E$-interactions as $\Lambda \cdot X_1 X_2$, $\overline{\Lambda} \cdot X_2 X_1$. Let us list the first few terms of the partition function:

$$
\begin{aligned}
\mathcal{Z} = 1 &+ \frac{(\epsilon_1 + \epsilon_2)(v_1 - v_2)}{\epsilon_1 \epsilon_2} p \\
&+ \left( \frac{(\epsilon_1 + \epsilon_2)(2\epsilon_1 + \epsilon_2)(\epsilon_1 + v_1 - v_2)(v_1 - v_2)}{2\epsilon_1^2 \epsilon_2 (\epsilon_2 - \epsilon_1)} + \frac{(\epsilon_1 + \epsilon_2)(\epsilon_1 + 2\epsilon_2)(\epsilon_2 + v_1 - v_2)(v_1 - v_2)}{2\epsilon_1 \epsilon_2^2 (\epsilon_1 - \epsilon_2)} \right) p^2 \\
&+ \left( -\frac{(\epsilon_1 + \epsilon_2)(2\epsilon_1 + \epsilon_2)(\epsilon_1 + 2\epsilon_2)(\epsilon_1 + v_1 - v_2)(\epsilon_2 + v_1 - v_2)(v_1 - v_2)}{\epsilon_1^2 \epsilon_2^2 (\epsilon_1 - 2\epsilon_2)(2\epsilon_1 - \epsilon_2)} \right. \\
&+ \frac{(\epsilon_1 + \epsilon_2)(2\epsilon_1 + \epsilon_2)(3\epsilon_1 + \epsilon_2)(\epsilon_1 + v_1 - v_2)(2\epsilon_1 + v_1 - v_2)(v_1 - v_2)}{6\epsilon_1^3 \epsilon_2 (\epsilon_1 - \epsilon_2)(2\epsilon_1 - \epsilon_2)} \\
&+ \left. \frac{(\epsilon_1 + \epsilon_2)(\epsilon_1 + 2\epsilon_2)(\epsilon_1 + 3\epsilon_2)(\epsilon_2 + v_1 - v_2)(2\epsilon_2 + v_1 - v_2)(v_1 - v_2)}{6\epsilon_1 \epsilon_2^3 (\epsilon_1 - 2\epsilon_2)(\epsilon_1 - \epsilon_2)} \right) p^3 + \dots,
\end{aligned}
\tag{96}
$$

where we have used the rational version for simplicity. We have a two-dimensional crystal:

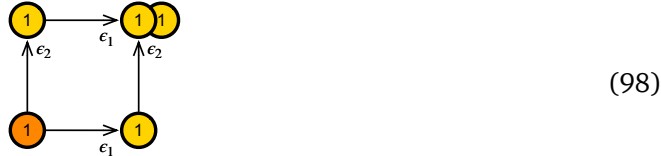

$$(97)$$

If we take the same quiver but with no relations so that $\epsilon_{1,2,3}$ are all independent, then there would be overlapping atoms:

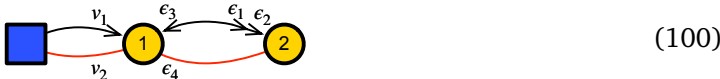

$$(98)$$

Indeed, for example, given the poles

$$ u_1 - v_1, \qquad u_2 - u_1 - \epsilon_1, \qquad u_3 - u_2 - \epsilon_2, \tag{99} $$

there would be a double pole $(u_4 - u_1 - \epsilon_2)^2$ for the corresponding atom to be added.

**Example 2** Consider the quiver

$$(100)$$

with the relation $\epsilon_4 = -\epsilon_1 - \epsilon_2 - \epsilon_3$. Let us list the first few terms of the partition function:

$$
\begin{aligned}
\mathcal{Z} = {} & 1 - (\nu_1 - \nu_2)p_1 + \left( \frac{(\epsilon_2 + \epsilon_3)(\nu_1 - \nu_2)}{(\epsilon_2 - \epsilon_1)(\epsilon_1 + \epsilon_3)} + \frac{(\epsilon_1 + \epsilon_3)(\nu_1 - \nu_2)}{(\epsilon_1 - \epsilon_2)(\epsilon_2 + \epsilon_3)} \right) p_1 p_2 - (\nu_1 - \nu_2)p_1 p_2^2 \\
& + \left( \frac{(\epsilon_1 + \epsilon_2 + 2\epsilon_3)((\epsilon_1 + \epsilon_3 + \nu_1 - \nu_2)(\nu_1 - \nu_2)}{(\epsilon_1 - \epsilon_2)(\epsilon_1 + \epsilon_3)} \right. \\
& \left. + \frac{(\epsilon_1 + \epsilon_2 + 2\epsilon_3)((\epsilon_2 + \epsilon_3 + \nu_1 - \nu_2)(\nu_1 - \nu_2)}{(\epsilon_2 - \epsilon_1)(\epsilon_2 + \epsilon_3)} \right) p_1^2 p_2 + \dots,
\end{aligned}
\tag{101}
$$

where we have used the rational version for simplicity. We have a three-dimensional crystal:

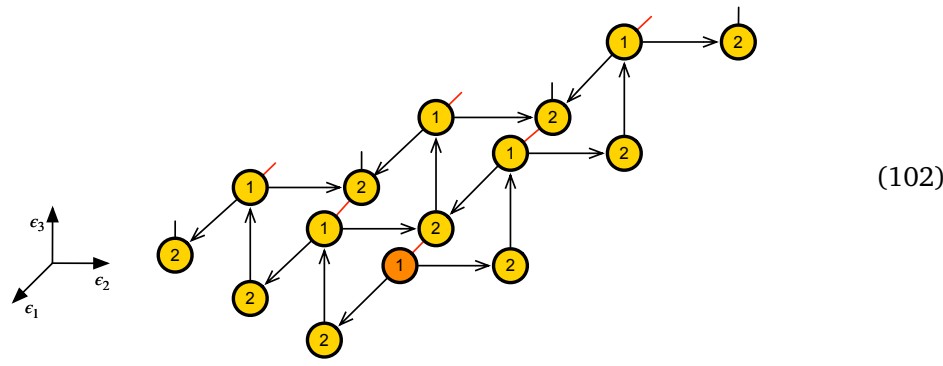

$$\tag{102}$$

# 5 Double quiver algebras for theories with four supercharges

In this section, we shall focus on the theories with four supercharges and construct their quiver algebras called the double quiver algebras. (The origin of the name "double" will be explained in §6.2.) The strategy is to find the operators that act on the crystal states[17] by adding or removing atoms with the help of the one-loop determinant. Therefore, this is restricted to the cases where the JK residue formula applies, especially for those satisfying the no-overlap condition in §3. We will mainly focus on the cyclic chambers, and will comment on more general chambers at the end of §5.2.

## 5.1 Defining relations

Let us first define the double quiver algebra $\widetilde{Y}$. Given a quiver with relations, the double quiver algebra $\widetilde{Y}$ has four sets of generating currents, $\widetilde{\psi}^{(a)}(z)$, $\widetilde{e}^{(a)}(z)$, $\widetilde{f}^{(a)}(z)$ and $\widetilde{\omega}^{(a)}(z)$, where $a \in Q_0$ labels the nodes in the quiver. The defining relations are

$$
\widetilde{\omega}^{(a)}(z)\widetilde{\omega}^{(b)}(w) \simeq \widetilde{\omega}^{(b)}(w)\widetilde{\omega}^{(a)}(z),
\tag{103a}
$$

$$
\widetilde{\psi}_+^{(a)}(z)\widetilde{\psi}_+^{(b)}(w) \simeq \widetilde{\psi}_+^{(b)}(w)\widetilde{\psi}_+^{(a)}(z),
\tag{103b}
$$

$$
\widetilde{\psi}_-^{(a)}(z)\widetilde{\psi}_-^{(b)}(w) \simeq \widetilde{\phi}^{a \Leftarrow b}(z - w + 2c)\widetilde{\phi}^{a \Leftarrow b}(z - w - 2c)^{-1}\widetilde{\psi}_-^{(b)}(w)\widetilde{\psi}_-^{(a)}(z),
\tag{103c}
$$

$$
\widetilde{\psi}_+^{(a)}(z)\widetilde{\psi}_-^{(b)}(w) \simeq \widetilde{\phi}^{a \Leftarrow b}(z - w + c)\widetilde{\phi}^{a \Leftarrow b}(z - w - c)^{-1}\widetilde{\psi}_-^{(b)}(w)\widetilde{\psi}_+^{(a)}(z),
\tag{103d}
$$

$$
\widetilde{\psi}_\pm^{(a)}(z)\widetilde{\omega}^{(b)}(w) \simeq \widetilde{\phi}^{a \Leftarrow b}(z - w + c \mp c/2)^{-1}\widetilde{\phi}^{a \Leftarrow b}(z - w - c \pm c/2)\widetilde{\omega}^{(b)}(w)\widetilde{\psi}_\pm^{(a)}(z),
\tag{103e}
$$

$$
\widetilde{d}^{(ba)}(z - w)\widetilde{\psi}_\pm^{(a)}(z)\widetilde{e}^{(b)}(w) \simeq \widetilde{\phi}^{a \Leftarrow b}(z - w + c \mp c/2)\widetilde{d}^{(ba)}\widetilde{e}^{(b)}(w)\widetilde{\psi}_\pm^{(a)}(z),
\tag{103f}
$$

---

[17]Recall that we would also refer to the molecule as the crystal state in our discussions. Therefore, when we have an operation that adds (resp. removes) an atom to (resp. from) the crystal state, we mean that we add (resp. remove) an atom to (resp. from) the molecule.

$$\widetilde{d}^{(ab)}(z-w)\widetilde{\psi}_{\pm}^{(a)}(z)\widetilde{f}^{(b)}(w) \simeq \widetilde{\phi}^{a\Leftarrow b}(z-w-c\pm c/2)^{-1}\widetilde{d}^{(ab)}(z-w)\widetilde{f}^{(b)}(w)\widetilde{\psi}_{\pm}^{(a)}(z), \quad (103g)$$

$$\delta(z-w)\widetilde{\phi}^{d\Leftarrow a}(u-z-c)\widetilde{e}^{(d)}(u)\widetilde{\omega}^{(a)}(z) + \delta(u-w)\widetilde{\phi}^{a\Leftarrow d}(z-w-c)\widetilde{e}^{(a)}(z)\widetilde{\omega}^{(d)}(w)$$
$$\simeq \delta(z-w)\widetilde{\omega}^{(a)}(z)\widetilde{e}^{(d)}(u) + \delta(u-w)\widetilde{\omega}^{(d)}(u)\widetilde{e}^{(a)}(z), \quad (103h)$$

$$\delta(z-w)\widetilde{\phi}^{d\Leftarrow a}(u-z-c)^{-1}\widetilde{f}^{(d)}(u)\widetilde{\omega}^{(a)}(z) + \delta(u-w)\widetilde{\phi}^{a\Leftarrow d}(z-w-c)^{-1}\widetilde{f}^{(a)}(z)\widetilde{\omega}^{(d)}(w)$$
$$\simeq \delta(z-w)\widetilde{\omega}^{(a)}(z)\widetilde{f}^{(d)}(u) + \delta(u-w)\widetilde{\omega}^{(d)}(u)\widetilde{f}^{(a)}(z), \quad (103i)$$

$$\widetilde{d}^{(ba)}(z-w)\widetilde{e}^{(a)}(z)\widetilde{e}^{(b)}(w) \simeq (-1)^{|a||b|}\widetilde{d}^{(ba)}(z-w)\widetilde{e}^{(b)}(w)\widetilde{e}^{(a)}(z), \quad (103j)$$

$$\widetilde{d}^{(ab)}(z-w)\widetilde{f}^{(a)}(z)\widetilde{f}^{(b)}(w) \simeq (-1)^{|a||b|}\widetilde{d}^{(ab)}(z-w)\widetilde{f}^{(b)}(w)\widetilde{f}^{(a)}(z), \quad (103k)$$

$$\widetilde{\phi}^{a\Leftarrow b}(z-w-c)\widetilde{e}^{(a)}(z)\widetilde{f}^{(b)}(w) - (-1)^{|a||b|}\widetilde{f}^{(b)}(w)\widetilde{e}^{(a)}(z)$$
$$\simeq \delta_{ab}\Big(\delta(z-w-c)\widetilde{\psi}_{+}^{(a)}(w+c/2) - \delta(z-w+c)\widetilde{\psi}_{-}^{(a)}(z+c/2) - \delta(z-w)\widetilde{\omega}^{(a)}(z)\Big). \quad (103l)$$

The relations require some explanations:

- The key factor in the above relations is the bond factor[18]

$$\widetilde{\phi}^{a\Leftarrow b}(z) := \begin{cases} \dfrac{\zeta(z)\zeta(-z)}{\zeta(z+\epsilon)\zeta(-z+\epsilon)}\left(\displaystyle\prod_{I\in\{a\to a\}}\dfrac{-\zeta(\epsilon-\epsilon_I)}{\zeta(-\epsilon_I)}\dfrac{\zeta(z+\epsilon-\epsilon_I)\zeta(z-\epsilon+\epsilon_I)}{\zeta(z-\epsilon_I)\zeta(z+\epsilon_I)}\right), & b=a, \\[3ex] \left(\displaystyle\prod_{I\in\{a\to b\}}\dfrac{-\zeta(z-\epsilon+\epsilon_I)}{\zeta(z+\epsilon_I)}\right)\left(\displaystyle\prod_{I\in\{b\to a\}}\dfrac{-\zeta(z+\epsilon-\epsilon_I)}{\zeta(z-\epsilon_I)}\right), & b\neq a. \end{cases}$$
$$(104)$$

  It satisfies

$$\widetilde{\phi}^{b\Leftarrow a}(-z) = \widetilde{\phi}^{a\Leftarrow b}(z). \quad (105)$$

  In the some of the relations, the factor $\widetilde{d}^{(ab)}(z)$ is basically the denominator of the bond factor:

$$\widetilde{d}^{(ab)}(z) := \left(\prod_{I\in\{a\to b\}}(z+\epsilon_I)\right)\left(\prod_{I\in\{b\to a\}}(-z+\epsilon_I)\right). \quad (106)$$

  Notice that

$$\widetilde{d}^{(ab)}(z) = \widetilde{d}^{(ba)}(-z). \quad (107)$$

  In the current relations, the factor $\widetilde{d}^{(ab)}(z)$ seems to be redundant. However, this would be different if we consider the relations of the actual generators, namely the modes of the currents, as we shall discuss shortly. Moreover, this is important for the crystals to be the representations of the algebras.

- We have a hierarchy of rational, trigonometric, elliptic double quiver algebras, depending on the choice of $\zeta$ as in (11). The rational algebra can also be called the double quiver Yangian, and the trigonometric algebras can also be called a double quiver quantum toroidal algebra, following the terminologies of the quiver Yangians [43, 44].

- We have used the formal $\delta$-function defined by

$$\delta(z) := \begin{cases} 1/z, & \text{rational}, \\ \displaystyle\sum_{k\in\mathbb{Z}} Z^k, & \text{trigonometric or elliptic}, \end{cases} \quad (108)$$

  where in the following we use the convention that variables written in the upper case are the exponentiated versions of variables written in the lower case:

$$Z = \mathrm{e}^{2\pi i z}, \quad W = \mathrm{e}^{2\pi i w}, \quad C = \mathrm{e}^{2\pi i c}, \quad \text{etc.} \quad (109)$$

---

[18]Clearly, it is also possible to remove the factors $-\zeta(\epsilon-\epsilon_I)/\zeta(-\epsilon_I)$ from the adjoint chirals when defining $\widetilde{\Psi}_{\mathscr{C},\pm}^{(a)}(z)$ (although we are keeping such factors here).

- In the rational case, "$\simeq$" means that the left and right hand sides are equal in their Taylor expansions around $z = \infty$. In other words, they are equivalent up to some $z^m w^n$ terms. For the trigonometric and elliptic algebras, the equality is in the sense of the Laurent series in $Z$ and $W$. If the relation is $L \simeq \left(P^{-1}Q\right)R$, then the Laurent expansions of $PL$ and $QR$ would agree.

- Similar to the quiver Yangians in the literature, there is a $\mathbb{Z}_2$-grading such that

$$\left|\widetilde{e}^{(a)}\right| = \left|\widetilde{f}^{(a)}\right| = |a|, \tag{110}$$

$$\left|\widetilde{\psi}_{\pm}^{(a)}\right| = \left|\widetilde{\omega}^{(a)}\right| = 0, \tag{111}$$

where

$$|a| \equiv |a \to a| + 1 \pmod 2. \tag{112}$$

We say that an operator $X^{(a)}$ is bosonic (resp. fermionic) if $\left|X^{(a)}\right| = 0$ (resp. $\left|X^{(a)}\right| = 1$). In the rational case, we simply have $\widetilde{\psi}^{(a)}(z) := \widetilde{\psi}_+^{(a)}(z) = \widetilde{\psi}_-^{(a)}(z)$.

- We have also introduced an extra central element $c$. In the rational case, we have $c = 0$. However, it can be non-trivial for the trigonometric and elliptic cases.[19] Notice that the $\widetilde{\psi}_{\pm}$ and $\widetilde{\omega}$ currents commute among themselves when $c = 0$ (or equivalently, $C = 1$). Only when $c = 0$, the algebras admit the molecules/crystal states as their representations which will be discussed below.

We notice that the $\widetilde{e}\widetilde{\omega}$ relation (and likewise for the $\widetilde{f}\widetilde{\omega}$ relation) is not independent of the other relations. It can be derived from the other relations as follows. Starting from the $\widetilde{e}\widetilde{f}$ relation:

$$\widetilde{\phi}^{a \Leftarrow b}(z - w - c)\widetilde{e}^{(a)}(z)\widetilde{f}^{(b)}(w) - (-1)^{|a||b|}\widetilde{f}^{(b)}(w)\widetilde{e}^{(a)}(z) \tag{113}$$
$$\simeq \delta_{ab}\left(\delta(z - w - c)\widetilde{\psi}_+^{(a)}(w + c/2) - \delta(z - w + c)\widetilde{\psi}_-^{(a)}(z + c/2) - \delta(z - w)\widetilde{\omega}^{(a)}(z)\right),$$

multiply both sides by $\widetilde{e}^{(d)}(u)$:

$$\left(\widetilde{\phi}^{a \Leftarrow b}(z - w - c)\widetilde{e}^{(a)}(z)\widetilde{f}^{(b)}(w) - (-1)^{|a||b|}\widetilde{f}^{(b)}(w)\widetilde{e}^{(a)}(z)\right)\widetilde{e}^{(d)}(u) \tag{114}$$
$$\simeq \delta_{ab}\left(\delta(z - w - c)\widetilde{\psi}_+^{(a)}(w + c/2) - \delta(z - w + c)\widetilde{\psi}_-^{(a)}(z + c/2) - \delta(z - w)\widetilde{\omega}^{(a)}(z)\right)\widetilde{e}^{(d)}(u).$$

Apply the $\widetilde{e}\widetilde{e}$ and $\widetilde{e}\widetilde{f}$ relations on the left hand side to move $\widetilde{e}^{(d)}(u)$ to the leftmost. We get

$$\text{(LHS)} \simeq \left(\widetilde{\phi}^{d \Leftarrow a}(u - w - c)\widetilde{e}^{(d)}(u)\left(\delta(z - w - c)\widetilde{\psi}_+^{(a)}(w + c/2)\right.\right.$$
$$\left.- \delta(z - w + c)\widetilde{\psi}_-^{(a)}(w - c/2) - \delta(z - w)\widetilde{\omega}^{(a)}(z)\right)$$
$$+ \widetilde{\phi}^{a \Leftarrow d}(z - w - c)\widetilde{e}^{(a)}(z)\left(\delta(u - w - c)\widetilde{\psi}_+^{(d)}(w + c/2)\right.$$
$$\left.- \delta(u - w + c)\widetilde{\psi}_-^{(d)}(w - c/2) - \delta(u - w)\widetilde{\omega}^{(d)}(u)\right)$$
$$- \left(\delta(u - w - c)\widetilde{\psi}_+^{(d)}(w + c/2) - \delta(u - w + c)\widetilde{\psi}_-^{(d)}(w - c/2)\right.$$
$$\left.\left.- \delta(u - w)\widetilde{\omega}^{(d)}(u)\right)\widetilde{e}^{(a)}(z)\right)\widetilde{d}^{da}(z - u). \tag{115}$$

---

[19]Here, we have introduced the central element in a way that $\widetilde{\psi}_+$ would always commute with themselves. It could also be possible to include the central element that makes the relations more symmetric in $\widetilde{\psi}_{\pm}$ (which is also given in Appendix A).

This is equal to the right hand side of (115). Moreover, the factors contain $\widetilde{\psi}$ get cancelled due to the $\widetilde{\psi}\widetilde{e}$ relation.[20] In other words, we have the $\widetilde{e}\widetilde{\omega}$ relation:

$$\delta(z-w)\widetilde{\phi}^{d\Leftarrow a}(u-z-c)\widetilde{e}^{(d)}(u)\widetilde{\omega}^{(a)}(z) + \delta(u-w)\widetilde{\phi}^{a\Leftarrow d}(z-w-c)\widetilde{e}^{(a)}(z)\widetilde{\omega}^{(d)}(w)$$
$$\simeq \delta(z-w)\widetilde{\omega}^{(a)}(z)\widetilde{e}^{(d)}(u) + \delta(u-w)\widetilde{\omega}^{(d)}(u)\widetilde{e}^{(a)}(z). \tag{116}$$

The mode generators can be obtained from the expansions of the currents:

$$\widetilde{\psi}_{\pm}^{(a)}(z) = \begin{cases} \sum_{n\in\mathbb{Z}_{\geq 0}} \widetilde{\psi}_n^{(a)} z^{-n}, & \text{rational,} \\ \sum_{n\in\mathbb{Z}_{\geq 0}} \widetilde{\psi}_{\pm,n}^{(a)} Z^{\mp n}, & \text{trigonometric,} \\ \sum_{n\in\mathbb{Z}_{\geq 0}} \widetilde{\psi}_{\pm,n,0}^{(a)} Z^{\mp n} + \sum_{n\in\mathbb{Z}} \sum_{\alpha\in\mathbb{Z}_{>0}} \widetilde{\psi}_{\pm,n,\alpha}^{(a)} Z^{\mp n}\mathfrak{q}^{\alpha}, & \text{elliptic,} \end{cases} \tag{117a}$$

$$\widetilde{e}^{(a)}(z) = \begin{cases} \sum_{n\in\mathbb{Z}_{>0}} \widetilde{e}_n^{(a)} z^{-n}, & \text{rational,} \\ \sum_{n\in\mathbb{Z}} \widetilde{e}_n^{(a)} Z^{-n}, & \text{trigonometric,} \\ \sum_{n\in\mathbb{Z}} \sum_{\alpha\in\mathbb{Z}_{\geq 0}} \widetilde{e}_{n,\alpha}^{(a)} Z^{-n}\mathfrak{q}^{\alpha}, & \text{elliptic,} \end{cases} \tag{117b}$$

$$\widetilde{f}^{(a)}(z) = \begin{cases} \sum_{n\in\mathbb{Z}_{>0}} \widetilde{f}_n^{(a)} z^{-n}, & \text{rational,} \\ \sum_{n\in\mathbb{Z}} \widetilde{f}_n^{(a)} Z^{-n}, & \text{trigonometric,} \\ \sum_{n\in\mathbb{Z}} \sum_{\alpha\in\mathbb{Z}_{\geq 0}} \widetilde{f}_{n,\alpha}^{(a)} Z^{-n}\mathfrak{q}^{\alpha}, & \text{elliptic.} \end{cases} \tag{117c}$$

Notice that the constraint on the range of $n$ in the mode expansions of $\widetilde{\psi}_{\pm}$ comes from the fact that $\widetilde{\Psi}^{(a)}(z)$ defined below in (124) are homogeneous. Due to the same reason, there are no shift factors in these mode expansions, unlike the cases for quiver Yangians in [42,43] and double quiver algebras for theories with two supercharges to be discussed in §7.

From the mode expansions, it is clear that the factor $\widetilde{d}^{(ab)}(z)$ is not redundant. Let us illustrate this with a simple example in the rational case. Suppose that there is one single arrow of weight $\epsilon_I$ from node $(b)$ to node $(a)$. If there were no such factors, the modes $\widetilde{e}_m^{(a)}$ and $\widetilde{e}_n^{(b)}$ would simply commute. However, the actual relation of the generators reads

$$\widetilde{e}_{m+1}^{(a)}\widetilde{e}_n^{(b)} - \widetilde{e}_m^{(a)}\widetilde{e}_{n+1}^{(b)} + \epsilon_I \widetilde{e}_m^{(a)}\widetilde{e}_n^{(b)} = \widetilde{e}_n^{(b)}\widetilde{e}_{m+1}^{(a)} - \widetilde{e}_{n+1}^{(b)}\widetilde{e}_m^{(a)} + \epsilon_I \widetilde{e}_n^{(b)}\widetilde{e}_m^{(a)}. \tag{118}$$

## 5.2 Crystal representations

When $c = 0$, we have the molecules (i.e. crystal states) as representations of the double quiver algebra $\widetilde{\mathsf{Y}}$. This representation is closely related to the JK residue formula for the partition functions. Roughly speaking, the $\widetilde{e}$ (resp. $\widetilde{f}$) currents are creation (resp. annihilation) operators that add atoms to (resp. remove atoms from) the molecules/crystal states. Moreover, the crystal states are eigenstates of the $\widetilde{\psi}$ currents. The $\widetilde{\omega}$ currents collect all the inadmissible poles.[21] To write down the actions of the currents on the crystal states, it would be convenient to introduce some useful functions as follows.

---

[20]In particular, the factors $\widetilde{d}^{(ab)}(z-w)$ and $\widetilde{d}^{(ba)}(z-w)$ in the $\widetilde{\psi}\widetilde{e}$ and $\widetilde{\psi}\widetilde{f}$ relations are due to the consistency of the relations so that we can derive the $\widetilde{e}\widetilde{\omega}$ and $\widetilde{f}\widetilde{\omega}$ relations. For the $\widetilde{e}\widetilde{e}$ and $\widetilde{f}\widetilde{f}$ relations, such factors are necessary for the crystal representations as we shall discuss in the next subsection.

[21]In general, we would expect that the $\widetilde{\omega}$ currents would depend on the choice of $\eta$. Here, we shall only consider the algebras for $\eta = (1, 1, \ldots, 1)$. We shall briefly comment on the general case at the end of this subsection.

Suppose that the partition function at level $N$ is given by the poles $u_1^*, \ldots, u_N^*$. According to the constructive definition of the JK residues as reviewed above, we have

$$Z_{\text{1-loop}}(u_1, \ldots, u_{N+1}) = Z_{\text{1-loop}}(u_1, \ldots, u_N) \Delta Z(u_1, \ldots, u_{N+1}). \tag{119}$$

In other words, the information of the next level is all contained in $\Delta Z(u_1^*, \ldots, u_N^*, u_{N+1})$. Now, given a crystal state $|\mathscr{C}\rangle$ with $N$ atoms, to implement the raising/lowering to $N \pm 1$, let us introduce the charge functions

$$\widetilde{\Psi}_{\mathscr{C}}^{(a)}(z) := \Delta \widetilde{Z}\left(u_1^*, \ldots, u_N^*, u_{N+1} = u_{N_a+1}^{(a)} = z\right), \tag{120}$$

where the superscript $(a)$ indicates that the atom to be added/removed is of colour $a$. Here, $\Delta \widetilde{Z}$ means that we do not include the factor $\xi_{\mathcal{N}=4}$. It would be convenient to introduce the functions indicating the contributions from the vector multiplets:

$$\widetilde{\Psi}_{V,\mathscr{C}}^{(a)}(z) := \Delta \widetilde{Z}_V = \prod_{\mathfrak{a} \in \mathscr{C}} \frac{\zeta(z - \epsilon_{\mathfrak{a}})\zeta(\epsilon_{\mathfrak{a}} - z)}{\zeta(z - \epsilon_{\mathfrak{a}} + \epsilon)\zeta(\epsilon_{\mathfrak{a}} - z + \epsilon)}, \tag{121}$$

where $\epsilon_{\mathfrak{a}}$ denotes the weight of the atom $\mathfrak{a}$, and $\Delta \widetilde{Z}_V$ again indicates the removal of the factor $\xi_{\mathcal{N}=4}$. Likewise, for the contributions from the chiral multiplets, we have

$$\widetilde{\Psi}_{\chi,\mathscr{C}}^{(a)}(z) := \Delta Z_\chi = {}^\#\widetilde{\psi}^{(a)}(z)\left(\prod_{\mathfrak{a} \in \mathscr{C}} \prod_{I \in \{a \to a\}} \frac{-\zeta(\epsilon - \epsilon_I)}{\zeta(-\epsilon_I)} \frac{\zeta(z - \epsilon_{\mathfrak{a}} + \epsilon - \epsilon_I)\zeta(\epsilon_{\mathfrak{a}} - z + \epsilon - \epsilon_I)}{\zeta(z - \epsilon_{\mathfrak{a}} - \epsilon_I)\zeta(\epsilon_{\mathfrak{a}} - z - \epsilon_I)}\right) \tag{122}$$
$$\times \left(\prod_{b \neq a} \prod_{\mathfrak{b} \in \mathscr{C}} \left(\prod_{I \in \{a \to b\}} \frac{-\zeta(\epsilon_{\mathfrak{b}} - z + \epsilon - \epsilon_I)}{\zeta(\epsilon_{\mathfrak{b}} - z - \epsilon_I)}\right)\left(\prod_{I \in \{b \to a\}} \frac{-\zeta(z - \epsilon_{\mathfrak{b}} + \epsilon - \epsilon_I)}{\zeta(z - \epsilon_{\mathfrak{b}} - \epsilon_I)}\right)\right).$$

We have included the factor from the framing (whose node is denoted as $\infty$) explicitly:

$$^\#\widetilde{\psi}^{(a)}(z) := \left(\prod_{I \in \{a \to \infty\}} \frac{-\zeta(-z + \epsilon - \nu_I)}{\zeta(-z - \nu_I)}\right)\left(\prod_{I \in \{\infty \to a\}} \frac{-\zeta(z + \epsilon - \nu_I)}{\zeta(z - \nu_I)}\right), \tag{123}$$

where the weights of the arrows are $\nu_I$. As a result,

$$\widetilde{\Psi}_{\mathscr{C}}^{(a)}(z) = \widetilde{\Psi}_{V,\mathscr{C}}^{(a)}(z)\widetilde{\Psi}_{\chi,\mathscr{C}}^{(a)}(z). \tag{124}$$

Since this comes from the JK residue formula in the index computation, it is guaranteed that the poles are always simple poles for the quivers satisfying our conditions. Moreover, in the resulting algebra, we should always keep the refinement ($\epsilon \neq 0$) so that $\widetilde{\Psi}_{\mathscr{C}}^{(a)}(z)$ would remain non-trivial to keep the important information of poles.[22]

The actions of the generating currents on a crystal state $\mathscr{C}$ then read

$$\widetilde{\psi}_{\pm}^{(a)}(z)|\mathscr{C}\rangle = \begin{cases} \widetilde{\Psi}_{\mathscr{C}}^{(a)}(z)|\mathscr{C}\rangle, & \text{rational}, \\ \left[\widetilde{\Psi}_{\mathscr{C}}^{(a)}(Z)\right]_\pm |\mathscr{C}\rangle, & \text{trigonometric/elliptic}, \end{cases} \tag{125a}$$

$$\widetilde{\omega}^{(a)}(z)|\mathscr{C}\rangle = \sum_{\text{Inad}\left(\Psi_{\mathscr{C}}^{(a)}(z),\eta\right)} \delta(z - \epsilon_{\mathfrak{a}}) \lim_{x = \epsilon_{\mathfrak{a}}} \zeta(x - \epsilon_{\mathfrak{a}})\widetilde{\Psi}_{\mathscr{C}}^{(a)}(x)|\mathscr{C}\rangle, \tag{125b}$$

---

[22]Notice that however, this does not contradict the fact that the partition functions can be unrefined. This is because $\epsilon \to 0$ can be taken after the evaluation of the JK residues.

$$\widetilde{e}^{(a)}(z)|\mathscr{C}\rangle = \sum_{\mathfrak{a}\in\mathrm{Add}(\mathscr{C})} \pm\delta(z-\epsilon_\mathfrak{a})\left(\pm\lim_{x=\epsilon_\mathfrak{a}}\zeta(x-\epsilon_\mathfrak{a})\widetilde{\Psi}^{(a)}_{\mathscr{C}}(x)\right)^{1/2}|\mathscr{C}+\mathfrak{a}\rangle,\qquad(125c)$$

$$\widetilde{f}^{(a)}(z)|\mathscr{C}\rangle = \sum_{\mathfrak{a}\in\mathrm{Rem}(\mathscr{C})} \pm\delta(z-\epsilon_\mathfrak{a})\left(\pm\lim_{x=\epsilon_\mathfrak{a}}\zeta(x-\epsilon_\mathfrak{a})\widetilde{\Psi}^{(a)}_{\mathscr{C}-\mathfrak{a}}(x)\right)^{1/2}|\mathscr{C}-\mathfrak{a}\rangle,\qquad(125d)$$

where $\mathrm{Add}(\mathscr{C})$ (resp. $\mathrm{Rem}(\mathscr{C})$) denotes the set of addable (resp. removable) atoms of $\mathscr{C}$ and $\mathrm{Inad}\left(\Psi^{(a)}_{\mathscr{C}}(z),\eta\right)$ denotes the set of inadmissible poles of $\Psi^{(a)}_{\mathscr{C}}(z)$ determined by $\eta$. In the trigonometric and elliptic cases, the notation $[F(X)]_+$ (resp. $[F(X)]_-$) means the expansion of $F(X)$ around $X=\infty$ (resp. $X=0$). The function $\zeta(z)$ is defined in (11). There are some signs in the actions of the $\widetilde{e}$ and $\widetilde{f}$ currents which depend on different cases.

Let us verify that the actions satisfy the relations of the currents. The $\widetilde{\psi}\widetilde{\psi}$, $\widetilde{\omega}\widetilde{\omega}$ and $\widetilde{\psi}\widetilde{\omega}$ relations are straightforward since the crystal states are eigenstates of them. There is no need to check the $\widetilde{e}\widetilde{\omega}$ and $\widetilde{f}\widetilde{\omega}$ relations since they are derived from the other relations. Let us now check the $\widetilde{f}\widetilde{f}$ relation.

Consider the operator $\widetilde{f}^{(a)}(z)$ acting on the crystal state by

$$\widetilde{f}^{(a)}(z)|\mathscr{C}\rangle = \sum_{\mathfrak{a}\in\mathrm{Rem}(\mathscr{C})} \delta(z-\epsilon_\mathfrak{a})\widetilde{F}[\mathscr{C}\to\mathscr{C}-\mathfrak{a}]|\mathscr{C}-\mathfrak{a}\rangle,\qquad(126)$$

where

$$\widetilde{F}[\mathscr{C}\to\mathscr{C}-\mathfrak{a}] = \varsigma(\mathscr{C}\to\mathscr{C}-\mathfrak{a})\left(\varrho_{\mathscr{C}\to\mathscr{C}-\mathfrak{a}}\lim_{x=\epsilon_\mathfrak{a}}\zeta(x-\epsilon_\mathfrak{a})\widetilde{\Psi}^{(a)}_{\mathscr{C}-\mathfrak{a}}(x)\right)^{1/2},\qquad(127)$$

for some signs $\varsigma$ and $\varrho$. For $\widetilde{f}^{(a)}(z)\widetilde{f}^{(b)}(w)|\mathscr{C}\rangle$, we have

$$\widetilde{F}[\mathscr{C}-\mathfrak{b}\to\mathscr{C}-\mathfrak{b}-\mathfrak{a}]\widetilde{F}[\mathscr{C}\to\mathscr{C}-\mathfrak{b}] = \pm\left(\pm\lim_{x=\epsilon_\mathfrak{a}}\zeta(x-\epsilon_\mathfrak{a})\widetilde{\Psi}^{(a)}_{\mathscr{C}-\mathfrak{b}-\mathfrak{a}}(x)\lim_{x=\epsilon_\mathfrak{b}}\zeta(x-\epsilon_\mathfrak{b})\widetilde{\Psi}^{(b)}_{\mathscr{C}-\mathfrak{b}}(x)\right)^{1/2}.$$
$$(128)$$

In particular, $\widetilde{\Psi}_{\mathscr{C}-\mathfrak{b}}(x)$ and $\widetilde{\Psi}_{\mathscr{C}-\mathfrak{b}-\mathfrak{a}}(x)$ follow from the definition in (124). Likewise, for $\widetilde{f}^{(b)}(w)\widetilde{f}^{(a)}(z)|\mathscr{C}\rangle$, we have

$$\widetilde{F}[\mathscr{C}-\mathfrak{a}\to\mathscr{C}-\mathfrak{a}-\mathfrak{b}]\widetilde{F}[\mathscr{C}\to\mathscr{C}-\mathfrak{b}] = \pm\left(\pm\lim_{x=\epsilon_\mathfrak{b}}\zeta(x-\epsilon_\mathfrak{b})\widetilde{\Psi}^{(b)}_{\mathscr{C}-\mathfrak{a}-\mathfrak{b}}(x)\lim_{x=\epsilon_\mathfrak{a}}\zeta(x-\epsilon_\mathfrak{a})\widetilde{\Psi}^{(a)}_{\mathscr{C}-\mathfrak{a}}(x)\right)^{1/2}.$$
$$(129)$$

Suppose that all the configurations appeared above are valid molecules. In other words, all the $\widetilde{\Psi}$ factors are non-vanishing. Then the two expressions would coincide given that

$$\frac{\varsigma(\mathscr{C}-\mathfrak{b}\to\mathscr{C}-\mathfrak{a}-\mathfrak{b})\varsigma(\mathscr{C}\to\mathscr{C}-\mathfrak{b})}{\varsigma(\mathscr{C}-\mathfrak{a}\to\mathscr{C}-\mathfrak{a}-\mathfrak{b})\varsigma(\mathscr{C}\to\mathscr{C}-\mathfrak{a})} = (-1)^{|a||b|},\qquad(130)$$

and the "cross ratio" of $\varrho$ equals 1. If the removed atoms are at non-generic positions, it could be possible that the atom $\mathfrak{a}$ is connected to the atom $\mathfrak{b}$ so that $\mathscr{C}-\mathfrak{b}-\mathfrak{a}$ is a state while $\mathscr{C}-\mathfrak{a}-\mathfrak{b}$ is not a valid configuration. Then the corresponding coefficient in $\widetilde{f}^{(b)}(w)\widetilde{f}^{(a)}(z)|\mathscr{C}\rangle$ would be zero. In this case, we can see that $\widetilde{d}^{(ab)}(z-w)\widetilde{f}^{(a)}(z)\widetilde{f}^{(b)}(w)|\mathscr{C}\rangle$ would also become zero for the configuration $\mathscr{C}-\mathfrak{b}-\mathfrak{a}$ due to the factor $\widetilde{d}^{(ab)}(z-w)$ and the $\delta$-functions in the actions of the currents. As a result,

$$\widetilde{d}^{(ab)}(z-w)\widetilde{f}^{(a)}(z)\widetilde{f}^{(b)}(w) \simeq (-1)^{|a||b|}\widetilde{d}^{(ab)}(z-w)\widetilde{f}^{(b)}(w)\widetilde{f}^{(a)}(z).\qquad(131)$$

Likewise, the operator $\widetilde{e}^{(a)}(z)$ acts as

$$\widetilde{e}^{(a)}(z)|\mathscr{C}\rangle = \sum_{\mathfrak{a}\in\mathrm{Add}(\mathscr{C})} \delta(z-\epsilon_\mathfrak{a})\widetilde{E}[\mathscr{C}\to\mathscr{C}-\mathfrak{a}]|\mathscr{C}-\mathfrak{a}\rangle,\qquad(132)$$

where

$$\widetilde{E}[\mathscr{C} \to \mathscr{C} + \mathfrak{a}] = \varsigma(\mathscr{C} \to \mathscr{C} + \mathfrak{a})\left(\varpi_{\mathscr{C} \to \mathscr{C} + \mathfrak{a}} \lim_{x=\epsilon_\mathfrak{a}} \zeta(x - \epsilon_\mathfrak{a})\widetilde{\Psi}_{\mathscr{C}}^{(a)}(x)\right)^{1/2}, \tag{133}$$

for some signs $\varsigma$ and $\varpi$. Therefore,

$$\widetilde{d}^{(ba)}(z - w)\widetilde{e}^{(a)}(z)\widetilde{e}^{(b)}(w) \simeq (-1)^{|a||b|}\widetilde{d}^{(ba)}(z - w)\widetilde{e}^{(b)}(w)\widetilde{e}^{(a)}(z), \tag{134}$$

with a similar condition on the signs.

For the $\widetilde{\psi}\widetilde{e}$ relation, consider adding the atom as $|\mathscr{C}\rangle \to |\mathscr{C} + \mathfrak{b}\rangle$. The ratio of the corresponding coefficients in $\widetilde{\psi}_\pm^{(a)}(z)\widetilde{e}^{(b)}(w)|\mathscr{C}\rangle$ and $\widetilde{e}^{(b)}(w)\widetilde{\psi}_\pm^{(a)}(z)|\mathscr{C}\rangle$ is

$$\frac{\widetilde{\Psi}_{\mathscr{C}+\mathfrak{b}}^{(a)}(z)}{\widetilde{\Psi}_{\mathscr{C}}^{(a)}(z)} = \widetilde{\phi}^{a \Leftarrow b}(z - \epsilon_\mathfrak{b}). \tag{135}$$

As $w \to \epsilon_\mathfrak{b}$, this recovers the $\widetilde{\psi}\widetilde{e}$ relation. The $\widetilde{\psi}\widetilde{f}$ relation can be checked similarly.

To verify the $\widetilde{e}\widetilde{f}$ relation, consider the terms in $\widetilde{e}^{(a)}(z)\widetilde{f}^{(a)}(w)|\mathscr{C}\rangle$ such that $|\mathscr{C}\rangle \to |\mathscr{C} - \mathfrak{a}\rangle \to |\mathscr{C}\rangle$. The coefficients are

$$\widetilde{E}[\mathscr{C} - \mathfrak{a} \to \mathscr{C}]\widetilde{F}[\mathscr{C} \to \mathscr{C} - \mathfrak{a}] = \pm\left(\pm \lim_{x=\epsilon_\mathfrak{a}} \zeta(x - \epsilon_\mathfrak{a})\widetilde{\Psi}_{\mathscr{C}-\mathfrak{a}}^{(a)}(x) \lim_{x=\epsilon_\mathfrak{a}} \zeta(x - \epsilon_\mathfrak{a})\widetilde{\Psi}_{\mathscr{C}-\mathfrak{a}}^{(a)}(x)\right)^{1/2}$$

$$= \pm \lim_{x=\epsilon_\mathfrak{a}} \zeta(x - \epsilon_\mathfrak{a})\widetilde{\phi}^{a \Leftarrow a}(x - \epsilon_\mathfrak{a})^{-1}\widetilde{\Psi}_{\mathscr{C}}^{(a)}(x). \tag{136}$$

Likewise, for $|\mathscr{C}\rangle \to |\mathscr{C} + \mathfrak{a}\rangle \to |\mathscr{C}\rangle$ in $\widetilde{f}^{(a)}(w)\widetilde{e}^{(a)}(z)|\mathscr{C}\rangle$, we have

$$\widetilde{F}[\mathscr{C} + \mathfrak{a} \to \mathscr{C}]\widetilde{E}[\mathscr{C} \to \mathscr{C} + \mathfrak{a}] = \pm \lim_{x=\epsilon_\mathfrak{a}} \zeta(x - \epsilon_\mathfrak{a})\widetilde{\Psi}_{\mathscr{C}}^{(a)}(x)$$

$$= \pm \lim_{x=\epsilon_\mathfrak{a}} \zeta(x - \epsilon_\mathfrak{a})\widetilde{\Psi}_{\mathscr{C}}^{(a)}(x). \tag{137}$$

On the other hand, when $\mathfrak{a} \neq \mathfrak{b}$ (for either $a = b$ or $a \neq b$), the ratio of the corresponding coefficients in $\widetilde{e}^{(a)}(z)\widetilde{f}^{(b)}(w)|\mathscr{C}\rangle$ and $\widetilde{f}^{(b)}(w)\widetilde{e}^{(a)}(z)|\mathscr{C}\rangle$ is

$$\pm\left(\frac{\lim_{x=\epsilon_\mathfrak{a}} \zeta(x - \epsilon_\mathfrak{a})\widetilde{\Psi}_{\mathscr{C}-\mathfrak{b}}^{(a)}(x) \lim_{x=\epsilon_\mathfrak{b}} \zeta(x - \epsilon_\mathfrak{b})\widetilde{\Psi}_{\mathscr{C}-\mathfrak{b}}^{(b)}(x)}{\lim_{x=\epsilon_\mathfrak{b}} \zeta(x - \epsilon_\mathfrak{b})\widetilde{\Psi}_{\mathscr{C}+\mathfrak{a}-\mathfrak{b}}^{(b)}(x) \lim_{x=\epsilon_\mathfrak{a}} \zeta(x - \epsilon_\mathfrak{a})\widetilde{\Psi}_{\mathscr{C}}^{(a)}(x)}\right)^{1/2}$$

$$= \begin{cases} \pm\left(\displaystyle\prod_{I \in \{a \to b\}} \dfrac{-\zeta(\epsilon_\mathfrak{b} - \epsilon_\mathfrak{a} + \epsilon - \epsilon_I)}{\zeta(\epsilon_\mathfrak{b} - \epsilon_\mathfrak{a} - \epsilon_I)}\right)^{-1}[a \leftrightarrow b], & a \neq b, \\[20pt] \pm\left(\displaystyle\prod_{I \in \{a \to a\}} \dfrac{-\zeta(\epsilon - \epsilon_I)}{\zeta(-\epsilon_I)}\left(\dfrac{-\zeta(\epsilon_\mathfrak{b} - \epsilon_\mathfrak{a} + \epsilon - \epsilon_I)}{\zeta(\epsilon_\mathfrak{b} - \epsilon_\mathfrak{a} - \epsilon_I)}[\mathfrak{a} \leftrightarrow \mathfrak{b}]\right)\right)^{-1}, & a = b. \end{cases} \tag{138}$$

As $z \to \epsilon_\mathfrak{a}$ and $w \to \epsilon_\mathfrak{b}$, we have

$$\widetilde{\phi}^{a \Leftarrow b}(z - w)\widetilde{e}^{(a)}(z)\widetilde{f}^{(b)}(w) - (-1)^{|a||b|}\widetilde{f}^{(b)}(w)\widetilde{e}^{(a)}(z) \simeq \delta_{ab}\delta(z - w)\left(\widetilde{\psi}_+^{(a)}(w) - \widetilde{\psi}_-^{(a)}(z) - \widetilde{\omega}^{(a)}(z)\right), \tag{139}$$

with proper choices of the branch cut of the square root and the signs $\varsigma, \varpi, \varrho$. The signs can be determined in a manner similar to [40, §6] and [43, Appendix E]. Here, we have used the fact that

$$\sum_{p \in \text{poles}(F)} \lim_{x \to p} \zeta(x - p)F(x)\delta(z - p)\delta(w - p) = \delta(z - w)\left([F(w)]_+ - [F(z)]_-\right), \tag{140}$$

for functions $F(x)$, whose numerator and denominator are products of $\zeta(x + \dots)$, with only simple poles.

**General chambers** In the construction above of the double quiver algebra, we have assumed that every time a single atom (as opposed to multiple atoms) is added to/removed from the crystal state. As discussed in §3.3, this can only be the case when we are in the cyclic chambers. For the non-cyclic chambers,[23] there could possibly exist a crystal $\mathscr{C}$ such that $\mathscr{C} + \mathfrak{a}$ is not allowed as it has an inadmissible pole, while $\mathscr{C} + \mathfrak{a} + \mathfrak{b}$ is still allowed. In other words, we need to add (or remove) more than one atom at a single step.

We can ask if we can express such situations in the crystal representations. The naive answer is no. Suppose that there are no other configurations from adding an atom of colour $a$ to $\mathscr{C}$. Then if we take the raising operator $\widetilde{e}^{(a)}(z)$, we have

$$\widetilde{e}^{(a)}(z)|\mathscr{C}\rangle = 0. \tag{141}$$

As a result,

$$\widetilde{e}^{(b)}(w)\widetilde{e}^{(a)}(z)|\mathscr{C}\rangle = 0, \tag{142}$$

which implies that we do not obtain the crystal $\mathscr{C} + \mathfrak{a} + \mathfrak{b}$.

One possible way to resolve this issue is to introduce a new tuple of generators

$$\left(\widetilde{\psi}_{\pm}^{(ab)}, \widetilde{e}^{(ab)}, \widetilde{f}^{(ab)}, \widetilde{\omega}^{(ab)}\right), \tag{143}$$

such that they would directly connect the two crystals $|\mathscr{C}\rangle \longleftrightarrow |\mathscr{C} + \mathfrak{a} + \mathfrak{b}\rangle$. It is natural to expect that they would still satisfy the basic forms of the relations, with the bond factors and the eigenfunction $\widetilde{\Psi}^{(ab)}(z)$ changed accordingly.[24] The details of this possible extension, such as the number of required new generators, etc., still require further explorations, and we leave the study on the non-cyclic chambers to future work.[25]

## 5.3 Refined countings from the double quiver algebras

For theories with four supercharges, the unrefined partition functions are fully encoded by the crystals (up to signs) as the coefficients are just numbers. They are in one-to-one correspondence with the crystal states. In such cases, we can further refine the partition functions.

From the above prescription of the JK residues, this provides an equivariant refinement with respect to an extra U(1) action [91]. In other words, when computing the partition functions, we keep $\epsilon$ generic (instead of taking it to be 0) after the evaluation of the residues.

As we have seen above, in the double quiver algebra $\widetilde{\mathsf{Y}}$, we also need to make $\epsilon$ non-zero in order to keep the bond factors and the charge functions non-trivial. Therefore, we should be able to recover the refined counting from $\widetilde{\mathsf{Y}}$. This is quite straightforward by, for example, considering the actions of the $\widetilde{e}$ currents. Suppose that we start with the empty state $|\varnothing\rangle$ and reach the state $|\mathscr{C}\rangle$ following the process

$$|\varnothing\rangle \to |\mathfrak{a}_1\rangle \to |\mathfrak{a}_1 + \mathfrak{a}_2\rangle \to \cdots \to |\mathfrak{a}_1 + \mathfrak{a}_2 + \cdots + \mathfrak{a}_N\rangle = |\mathscr{C}\rangle, \tag{144}$$

which is allowed as long as the configuration is allowed at each step (namely, satisfying the melting rule). Then this state is one of the summands in

$$\widetilde{e}^{(a_N)}(z_N)\ldots\widetilde{e}^{(a_2)}(z_2)\widetilde{e}^{(a_1)}(z_1)|\varnothing\rangle. \tag{145}$$

---

[23]We assume that we still get the integer indices from the formula, or at least we would have some combinatorial structure for the twisted partition function.

[24]Recall that the factor $\widetilde{d}^{(ab)}(z-w)$ guarantees that the atoms added or removed at non-generic positions would also satisfy the relations of the algebra. It could also be possible that there is a formulation with similar extra generators such as $\widetilde{e}^{(ab)}(z)$ so that there would be no need to include the factors $\widetilde{d}^{(ab)}(z-w)$. This could potentially involve a tower of new currents $\widetilde{e}^{(a_1 a_2 \cdots)}(z)$.

[25]We should also mention that this does not mean that the double quiver algebra $\widetilde{\mathsf{Y}}$ and the quiver algebra $\mathsf{Y}$ (see §6 and Appendix A) cannot describe the non-cyclic chambers. For instance, in the case of the conifold, it was argued in [82] that all the chambers can be packaged as the semi-Fock representations [90] of the corresponding quiver Yangian.

Recall that

$$\widetilde{E}[\mathscr{C} \to \mathscr{C} + \mathfrak{a}] = \pm \left( \pm \lim_{x = \epsilon_{\mathfrak{a}}} \zeta(x - \epsilon_{\mathfrak{a}}) \widetilde{\Psi}_{\mathscr{C}}^{(a)}(x) \right)^{1/2}. \tag{146}$$

Denoting the state at the $i^{\text{th}}$ step as $|\mathscr{C}_i\rangle$, the refined coefficient for this state is

$$\widetilde{\xi}_{\mathcal{N}=4} \prod_{i=0}^{N-1} \widetilde{E}[\mathscr{C}_i \to \mathscr{C}_{i+1}]^2, \tag{147}$$

where we have restored the factor

$$\widetilde{\xi}_{\mathcal{N}=4} = \begin{cases} \left( -\dfrac{\eta(\tau)^3}{i\theta_1(\tau,\epsilon)} \right)^N, & \text{elliptic}, \\[3mm] \left( -\dfrac{1}{2i\sin(\pi\epsilon)} \right)^N, & \text{trigonometric}, \\[3mm] \left( -\dfrac{1}{\epsilon} \right)^N, & \text{rational}. \end{cases} \tag{148}$$

Equivalently, we may consider residues of the overlap the state (145) with the crystal state $|\mathscr{C}\rangle$:

$$\text{Res}_{z_N = \epsilon(\mathfrak{a}_N)} \dots \text{Res}_{z_1 = \epsilon(\mathfrak{a}_1)} \left\langle \mathscr{C} \left| \widetilde{e}^{(a_N)}(z_N) \dots \widetilde{e}^{(a_2)}(z_2) \widetilde{e}^{(a_1)}(z_1) \right| \varnothing \right\rangle. \tag{149}$$

Then the refined coefficient for this state is obtained by an absolute square of this expression, together with a factor of $\widetilde{\xi}_{\mathcal{N}=4}$.

# 6 Comparisons of quiver algebras

As we are now going to see, at least in the cyclic chambers, the admissible and inadmissible poles can be separated in the sense that one can construct functions $\Psi_{\mathscr{C},\pm}^{(a)}(z)$ and $\phi^{a \Leftarrow b}(z)$ similar to their tilded versions but with only admissible poles. This in fact gives rise to the quiver Yangians [40–43]. Again, we would only consider the cases satisfying the no-overlap condition in §3.

In the following, we will refer to the quiver BPS algebra of [40–43] as the *single quiver algebra*, when we want to emphasize that this is different from the double quiver algebra discussed in earlier sections.

## 6.1 Quiver BPS algebras Υ from JK residues

The key function that appeared in the single quiver algebra is

$$\Psi_{\mathscr{C}}^{(a)}(z) = {}^{\#}\psi^{(a)}(z) \prod_{b \in Q_0} \prod_{\mathfrak{b} \in \mathscr{C}} \phi^{a \Leftarrow b}(z - \epsilon_{\mathfrak{b}}), \tag{150}$$

where[26]

$$\phi^{a \Leftarrow b}(z) = (-1)^{|b \to a|} \frac{\prod\limits_{I \in \{a \to b\}} \zeta(z + \epsilon_I)}{\prod\limits_{I \in \{b \to a\}} \zeta(z - \epsilon_I)}. \tag{151}$$

Here, $|b \to a|$ denotes the number of arrows from $b$ to $a$ so that

$$\phi^{a \Leftarrow b}(z) \phi^{b \Leftarrow a}(-z) = 1. \tag{152}$$

---

[26]For the rational case, $\zeta(z)$ is the same as the one in [43]. For the trigonometric case, there is just a rescaling of the variable. For the elliptic case, they also agree up to an overall constant coefficient.

We are now going to show that $\Psi_{\mathscr{C}}^{(a)}(z)$ contains the poles that precisely correspond to the addable and removable atoms in $\mathscr{C}$.

The argument is very similar to obtaining the crystal structure from JK residues in [38]. In particular, given $\eta = e_1 + \cdots + e_N$ where $e_i \in \mathbb{R}^N$ has 1 in its $i^{\text{th}}$ position with other entries vanishing, the admissible hyperplanes/vectors has a tree structure. First, we have at least one $e_j$, and then $e_k - e_j$ is allowed while $e_j - e_k$ is not. If $e_k - e_j$ are chosen, then $e_l - e_k$ is allowed while $e_k - e_l$ is not, and this procedure can be repeated. Moreover, we learn from the analysis therein that the poles of the vector multiplest would never contribute. To compare with the JK residue formula, we shall use $u_i^{(a)}$ to label the weights $\epsilon_{\mathfrak{a}}$ in the crystal. We have the following possibilities (with an extra case 0):

0. If the pole $\left(z - u_j^{(b)} - \epsilon_I\right)^{-1}$ corresponds to an existing atom on the boundary (but not the surface) of the molecule/crystal state, namely, $\mathfrak{a} \in \mathscr{C}$ but $I \cdot \mathfrak{a} \notin \mathscr{C}$ for any $I$, then this pole is not cancelled by any factor in the numerator in $\Psi_{\mathscr{C}}^{(a)}(z)$ (as opposed to $Z_{\text{1-loop}}$ in the JK residue formula). This gives a removable atom:

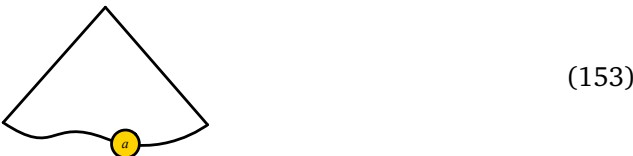

$$(153)$$

1. The pole $\left(z - u_j^{(b)} - \epsilon_I\right)^{-1}$ could also correspond to an existing atom $u_i^{(a)}$ that does not belong to case 0. In other words, there exists at least one $I'$ such that $I' \cdot \mathfrak{a} \in \mathscr{C}$. Then the numerator contains the factor[27]

$$-\left(u_{j'}^{(b')} - z + \epsilon - \epsilon_{I'}\right) = z - u_{j'}^{(b')} + \epsilon_{I'} \qquad (\epsilon = 0), \qquad (154)$$

which cancels the pole. Pictorially, we have

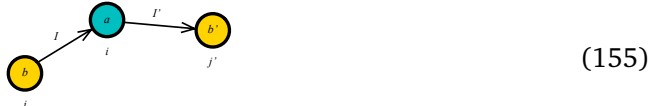

$$(155)$$

where the corresponding pole is coloured blue.

2. For an addable atom, there is a pole $\left(z - u_{j_1}^{(b_1)} - \epsilon_{I_1}\right)^{-1}$. There could be multiple factors in the denominator giving this pole, i.e., $z = u_{j_1}^{(b_1)} + \epsilon_{I_1} = u_{j_2}^{(b_2)} + \epsilon_{I_2} = \dots$. In the numerator, there would be some factors

$$z = u_{j_1}^{(b_1)} + \epsilon_{I_1} = u_{i_1}^{(d_1)} + \epsilon_{J_1} + \epsilon_{I_1} = \cdots = u_{i_n}^{(d_n)} + \epsilon_{J_n} + \cdots + \epsilon_{I_1} = u_{i_n}^{(d_n)} - \epsilon_J, \quad (156)$$

where $-\epsilon_J$ in the last equality with $J$ pointing backwards in the molecule/crystal state comes from the loop constraints. One such factor would cancel one of the poles from $z = u_{j_1}^{(b_1)} + \epsilon_{I_1} = u_{j_2}^{(b_2)} + \epsilon_{I_2}$, and the remaining pole will form another pair with $z = u_{j_3}^{(b_3)} + \epsilon_{I_3}$ which can be cancelled by another factor whose corresponding arrow points backwards in the numerator. This can be repeated pairwise and there will be a simple pole left. In diagrams, we have

---

[27] Since the numerator and the denominator are not paired anymore, we can safely take $\epsilon \to 0$.

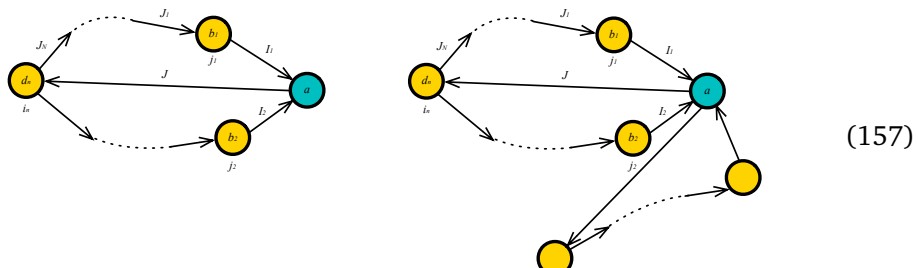

$$(157)$$

3. If a pole corresponds to an atom that is neither addable nor in the molecule/crystal state, then the cancellation of this pole is the same as in (156). This can be depicted as

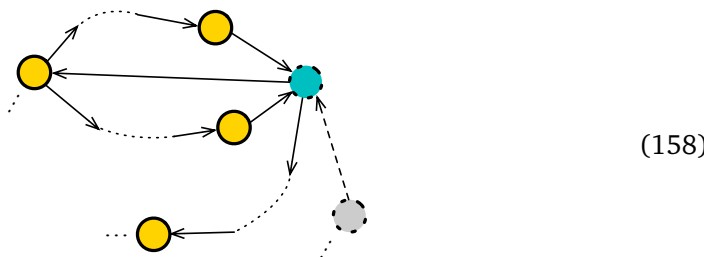

$$(158)$$

where the dashed grey atom is not in the molecule/crystal state so that the blue one ($z$) cannot be added.

As a result, the poles in $\Psi_{\mathscr{C}}^{(a)}(z)$ give exactly the addable and removable atoms in the molecule/crystal state. Following the derivation similar to the ones in §5.2 and in [40], one can get the defining relations of the quiver Yangians/quiver BPS algebras and the crystal representations. We list them in Appendix A for reference.

## 6.2 Connections to the double quiver algebras $\widetilde{\mathsf{Y}}$

As the quiver Yangians are the BPS algebras for the quiver gauge theories, it is natural to wonder how the double quiver algebra $\widetilde{\mathsf{Y}}$ introduced in §5, which are also constructed to encode the information of BPS states, can be related to the single quiver algebra.

On the one hand, the single quiver algebras $\mathsf{Y}$ capture the addable and removable atoms in the crystal states without any extra poles. They enjoy many nice properties, including the coproduct structure and the connections to integrable systems via the Bethe/gauge correspondence, and are expected to serve as some "universal algebras" for certain vertex operator algebras. On the other hand, the double quiver algebras $\widetilde{\mathsf{Y}}$ are designed to contain the full information from the JK residue formula. Hence, this method would also be applicable to the quiver gauge theories with two supercharges in §7.

One might also wonder why there could be two different algebras that encode the BPS states. Notice that the single quiver Yangian $\mathsf{Y}$ serves as the BPS algebra that counts the unrefined indices while the double quiver algebra $\widetilde{\mathsf{Y}}$ contains the information of the refined indices. In other words, there might be more structures from the refinement, and a crystal state for $\widetilde{\mathsf{Y}}$ is some "equivalence class" of the BPS states belonging to the same fixed point. It is natural to expect that there could exist a larger algebra whose representation has states including these more structures, either for theories with four supercharges or for those with two supercharges.

**Differences from crystal representations** As the two algebras both have crystal representations, it is most straightforward to compare them via their actions on the crystal states. Let us lift the relevant functions $\widetilde{\phi}^{a\Leftarrow b}(z)$ and $\widetilde{\Psi}_{\mathscr{C}}^{(a)}(z)$ with $n$ parameters $\epsilon_{1,\dots,n}$ to $(n+1)$-parametric ones with an extra parameter $\varepsilon$ as

$$
\widetilde{\phi}_{\varepsilon}^{a\Leftarrow b}(z) := \begin{cases} \left( \dfrac{\zeta(z)\zeta(-z)}{\zeta(z+\epsilon)\zeta(-z+\epsilon)} \right)^{\varepsilon} \left( \displaystyle\prod_{I\in\{a\to a\}} \left( \dfrac{\zeta(\epsilon-\epsilon_I)\,\zeta(z+\epsilon-\epsilon_I)}{\zeta(-\epsilon_I)\zeta(z+\epsilon_I)} \right)^{\varepsilon} \dfrac{-\zeta(z-\epsilon+\epsilon_I)}{\zeta(z-\epsilon_I)} \right), & b=a\,, \\[2.5em] \left( \displaystyle\prod_{I\in\{a\to b\}} \dfrac{(-1)^{\varepsilon}\zeta(z-\epsilon+\epsilon_I)}{\zeta(z+\epsilon_I)^{\varepsilon}} \right) \left( \displaystyle\prod_{I\in\{b\to a\}} \dfrac{-\zeta(z+\epsilon-\epsilon_I)^{\varepsilon}}{\zeta(z-\epsilon_I)} \right), & b\neq a\,, \end{cases}
$$
(159)

and

$$
\widetilde{\Psi}_{\mathscr{C},\varepsilon}^{(a)}(z) := {}^{\#}\widetilde{\psi}_{\varepsilon}^{(a)}(z) \prod_{b\in Q_0}\prod_{\mathfrak{b}\in\mathscr{C}} \widetilde{\phi}_{\varepsilon}^{a\Leftarrow b}(z-\epsilon_{\mathfrak{b}})\,,
$$
(160)

where the framing factor is also modified as

$$
{}^{\#}\widetilde{\psi}_{\varepsilon}^{(a)}(z) := \left( \prod_{I\in\{a\to\infty\}} \frac{(-1)^{\varepsilon}\zeta(-z+\epsilon-v_I)}{\zeta(-z-v_I)^{\varepsilon}} \right) \left( \prod_{I\in\{\infty\to a\}} \frac{-\zeta(z+\epsilon-v_I)^{\varepsilon}}{\zeta(z-v_I)} \right).
$$
(161)

It is straightforward to see that

$$
\widetilde{\phi}_{\varepsilon=1}^{a\Leftarrow b}(z) = \widetilde{\phi}^{a\Leftarrow b}(z)\,, \qquad \widetilde{\phi}_{\varepsilon=0}^{a\Leftarrow b}(z)\Big|_{\epsilon=0} = \phi^{a\Leftarrow b}(z)\,,
$$
(162)

and

$$
\widetilde{\Psi}_{\mathscr{C},\varepsilon=1}^{(a)}(z) = \widetilde{\Psi}_{\mathscr{C}}^{(a)}(z)\,, \qquad \widetilde{\Psi}_{\mathscr{C},\varepsilon=0}^{(a)}(z)\Big|_{\epsilon=0} = \Psi_{\mathscr{C}}^{(a)}(z)\,.
$$
(163)

In other words, the bond factors of the quiver Yangian $\mathsf{Y}$ are "half" of those of the quiver algebra $\widetilde{\mathsf{Y}}$. This is the origin of the name "double quiver algebra."[28] The physics intuition is that the "double quiver algebra" incorporates generators for both BPS particles and BPS antiparticles, and hence all the possible particles inside the theory; the single quiver Yangian, by contrast, contains generators only for BPS particles.

We may also consider the uplifted currents with the relations such as

$$
\widetilde{d}^{(ba)}(z-w)^{\varepsilon}\widetilde{e}_{\varepsilon}^{(a)}(z)\widetilde{e}_{\varepsilon}^{(b)}(w) \simeq (-1)^{|a||b|}\left( \frac{\widetilde{\phi}_{\varepsilon}^{a\Leftarrow b}(z-w)}{\widetilde{\phi}_{\varepsilon}^{b\Leftarrow a}(w-z)} \right)^{1/2} \widetilde{d}^{(ba)}(z-w)^{\varepsilon}\widetilde{e}_{\varepsilon}^{(b)}(w)\widetilde{e}_{\varepsilon}^{(a)}(z)\,,
$$
(164a)

$$
\widetilde{d}^{(ab)}(z-w)^{\varepsilon}\widetilde{f}_{\varepsilon}^{(a)}(z)\widetilde{f}_{\varepsilon}^{(b)}(w) \simeq (-1)^{|a||b|}\widetilde{d}^{(ab)}(z-w)^{\varepsilon}\left( \frac{\widetilde{\phi}_{\varepsilon}^{a\Leftarrow b}(z-w)}{\widetilde{\phi}_{\varepsilon}^{b\Leftarrow a}(w-z)} \right)^{-1/2} \widetilde{f}_{\varepsilon}^{(b)}(w)\widetilde{f}_{\varepsilon}^{(a)}(z)\,,
$$
(164b)

$$
\widetilde{\phi}_{\varepsilon}^{a\Leftarrow b}(z-w-c)^{\varepsilon/2}\widetilde{e}_{\varepsilon}^{(a)}(z)\widetilde{f}_{\varepsilon}^{(b)}(w) - (-1)^{|a||b|}\widetilde{\phi}_{\varepsilon}^{a\Leftarrow b}(z-w-c)^{-\varepsilon/2}\widetilde{f}_{\varepsilon}^{(b)}(w)\widetilde{e}_{\varepsilon}^{(a)}(z)
$$
$$
\simeq \delta_{ab}\left( \delta(z-w-c)\widetilde{\psi}_{+,\varepsilon}^{(a)}(w+c/2) - \delta(z-w+c)\widetilde{\psi}_{-,\varepsilon}^{(a)}(z+c/2) - \delta(z-w)\varepsilon\widetilde{\omega}^{(a)}(z) \right).
$$
(164c)

The other relations can be written similarly. When $\varepsilon=1$, this recovers the double quiver algebra $\widetilde{\mathsf{Y}}$. Taking $\varepsilon=0$ and then $\epsilon=0$, we get the quiver Yangian $\mathsf{Y}$ (up to some overall factors $(ZW)^{t\chi_{ab}/2}$ that removes fractional powers for chiral quivers in the trigonometric and elliptic cases). Due to (140), we need simple poles in $\widetilde{\Psi}_{\varepsilon}^{(a)}(z)$ to obtain the crystal representations. In other words, only $\varepsilon=0,1$ are relevant for BPS counting.

---

[28]In the literature there are several concepts/defintions of doubles of the algebras (such as the quantum double). The "double" in the double quiver algebra is different from these concepts.

**Subalgebras of enhanced double quiver algebras** $\widetilde{\mathsf{Y}}^\sharp$ Since $\phi^{a\Leftarrow b}(z)$ is "half" of $\widetilde{\phi}^{a\Leftarrow b}(z)$, it is natural to wonder if we could realize the single quiver algebra $\mathsf{Y}$ as some subalgebra of the double quiver algebra $\widetilde{\mathsf{Y}}$. To simplify our discussions, we shall take $c = 0$ below as this is the only situation admitting the crystal representations.

It turns out that we need to slightly enlarge the double quiver algebra to recover the original quiver algebra. In the double quiver algebra $\widetilde{\mathsf{Y}}$, the $\widetilde{e}\widetilde{e}$ currents commute. On the other hand, exchanging the $ee$ currents picks up a bond factor in the quiver Yangian $\mathsf{Y}$. Therefore, to obtain the non-trivial commutation relations for the $ee$ currents, we need to combine the $\widetilde{e}$ currents with some other elements that would produce some non-trivial factors when passing through the $\widetilde{e}$ currents. The very first candidate would be the $\widetilde{\psi}$ currents, but the expressions of the form "$(\widetilde{\psi}\widetilde{e})(\widetilde{\psi}\widetilde{e})$" (which will be made precise below) does not give the desired $ee$ relations since the extra factors get cancelled.

To keep the non-trivial factors upon exchanging the currents, let us extend this subalgebra to the algebra $\widetilde{\mathsf{Y}}^\sharp$ (which we call the *enhanced double quiver algebra*),[29] with the extra currents $\overline{\psi}^{(a)}(z)$ satisfying

$$\overline{\psi}^{(a)}(z)\overline{\psi}^{(b)}(w) \simeq \overline{\psi}^{(b)}(w)\overline{\psi}^{(a)}(z), \tag{165a}$$

$$\overline{\psi}^{(a)}(z)\widetilde{\psi}_{\pm}^{(b)}(w) \simeq \widetilde{\psi}_{\pm}^{(b)}(w)\overline{\psi}^{(a)}(z), \tag{165b}$$

$$\overline{\psi}^{(a)}(z)\widetilde{\omega}^{(b)}(w) \simeq \widetilde{\omega}^{(b)}(w)\overline{\psi}^{(a)}(z), \tag{165c}$$

$$\overline{\psi}^{(a)}(z)\widetilde{e}^{(b)}(w) \simeq \overline{\phi}^{a\Leftarrow b}(z-w)\widetilde{e}^{(b)}(w)\overline{\psi}^{(a)}(z), \tag{165d}$$

$$\overline{\psi}^{(a)}(z)\widetilde{f}^{(b)}(w) \simeq \overline{\phi}^{b\Leftarrow a}(w-z)\widetilde{f}^{(b)}(w)\overline{\psi}^{(a)}(z), \tag{165e}$$

where

$$\overline{\phi}^{a\Leftarrow b}(z) = \prod_{I\in\{a\to b\}} \zeta(z + \epsilon_I). \tag{166}$$

Then

$$\overline{\phi}^{a\Leftarrow b}(z)\overline{\phi}^{b\Leftarrow a}(-z)^{-1} = \phi^{a\Leftarrow b}(z). \tag{167}$$

As a result,

$$\overline{\phi}^{b\Leftarrow a}(-z)\overline{\phi}^{a\Leftarrow b}(z)^{-1} = \phi^{b\Leftarrow a}(-z) = \phi^{a\Leftarrow b}(z)^{-1}. \tag{168}$$

Moreover, compared to the single quiver Yangian $\mathsf{Y}$, there are some extra currents $\widetilde{\omega}^{(a)}(z)$ in $\widetilde{\mathsf{Y}}$ and $\widetilde{\mathsf{Y}}^\sharp$. Therefore, let us take the subalgebra $\widetilde{\mathsf{Y}}^\sharp/\sim$, where the equivalence relation is given by $\widetilde{\omega}_n^{(a)} \sim 0$. This subalgebra would then have no $\widetilde{\omega}$ currents involved.

Now, roughly speaking, the combination $\overline{\psi}\widetilde{e}$ should give the right factors in the $ee$ relations following (167). To discuss this precisely, we should define the combination of the currents properly. Let us consider the normal ordering $(\dots)$ of two operators defined as

$$(A(z)B(z)) = \frac{1}{2\pi i}\oint_z dw\,(w-z)^{-1}A(w)B(z). \tag{169}$$

Then we have

$$\left(\overline{\psi}^{(a)}(z)\widetilde{e}^{(a)}(z)\right)\left(\overline{\psi}^{(b)}(w)\widetilde{e}^{(b)}(w)\right)$$
$$= \frac{1}{(2\pi i)^2}\oint_z\oint_w dz\,dw\,(z'-z)^{-1}(w'-w)^{-1}\widetilde{\psi}_V(z')\overline{\psi}^{(a)}(z')\widetilde{e}^{(a)}(z)\widetilde{\psi}_V(w')\overline{\psi}^{(b)}(w')\widetilde{e}^{(b)}(w)$$

---

[29]We have the enhanced double quiver Yangian for the rational case, and enhanced double quiver quantum toroidal algebra for the trigonometric case.

$$\simeq \frac{\overline{\phi}^{a\Leftarrow b}(z-w)\overline{\phi}^{b\Leftarrow a}(w-z)^{-1}}{(2\pi i)^2}\oint_z\oint_w dzdw\,(z'-z)^{-1}(w'-w)^{-1}\overline{\psi}^{(b)}(w')\widetilde{e}^{(b)}(w)\overline{\psi}^{(a)}(z')\widetilde{e}^{(a)}(z)$$

$$= \phi^{a\Leftarrow b}(z-w)\left(\overline{\psi}^{(b)}(w)\widetilde{e}^{(b)}(w)\right)\left(\overline{\psi}^{(a)}(z)\widetilde{e}^{(a)}(z)\right). \tag{170}$$

Therefore, the identification is

$$e^{(a)}(z) = \left(\overline{\psi}^{(a)}(z)\widetilde{e}^{(a)}(z)\right). \tag{171}$$

Likewise,

$$f^{(a)}(z) = \left(\overline{\psi}^{(a)}(z)\widetilde{f}^{(a)}(z)\right). \tag{172}$$

Using the $\widetilde{e}\widetilde{f}$ relations and considering $\left[\left(\overline{\psi}^{(a)}(z)\widetilde{e}^{(a)}(z)\right),\left(\overline{\psi}^{(b)}(w)\widetilde{f}^{(b)}(w)\right)\right]$, we can also get the expressions for the $\psi_\pm$ currents:

$$\psi_\pm^{(a)}(z) = \left(\overline{\psi}^{(a)}(z)\widetilde{\psi}_\pm^{(a)}(z)\overline{\psi}^{(a)}(z)\right), \tag{173}$$

in the limit $\epsilon \to 0$. In other words, the quiver Yangian $\mathsf{Y}$ is a subalgebra of the enhanced double quiver algebra $\widetilde{\mathsf{Y}}^\sharp$ with the parameter $\epsilon$ specified to be 0. In particular, this means that the converse does not fully recover the double quiver algebra $\widetilde{\mathsf{Y}}$. With the currents such as $\left(\overline{\psi}e\right)$, $\left(\overline{\psi}f\right)$ and $\left(\overline{\psi}\psi_\pm\overline{\psi}\right)$, this yields $\widetilde{\mathsf{Y}}\big|_{\widetilde{\omega}\sim 0,\epsilon=0}$ where (the left-hand side of) the $\widetilde{e}\widetilde{f}$ relations become commutators and the other pairs of currents commute.

# 7 Double quiver algebras for theories with two supercharges

Now, let us consider the theories with two supercharges. The double quiver algebras can be constructed in a similar manner to those in §5. Therefore, this is still restricted to the cases where the JK residue formula applies,[30] in particular those satisfying the no-overlap condition in §3.

## 7.1 Defining relations

Let us first define the double quiver algebra $\widetilde{\mathsf{Y}}$. Given a quiver and some $J$-/$E$-term relations, the double quiver algebra $\widetilde{\mathsf{Y}}$ has four sets of generating currents, $\widetilde{\psi}^{(a)}(z)$, $\widetilde{e}^{(a)}(z)$, $\widetilde{f}^{(a)}(z)$ and $\widetilde{\omega}^{(a)}(z)$, where $a \in Q_0$ labels the nodes in the quiver. The defining relations are

$$\widetilde{\omega}^{(a)}(z)\widetilde{\omega}^{(b)}(w) \simeq \widetilde{\omega}^{(b)}(w)\widetilde{\omega}^{(a)}(z), \tag{174a}$$

$$\widetilde{\psi}_+^{(a)}(z)\widetilde{\psi}_+^{(b)}(w) \simeq \widetilde{\psi}_+^{(b)}(w)\widetilde{\psi}_+^{(a)}(z), \tag{174b}$$

$$\widetilde{\psi}_-^{(a)}(z)\widetilde{\psi}_-^{(b)}(w) \simeq \widetilde{\phi}^{a\Leftarrow b}(z-w+2c)\widetilde{\phi}^{a\Leftarrow b}(z-w-2c)^{-1}\widetilde{\psi}_-^{(b)}(w)\widetilde{\psi}_-^{(a)}(z), \tag{174c}$$

$$\widetilde{\psi}_+^{(a)}(z)\widetilde{\psi}_-^{(b)}(w) \simeq \widetilde{\phi}^{a\Leftarrow b}(z-w+c)\widetilde{\phi}^{a\Leftarrow b}(z-w-c)^{-1}\widetilde{\psi}_-^{(b)}(w)\widetilde{\psi}_+^{(a)}(z), \tag{174d}$$

$$\widetilde{\psi}_\pm^{(a)}(z)\widetilde{\omega}^{(b)}(w) \simeq \widetilde{\phi}^{a\Leftarrow b}(z-w+c\mp c/2)^{-1}\widetilde{\phi}^{a\Leftarrow b}(z-w-c\pm c/2)\widetilde{\omega}^{(b)}(w)\widetilde{\psi}_\pm^{(a)}(z), \tag{174e}$$

$$\widetilde{d}^{(ba)}(z-w)\widetilde{\psi}_\pm^{(a)}(z)\widetilde{e}^{(b)}(w) \simeq \widetilde{\phi}^{a\Leftarrow b}(z-w+c\mp c/2)\widetilde{d}^{(ba)}(z-w)\widetilde{e}^{(b)}(w)\widetilde{\psi}_\pm^{(a)}(z), \tag{174f}$$

$$\widetilde{d}^{(ab)}(z-w)\widetilde{\psi}_\pm^{(a)}(z)\widetilde{f}^{(b)}(w) \simeq \widetilde{\phi}^{a\Leftarrow b}(z-w-c\pm c/2)^{-1}\widetilde{d}^{(ab)}(z-w)\widetilde{f}^{(b)}(w)\widetilde{\psi}_\pm^{(a)}(z), \tag{174g}$$

---

[30]As commented above in §5.1, we shall still mainly focus on the cyclic chambers.

$$\delta(z-w)\widetilde{\phi}^{d\Leftarrow a}(u-z-c)\widetilde{e}^{(d)}(u)\widetilde{\omega}^{(a)}(z) + \delta(u-w)\widetilde{\phi}^{a\Leftarrow d}(z-w-c)\widetilde{e}^{(a)}(z)\widetilde{\omega}^{(d)}(w)$$
$$\simeq \delta(z-w)\widetilde{\omega}^{(a)}(z)\widetilde{e}^{(d)}(u) + \delta(u-w)\widetilde{\omega}^{(a)}(u)\widetilde{e}^{(a)}(z), \tag{174h}$$

$$\delta(z-w)\widetilde{\phi}^{d\Leftarrow a}(u-z-c)^{-1}\widetilde{f}^{(d)}(u)\widetilde{\omega}^{(a)}(z) + \delta(u-w)\widetilde{\phi}^{a\Leftarrow d}(z-w-c)^{-1}\widetilde{f}^{(a)}(z)\widetilde{\omega}^{(d)}(w)$$
$$\simeq \delta(z-w)\widetilde{\omega}^{(a)}(z)\widetilde{f}^{(d)}(u) + \delta(u-w)\widetilde{\omega}^{(d)}(u)\widetilde{f}^{(a)}(z), \tag{174i}$$

$$\widetilde{d}^{(ba)}(z-w)\widetilde{e}^{(a)}(z)\widetilde{e}^{(b)}(w) \simeq (-1)^{|a||b|}\widetilde{d}^{(ba)}(z-w)\widetilde{e}^{(b)}(w)\widetilde{e}^{(a)}(z), \tag{174j}$$

$$\widetilde{d}^{(ab)}(z-w)\widetilde{f}^{(a)}(z)\widetilde{f}^{(b)}(w) \simeq (-1)^{|a||b|}\widetilde{d}^{(ab)}(z-w)\widetilde{f}^{(b)}(w)\widetilde{f}^{(a)}(z), \tag{174k}$$

$$\widetilde{\phi}^{a\Leftarrow b}(z-w-c)\widetilde{e}^{(a)}(z)\widetilde{f}^{(b)}(w) - (-1)^{|a||b|}\widetilde{f}^{(b)}(w)\widetilde{e}^{(a)}(z)$$
$$\simeq \delta_{ab}\left(\delta(z-w-c)\widetilde{\psi}_+^{(a)}(w+c/2) - \delta(z-w+c)\widetilde{\psi}_-^{(a)}(z+c/2) - \delta(z-w)\widetilde{\omega}^{(a)}(z)\right). \tag{174l}$$

The form of the relations looks the same as their four-supercharge counterparts, but the bond factors are now different:

$$\widetilde{\phi}^{a\Leftarrow b}(z) := \begin{cases} \zeta(z)\zeta(-z)\dfrac{\prod\limits_{\Lambda\in\{aa\}}\zeta(\epsilon_\Lambda)\zeta(z-\epsilon_\Lambda)\zeta(z+\epsilon_\Lambda)}{\prod\limits_{I\in\{a\to a\}}\zeta(\epsilon_I)\zeta(z-\epsilon_I)\zeta(z+\epsilon_I)}, & b=a, \\[2em] \dfrac{\prod\limits_{\Lambda\in\{ab\}}\left(-\zeta\left(\varsigma_\Lambda^{a\Leftarrow b}z-\epsilon_\Lambda\right)\right)}{\left(\prod\limits_{I\in\{a\to b\}}(-\zeta(z+\epsilon_I))\right)\left(\prod\limits_{I\in\{b\to a\}}\zeta(z-\epsilon_I)\right)}, & b\neq a. \end{cases} \tag{175}$$

Here, $\varsigma_\Lambda^{a\Leftarrow b}$ is the sign determined by the choice of the orientation of $\Lambda$, namely,

$$\varsigma_\Lambda^{a\Leftarrow b} := \begin{cases} +, & s(\Lambda)=b,\ t(\Lambda)=a, \\ -, & s(\Lambda)=a,\ t(\Lambda)=b. \end{cases} \tag{176}$$

In particular, this implies that

$$\varsigma_\Lambda^{b\Leftarrow a} = -\varsigma_\Lambda^{a\Leftarrow b}, \tag{177}$$

which guarantees that

$$\widetilde{\phi}^{b\Leftarrow a}(-z) = \widetilde{\phi}^{a\Leftarrow b}(z). \tag{178}$$

In some of the relations, the factor $\widetilde{d}^{(ab)}(z)$ is basically the denominator of the bond factor:

$$\widetilde{d}^{(ab)}(z) := \left(\prod\limits_{I\in\{a\to b\}}(z+\epsilon_I)\right)\left(\prod\limits_{I\in\{b\to a\}}(-z+\epsilon_I)\right). \tag{179}$$

Notice that

$$\widetilde{d}^{(ab)}(z) = \widetilde{d}^{(ba)}(-z). \tag{180}$$

Moreover, we have chosen the $\mathbb{Z}_2$-grading such that $|a| \equiv |\Lambda_{aa}| + 1 \pmod 2$ for theories with two supercharges.[31]

As the bond factors are not necessarily homogeneous, there could be fractional powers in the Laurent expansions of the currents for the trigonometric and elliptic cases. To avoid such fractional powers, we may rescale $\widetilde{\phi}^{a\Leftarrow b}(z)$ as

$$\widetilde{\phi}^{a\Leftarrow b}(z) \to Z^{\frac{1}{2}(|\Lambda|-|\chi|)}\widetilde{\phi}^{a\Leftarrow b}(z), \tag{181}$$

---

[31]Under dimensional reduction from the 4d $\mathcal{N}=1$ quivers to the 2d $\mathcal{N}=(2,2)$ quivers (which are always given in the $\mathcal{N}=(0,2)$ language), this guarantees that the Bose/Fermi statistics would not change for the quiver nodes (although it is still not clear how the quiver algebras behave under dimensional reductions).

where

$$
\mathfrak{t} := \begin{cases} 0, & \text{rational}, \\ 1, & \text{trigonometric/elliptic}, \end{cases}
\tag{182}
$$

and $|\chi|$ (resp. $|\Lambda|$) denotes the number of chirals (resp. Fermis) between $a$ and $b$.

The mode generators can be obtained from the expansions of the currents (with the balancing (181)):

$$
\widetilde{\psi}_{\pm}^{(a)}(z) = \begin{cases} \displaystyle\sum_{n \in \mathbb{Z}} \widetilde{\psi}_n^{(a)} z^{-\left(n + \mathfrak{s}^{(a)}\right)}, & \text{rational}, \\ \displaystyle\sum_{n \in \mathbb{Z}} \widetilde{\psi}_{\pm,n}^{(a)} Z^{\mp\left(n + \mathfrak{s}^{(a)}\right)}, & \text{trigonometric}, \\ \displaystyle\sum_{n \in \mathbb{Z}}\sum_{\alpha \in \mathbb{Z}_{\geq 0}} \widetilde{\psi}_{\pm,n,\alpha}^{(a)} Z^{\mp\left(n + \mathfrak{s}^{(a)}\right)} \mathfrak{q}^{\alpha}, & \text{elliptic}, \end{cases}
\tag{183a}
$$

$$
\widetilde{e}^{(a)}(z) = \begin{cases} \displaystyle\sum_{n \in \mathbb{Z}_{>0}} \widetilde{e}_n^{(a)} z^{-n}, & \text{rational}, \\ \displaystyle\sum_{n \in \mathbb{Z}} \widetilde{e}_n^{(a)} Z^{-n}, & \text{trigonometric}, \\ \displaystyle\sum_{n \in \mathbb{Z}}\sum_{\alpha \in \mathbb{Z}_{\geq 0}} \widetilde{e}_{n,\alpha}^{(a)} Z^{\mp n} \mathfrak{q}^{\alpha}, & \text{elliptic}, \end{cases}
\tag{183b}
$$

$$
\widetilde{f}^{(a)}(z) = \begin{cases} \displaystyle\sum_{n \in \mathbb{Z}_{>0}} \widetilde{f}_n^{(a)} z^{-n}, & \text{rational}, \\ \displaystyle\sum_{n \in \mathbb{Z}} \widetilde{f}_n^{(a)} Z^{-n}, & \text{trigonometric}, \\ \displaystyle\sum_{n \in \mathbb{Z}}\sum_{\alpha \in \mathbb{Z}_{\geq 0}} \widetilde{f}_{n,\alpha}^{(a)} Z^{\mp n} \mathfrak{q}^{\alpha}, & \text{elliptic}. \end{cases}
\tag{183c}
$$

As in the four-supercharge cases, the $\widetilde{\psi}_-$ currents are expanded around $Z = 0$ in the trigonometric and elliptic cases while the expansions are around $\infty$ for all other cases. In the $\widetilde{\psi}_{\pm}$ currents, we have introduced the shift

$$
\mathfrak{s}^{(a)} := \deg(Q) - \deg(P),
\tag{184}
$$

where $^{\#}\widetilde{\psi}_{\pm}^{(a)}(z) = P/Q$ for polynomials $P, Q$.

## 7.2 Algebras from crystals

We have introduced an extra central element $c$, which is zero in the rational algebra, but can be non-trivial for the trigonometric and elliptic algebras. Only when $c = 0$ do the algebras admit the crystal states as their representations. To write down the actions of the currents on the crystals, it would be convenient to introduce some useful functions as follows.

Suppose that the partition function at level $N$ is given by the poles $\boldsymbol{u} = u_1^*, \ldots, u_N^*$. (Here, for simplicity, we suppress the dependence on the quiver vertex $a \in Q_0$ in the notation $u_i^{(a)*}$.) According to the constructive definition of the JK residues as reviewed above, we have the factorization of the one-loop determinants as

$$
Z_{1\text{-loop}}(u_1, \ldots, u_{N+1}) = Z_{1\text{-loop}}(u_1, \ldots, u_N)\Delta Z(u_1, \ldots, u_{N+1}).
\tag{185}
$$

From [33–35, 38, 92–96], in contrast to the theories with four supercharges, we know that the partition functions now have non-trivial weights. Therefore, the coefficients in the actions of the generators on the crystal states would also carry the necessary information to recover the correct BPS counting.

Given a crystal state $\mathscr{C}$ with $N$ atoms, to implement the raising/lowering to $N \pm 1$, let us introduce the functions

$$
\widetilde{\Psi}_{\mathscr{C}}^{(a)}(z) := \Delta\widetilde{Z}\left(u_1^*, \ldots, u_N^*, u_{N+1} = u_{N_a+1}^{(a)} = z\right),
\tag{186}
$$

where the superscript $(a)$ indicates that the atom to be added/removed is of colour $a$. We need to make two comments regarding the charge functions:

- Here, $\Delta\widetilde{Z}$ means that we do not include the factor $\xi_{\mathcal{N}=2}$. Similarly, we shall introduce the functions for the contributions from the vector multiplets:

$$\widetilde{\Psi}_{V,\mathscr{C}}^{(a)}(z) = \Delta\widetilde{Z}_V = \prod_{\mathfrak{a}\in\mathscr{C}} \zeta(z-\epsilon_\mathfrak{a})\zeta(\epsilon_\mathfrak{a}-z), \tag{187}$$

and the functions for the contributions from the matters:

$$
\begin{aligned}
\widetilde{\Psi}_{\text{matter},\mathscr{C}}^{(a)}(z) &= \Delta Z_{\text{matter}} \\
&= {}^{\#}\widetilde{\psi}^{(a)}(z)\left(\prod_{\mathfrak{a}\in\mathscr{C}} \frac{\prod_{\Lambda\in\{aa\}} \zeta(\epsilon_\Lambda)\zeta(z-\epsilon_\mathfrak{a}-\epsilon_\Lambda)\zeta(z-\epsilon_\mathfrak{a}+\epsilon_\Lambda)}{\prod_{I\in\{a\to a\}} \zeta(\epsilon_I)\zeta(z-\epsilon_\mathfrak{a}-\epsilon_I)\zeta(z-\epsilon_\mathfrak{a}+\epsilon_I)}\right) \\
&\quad \times \left(\prod_{b\neq a}\prod_{\mathfrak{b}\in\mathscr{C}} \frac{\prod_{\Lambda\in\{ab\}} \left(-\zeta\left(\varsigma_\Lambda^{a\Leftarrow b}(z-\epsilon_\mathfrak{b})-\epsilon_\Lambda\right)\right)}{\left(\prod_{I\in\{a\to b\}} (-\zeta(z-\epsilon_\mathfrak{b}+\epsilon_I))\right)\left(\prod_{I\in\{b\to a\}} \zeta(z-\epsilon_\mathfrak{b}-\epsilon_I)\right)}\right).
\end{aligned}
\tag{188}
$$

We have included the factor from the framing explicitly:

$$
{}^{\#}\widetilde{\psi}^{(a)}(z) = \frac{\prod_{\Lambda\in\{a\infty\}} \left(-\zeta\left(\varsigma_\Lambda^{a\Leftarrow\infty}z-\nu_\Lambda\right)\right)}{\left(\prod_{I\in\{a\to\infty\}} (-\zeta(z+\nu_I))\right)\left(\prod_{I\in\{\infty\to a\}} \zeta(z-\nu_I)\right)}.
\tag{189}
$$

As a result

$$\widetilde{\Psi}_{\mathscr{C}}^{(a)}(z) = \widetilde{\Psi}_{V,\mathscr{C}}^{(a)}(z)\widetilde{\Psi}_{\text{matter},\mathscr{C}}^{(a)}(z). \tag{190}$$

- Unlike the double quiver algebras in §5, in the resulting algebras, we always have the $\epsilon = 0$. As discussed in [38], this is necessary to get the right pole structures.[32]

The actions of the generating currents on the crystal states then read

$$
\widetilde{\psi}_{\pm}^{(a)}(z)|\mathscr{C}\rangle = \begin{cases} \widetilde{\Psi}_{\mathscr{C}}^{(a)}(z)|\mathscr{C}\rangle, & \text{rational,} \\ \left[\widetilde{\Psi}_{\mathscr{C}}^{(a)}(Z)\right]_{\pm}|\mathscr{C}\rangle, & \text{trigonometric/elliptic,} \end{cases}
\tag{191a}
$$

$$
\widetilde{\omega}^{(a)}(z)|\mathscr{C}\rangle = \sum_{\text{Inad}\left(\Psi_{\mathscr{C}}^{(a)}(z),\eta\right)} \delta(z-\epsilon_\mathfrak{a})\lim_{x=\epsilon_\mathfrak{a}} \zeta(x-\epsilon_\mathfrak{a})\widetilde{\Psi}_{\mathscr{C}}^{(a)}(x)|\mathscr{C}\rangle,
\tag{191b}
$$

$$
\widetilde{e}^{(a)}(z)|\mathscr{C}\rangle = \sum_{\mathfrak{a}\in\text{Add}(\mathscr{C})} \pm\delta(z-\epsilon_\mathfrak{a})\left(\pm\lim_{x=\epsilon_\mathfrak{a}} \zeta(x-\epsilon_\mathfrak{a})\widetilde{\Psi}_{\mathscr{C}}^{(a)}(x)\right)^{1/2}|\mathscr{C}+\mathfrak{a}\rangle,
\tag{191c}
$$

$$
\widetilde{f}^{(a)}(z)|\mathscr{C}\rangle = \sum_{\mathfrak{a}\in\text{Rem}(\mathscr{C})} \pm\delta(z-\epsilon_\mathfrak{a})\left(\pm\lim_{x=\epsilon_\mathfrak{a}} \zeta(x-\epsilon_\mathfrak{a})\widetilde{\Psi}_{\mathscr{C}-\mathfrak{a}}^{(a)}(x)\right)^{1/2}|\mathscr{C}-\mathfrak{a}\rangle,
\tag{191d}
$$

---

[32]Even at the level of the partition functions, we are not aware of a way to get the refinement to the best of our knowledge.

where $\zeta(z)$ is defined in (11). Let us now verify that the actions satisfy the relations of the currents. The $\widetilde{\psi}\widetilde{\psi}$, $\widetilde{\omega}\widetilde{\omega}$ and $\widetilde{\psi}\widetilde{\omega}$ relations are straightforward since the crystal states are eigenstates of them. There is no need to check the $\widetilde{e}\widetilde{\omega}$ and $\widetilde{f}\widetilde{\omega}$ relations since they are derived from the other relations. Let us now check the $\widetilde{f}\widetilde{f}$ relation.

Consider the operator $\widetilde{f}^{(a)}(z)$ acting on the crystal state by

$$\widetilde{f}^{(a)}(z)|\mathscr{C}\rangle = \sum_{\mathfrak{a}\in\mathrm{Rem}(\mathscr{C})} \delta(z - \epsilon_\mathfrak{a})\widetilde{F}[\mathscr{C} \to \mathscr{C} - \mathfrak{a}]|\mathscr{C} - \mathfrak{a}\rangle, \tag{192}$$

where[33]

$$\widetilde{F}[\mathscr{C} \to \mathscr{C} - \mathfrak{a}] = \pm\left(\pm \lim_{x=\epsilon_\mathfrak{a}} \zeta(x - \epsilon_\mathfrak{a})\widetilde{\Psi}_\mathscr{C}^{(a)}(x)\right)^{1/2}. \tag{193}$$

For $\widetilde{f}^{(a)}(z)\widetilde{f}^{(b)}(w)|\mathscr{C}\rangle$, we have

$$\widetilde{F}[\mathscr{C} - \mathfrak{b} \to \mathscr{C} - \mathfrak{b} - \mathfrak{a}]\widetilde{F}[\mathscr{C} \to \mathscr{C} - \mathfrak{b}] = \pm\left(\pm \lim_{x=\epsilon_\mathfrak{a}} \zeta(x - \epsilon_\mathfrak{a})\widetilde{\Psi}_{\mathscr{C}-\mathfrak{b}-\mathfrak{a}}^{(a)}(x) \lim_{x=\epsilon_\mathfrak{b}} \zeta(x - \epsilon_\mathfrak{b})\widetilde{\Psi}_{\mathscr{C}-\mathfrak{b}}^{(b)}(x)\right)^{1/2}. \tag{194}$$

In particular, $\widetilde{\Psi}_{\mathscr{C}-\mathfrak{b}}(x)$ and $\widetilde{\Psi}_{\mathscr{C}-\mathfrak{b}-\mathfrak{a}}(x)$ follow from the definition in (124). Likewise, for $\widetilde{f}^{(b)}(w)\widetilde{f}^{(a)}(z)|\mathscr{C}\rangle$, we have

$$\widetilde{F}[\mathscr{C} - \mathfrak{a} \to \mathscr{C} - \mathfrak{a} - \mathfrak{b}]\widetilde{F}[\mathscr{C} \to \mathscr{C} - \mathfrak{b}] = \pm\left(\pm \lim_{x=\epsilon_\mathfrak{b}} \zeta(x - \epsilon_\mathfrak{b})\widetilde{\Psi}_{\mathscr{C}-\mathfrak{a}-\mathfrak{b}}^{(b)}(x) \lim_{x=\epsilon_\mathfrak{a}} \zeta(x - \epsilon_\mathfrak{a})\widetilde{\Psi}_{\mathscr{C}-\mathfrak{a}}^{(a)}(x)\right)^{1/2}. \tag{195}$$

Suppose that all the configurations appeared above are valid molecules. In other words, all the $\widetilde{\Psi}$ factors are non-vanishing. Then the two expressions would coincide. If the removed atoms are at non-generic positions, it could be possible that the atom $\mathfrak{a}$ is connected to the atom $\mathfrak{b}$ so that $\mathscr{C} - \mathfrak{b} - \mathfrak{a}$ is a state while $\mathscr{C} - \mathfrak{a} - \mathfrak{b}$ is not a valid configuration. Then the corresponding coefficient in $\widetilde{f}^{(b)}(w)\widetilde{f}^{(a)}(z)|\mathscr{C}\rangle$ would be zero. In this case, we can see that $\widetilde{d}^{(ab)}(z-w)\widetilde{f}^{(a)}(z)\widetilde{f}^{(b)}(w)|\mathscr{C}\rangle$ would also become zero for the configuration $\mathscr{C} - \mathfrak{b} - \mathfrak{a}$ due to the factor $\widetilde{d}^{(ab)}(z-w)$ and the $\delta$-functions in the actions of the currents. As a result,

$$\widetilde{d}^{(ab)}(z-w)\widetilde{f}^{(a)}(z)\widetilde{f}^{(b)}(w) \simeq (-1)^{|a||b|}\widetilde{d}^{(ab)}(z-w)\widetilde{f}^{(b)}(w)\widetilde{f}^{(a)}(z). \tag{196}$$

Likewise, the operator $\widetilde{e}^{(a)}(z)$ acts as

$$\widetilde{e}^{(a)}(z)|\mathscr{C}\rangle = \sum_{\mathfrak{a}\in\mathrm{Add}(\mathscr{C})} \delta(z - \epsilon_\mathfrak{a})\widetilde{E}[\mathscr{C} \to \mathscr{C} - \mathfrak{a}]|\mathscr{C} - \mathfrak{a}\rangle, \tag{197}$$

where

$$\widetilde{E}[\mathscr{C} \to \mathscr{C} + \mathfrak{a}] = \pm\left(\pm \lim_{x=\epsilon_\mathfrak{a}} \zeta(x - \epsilon_\mathfrak{a})\widetilde{\Psi}_\mathscr{C}^{(a)}(x)\right)^{1/2}. \tag{198}$$

Therefore,

$$\widetilde{d}^{(ba)}(z-w)\widetilde{e}^{(a)}(z)\widetilde{e}^{(b)}(w) \simeq (-1)^{|a||b|}\widetilde{d}^{ba}(z-w)\widetilde{e}^{(b)}(w)\widetilde{e}^{(a)}(z). \tag{199}$$

For the $\widetilde{\psi}\widetilde{e}$ relation, consider adding the atom as $|\mathscr{C}\rangle \to |\mathscr{C} + \mathfrak{b}\rangle$. The ratio of the corresponding coefficients in $\widetilde{\psi}_\pm^{(a)}(z)\widetilde{e}^{(b)}(w)|\mathscr{C}\rangle$ and $\widetilde{e}^{(b)}(w)\widetilde{\psi}_\pm^{(a)}(z)|\mathscr{C}\rangle$ is

$$\frac{\widetilde{\Psi}_{\mathscr{C}+\mathfrak{b}}^{(a)}(z)}{\widetilde{\Psi}_\mathscr{C}^{(a)}(z)} = \widetilde{\phi}^{a\Leftarrow b}(z - \epsilon_\mathfrak{b}). \tag{200}$$

As $w \to \epsilon_\mathfrak{b}$, this recovers the $\widetilde{\psi}\widetilde{e}$ relation. The $\widetilde{\psi}\widetilde{f}$ relation can be checked in a similar manner.

---

[33]Henceforth, we shall not repeat the discussions on the signs $\varsigma$, $\varpi$, $\varrho$.

To verify the $\widetilde{e}\widetilde{f}$ relation, consider the terms in $\widetilde{e}^{(a)}(z)\widetilde{f}^{(a)}(w)|\mathscr{C}\rangle$ such that $|\mathscr{C}\rangle \to |\mathscr{C}-\mathfrak{a}\rangle \to |\mathscr{C}\rangle$. The coefficient is

$$\widetilde{E}[\mathscr{C}-\mathfrak{a}\to\mathscr{C}]\widetilde{F}[\mathscr{C}\to\mathscr{C}-\mathfrak{a}] = \pm\left(\pm\lim_{x=\epsilon_\mathfrak{a}}\zeta(x-\epsilon_\mathfrak{a})\widetilde{\Psi}^{(a)}_{\mathscr{C}-\mathfrak{a}}(x)\lim_{x=\epsilon_\mathfrak{a}}\zeta(x-\epsilon_\mathfrak{a})\widetilde{\Psi}^{(a)}_{\mathscr{C}-\mathfrak{a}}(x)\right)^{1/2}$$

$$= \pm\lim_{x=\epsilon_\mathfrak{a}}\zeta(x-\epsilon_\mathfrak{a})\widetilde{\phi}^{a\Leftarrow a}(x-\epsilon_\mathfrak{a})^{-1}\widetilde{\Psi}^{(a)}_{\mathscr{C}}(x)$$

$$= \pm\widetilde{\phi}^{a\Leftarrow a}(0)^{-1}\lim_{x=\epsilon_\mathfrak{a}}\zeta(x-\epsilon_\mathfrak{a})\widetilde{\Psi}^{(a)}_{\mathscr{C}}(x). \qquad (201)$$

Likewise, for $|\mathscr{C}\rangle \to |\mathscr{C}+\mathfrak{a}\rangle \to |\mathscr{C}\rangle$ in $\widetilde{f}^{(a)}(w)\widetilde{e}^{(a)}(z)|\mathscr{C}\rangle$, we have

$$\widetilde{F}[\mathscr{C}+\mathfrak{a}\to\mathscr{C}]\widetilde{E}[\mathscr{C}\to\mathscr{C}+\mathfrak{a}] = \pm\lim_{x=\epsilon_\mathfrak{a}}\zeta(x-\epsilon_\mathfrak{a})\widetilde{\Psi}^{(a)}_{\mathscr{C}}(x)$$

$$= \pm\lim_{x=\epsilon_\mathfrak{a}}\zeta(x-\epsilon_\mathfrak{a})\widetilde{\Psi}^{(a)}_{\mathscr{C}}(x). \qquad (202)$$

On the other hand, when $\mathfrak{a}\neq\mathfrak{b}$ (for either $a=b$ or $a\neq b$), the ratio of the corresponding coefficients in $\widetilde{e}^{(a)}(z)\widetilde{f}^{(b)}(w)|\mathscr{C}\rangle$ and $\widetilde{f}^{(b)}(w)\widetilde{e}^{(a)}(z)|\mathscr{C}\rangle$ is

$$\pm\left(\frac{\lim_{x=\epsilon_\mathfrak{a}}\zeta(x-\epsilon_\mathfrak{a})\widetilde{\Psi}^{(a)}_{\mathscr{C}-\mathfrak{b}}(x)\lim_{x=\epsilon_\mathfrak{b}}\zeta(x-\epsilon_\mathfrak{b})\widetilde{\Psi}^{(b)}_{\mathscr{C}}(x)}{\lim_{x=\epsilon_\mathfrak{b}}\zeta(x-\epsilon_\mathfrak{b})\widetilde{\Psi}^{(b)}_{\mathscr{C}+\mathfrak{a}}(x)\lim_{x=\epsilon_\mathfrak{a}}\zeta(x-\epsilon_\mathfrak{a})\widetilde{\Psi}^{(a)}_{\mathscr{C}}(x)}\right)^{1/2}$$

$$= \begin{cases} \pm\left(\dfrac{\prod\limits_{\Lambda\in\{ab\}}(-\zeta(\epsilon(\mathfrak{t})-\epsilon(\mathfrak{s})-\epsilon_\Lambda))}{\left(\prod\limits_{I\in\{a\to b\}}\zeta(\epsilon_\mathfrak{b}-\epsilon_\mathfrak{a}-\epsilon_I)\right)\left(\prod\limits_{I\in\{b\to a\}}\zeta(\epsilon_\mathfrak{a}-\epsilon_\mathfrak{b}-\epsilon_I)\right)}\right)^{-1}, & a\neq b, \\[3em] \pm\left(\dfrac{\prod\limits_{\Lambda\in\{aa\}}\zeta(-\epsilon_\Lambda)\zeta(\epsilon_\mathfrak{b}-\epsilon_\mathfrak{a}-\epsilon_\Lambda)}{\prod\limits_{I\in\{a\to a\}}\zeta(-\epsilon_I)\zeta(\epsilon_\mathfrak{b}-\epsilon_\mathfrak{a}-\epsilon_I)}\right)^{-1}[\mathfrak{a}\leftrightarrow\mathfrak{b}]', & a=b. \end{cases} \qquad (203)$$

As $z\to\epsilon_\mathfrak{a}$ and $w\to\epsilon_\mathfrak{b}$, we have

$$\widetilde{\phi}^{a\Leftarrow b}(z-w)\widetilde{e}^{(a)}(z)\widetilde{f}^{(b)}(w) - (-1)^{|a||b|}\widetilde{f}^{(b)}(w)\widetilde{e}^{(a)}(z) \simeq \delta_{ab}\delta(z-w)\left(\widetilde{\psi}^{(a)}_+(w)-\widetilde{\psi}^{(a)}_-(z)-\widetilde{\omega}^{(a)}(z)\right). \qquad (204)$$

Let us comment on the definition of $\widetilde{\phi}^{a\Leftarrow b}(z)$. When $b\neq a$, the factors from the Fermi multiplets are

$$-\zeta\left(\varsigma^{a\Leftarrow b}_\Lambda z-\epsilon_\Lambda\right). \qquad (205)$$

If there is an odd number of Fermi multiplets connecting $a$ and $b$, a different convention of the orientations would give either $-\zeta\left(u^{(b)}_j-u^{(a)}_i-\epsilon_{\Lambda_{ab}}\right)$ or $-\zeta\left(u^{(a)}_i-u^{(b)}_j-\epsilon_{\Lambda_{ba}}\right)$, where $\epsilon_{\Lambda_{ab}} = -\epsilon_{\Lambda_{ba}}$, and hence yields an extra minus sign. Since $\widetilde{\phi}^{a\Leftarrow b}(z)$ in different conventions would appear in the relations of the algebra for the same quiver and since different conventions would still give the same partition function, it is natural to conjecture that the algebras would be isomorphic.

Alternatively, we may write such factors as

$$i\zeta(\varsigma^{a\Leftarrow b}_\Lambda z-\epsilon_\Lambda)^{1/2}\zeta(-\varsigma^{a\Leftarrow b}_\Lambda z+\epsilon_\Lambda)^{1/2}. \qquad (206)$$

Now, the problem becomes the choice of the branch cuts of the square roots. In other words, we have either

$$i\zeta(\varsigma^{a\Leftarrow b}_\Lambda z-\epsilon_\Lambda)^{1/2}\zeta(-\varsigma^{a\Leftarrow b}_\Lambda z+\epsilon_\Lambda)^{1/2} = -\zeta(\varsigma^{a\Leftarrow b}_\Lambda z-\epsilon_\Lambda), \qquad (207)$$

or

$$i\zeta(\varsigma_\Lambda^{a\Leftarrow b}z - \epsilon_\Lambda)^{1/2}\zeta(-\varsigma_\Lambda^{a\Leftarrow b}z + \epsilon_\Lambda)^{1/2} = -\zeta(-\varsigma_\Lambda^{a\Leftarrow b}z + \epsilon_\Lambda). \tag{208}$$

Nevertheless, no matter which orientations we choose, we would still get the right information in the partition functions from the crystal representations of the double quiver algebra.

## 7.3 Countings from the double quiver algebras

For theories with two supercharges, the partition functions contain more information than the crystal configurations. Therefore, we would also like to have the non-trivial weights encoded by the double quiver algebra $\widetilde{\mathsf{Y}}$.

Similar to the refined partition functions for theories with four supercharges, the full counting information can be recovered by considering the coefficients in the crystal representations. Again, this can be done by considering the actions of the $\widetilde{e}$ currents. Suppose that we start with the empty state $|\varnothing\rangle$ and reach the state $|\mathscr{C}\rangle$ following the process

$$|\varnothing\rangle \to |\mathfrak{a}_1\rangle \to |\mathfrak{a}_1 + \mathfrak{a}_2\rangle \to \cdots \to |\mathfrak{a}_1 + \mathfrak{a}_2 + \cdots + \mathfrak{a}_N\rangle = |\mathscr{C}\rangle, \tag{209}$$

which is allowed as long as the configuration is allowed at each step (namely, satisfying the melting rule). Then this state is one of the summands in

$$\widetilde{e}^{(a_N)}(z_N)\ldots\widetilde{e}^{(a_2)}(z_2)\widetilde{e}^{(a_1)}(z_1)|\varnothing\rangle. \tag{210}$$

Recall that

$$\widetilde{E}[\mathscr{C} \to \mathscr{C} + \mathfrak{a}] = \pm\left(\pm\lim_{x=\epsilon_\mathfrak{a}}\zeta(x - \epsilon_\mathfrak{a})\widetilde{\Psi}_\mathscr{C}^{(a)}(x)\right)^{1/2}. \tag{211}$$

Denoting the state at the $i^{\text{th}}$ step as $|\mathscr{C}_i\rangle$, the refined coefficient for this state is

$$\widetilde{\xi}_{\mathcal{N}=2}\prod_{i=0}^{N-1}\widetilde{E}[\mathscr{C}_i \to \mathscr{C}_{i+1}]^2, \tag{212}$$

where we have restored the factor

$$\widetilde{\xi}_{\mathcal{N}=2} = \begin{cases} \eta(2\tau)^N, & \text{elliptic}, \\ 1, & \text{rational/trigonometric}. \end{cases} \tag{213}$$

Equivalently, we may consider residues of the overlap the state (210) with the crystal state $|\mathscr{C}\rangle$:

$$\text{Res}_{z_N=\epsilon(\mathfrak{a}_N)}\ldots\text{Res}_{z_1=\epsilon(\mathfrak{a}_1)}\left\langle\mathscr{C}\left|\widetilde{e}^{(a_N)}(z_N)\ldots\widetilde{e}^{(a_2)}(z_2)\widetilde{e}^{(a_1)}(z_1)\right|\varnothing\right\rangle. \tag{214}$$

Then the refined coefficient for this state is obtained by an absolute square of this expression, together with a factor of $\widetilde{\xi}_{\mathcal{N}=2}$.

## 7.4 Miscellaneous comments

From the defining relations of the double quiver algebras $\widetilde{\mathsf{Y}}$ for theories with four supercharges and those for two supercharges, we can see that they are actually of the same form. Their difference is encoded by the bond factors $\widetilde{\phi}^{a\Leftarrow b}(z)$, which depend on the one-loop determinants in the partition functions. Therefore, the bond factors would also satisfy certain properties as those for the one-loop determinants. For instance, the product of the contributions of a chiral with R-charge $R/2$ and of a Fermi with R-charge $R/2 - 1$ in the same representation in an $\mathcal{N}=2$ theory would be equal to the contribution of a chiral of R-charge $R$ in an $\mathcal{N}=4$ theory. Nevertheless, the precise relations between double quivers algebras $\widetilde{\mathsf{Y}}_{\mathcal{N}=4}$ and $\widetilde{\mathsf{Y}}_{\mathcal{N}=2}$ deserve further study.

One might also wonder whether it is possible to construct some single quiver Yangian(-like) algebras for theories with two supercharges. Unfortunately, it seems impossible to extract a state-independent part from the one-loop determinant whose poles are precisely the addable and removable atoms as we did in §6.1. For the case of $\mathbb{C}^4$, the charge function $\Psi(z)$ with only simple poles that have such structure was constructed in [36], which one might hope will be a seed ingredient for the quiver Yangian-like algebra in question. The expression, however, depends on the specific configurations of the crystals, and this makes it difficult to get the state-independent relations of the generators from the crystal representations.

We may also define some enlarged algebras $\widetilde{\mathsf{Y}}^\sharp$ for theories with two supercharges similar to what we did in §6.2. Now we need two sets of currents $\overline{\psi}_\chi^{(a)}(z)$ and $\overline{\psi}_\Lambda^{(a)}(z)$ for contributions from the chirals and the Fermis respectively. This would then introduce two types of bond factors $\overline{\phi}_\chi^{a \Leftarrow b}(z)$ and $\overline{\phi}_\Lambda^{a \Leftarrow b}(z)$ as opposed to (166). Motivated by the parallel with their four-supercharge counterparts, one may expect to extract the single quiver Yangian as a subalgebra of the resulting enhanced double quiver algebra. Nevertheless, it is still not clear how to extract the quiver Yangian(-like) algebras for $\mathcal{N} = 2$.

It could still be possible that the quiver Yangian(-like) algebras might look quite different from those for theories with four supercharges. After all, the crystals themselves are not sufficient for the counting problem. In the double quiver algebras $\widetilde{\mathsf{Y}}$, the non-trivial weights are encoded in the coefficients of the crystal representations. It would be desirable to also treat these non-trivial weights as part of the labels of the states (so that we have $|\mathscr{C}, \dots\rangle$ where the ellipsis denotes the extra labels from the non-trivial weights). As the non-trivial weights are rational functions of the fugacities, we can take the Taylor expansions, and we would like to understand them as enumerating the BPS states transform differently under the U(1) symmetries according to their fugacities. So far, it is not clear how to put these into the states and construct the representations.

# 8 Examples

Let us now consider some illustrative examples. We will first discuss some theories with four supercharges. Then we will see some examples where the double quiver algebras $\widetilde{\mathsf{Y}}$ and the single quiver algebras $\mathsf{Y}$ do not have well-defined representations if we construct the representations in the same way; the discussions there are similar to our discussions for the partition functions from the JK residue formula. For theories with two supercharges, the single quiver BPS algebras are not known, and we will discuss the double quiver algebras for both toric and non-toric quivers.

## 8.1 Theories with four supercharges

We shall start with theories that have four supercharges. The double quiver algebras $\widetilde{\mathsf{Y}}$ were constructed in §5, and we have in addition the single quiver algebras $\mathsf{Y}$ as in §6.

### 8.1.1 No superpotentials

We first take a look at the cases with $W = 0$. As the number of independent equivariant parameters is maximal, this is very similar to the toric $CY_3$ cases.

**Example 1** For the Jordan quiver (54), there is only one gauge node, and hence we shall omit the superscripts $(a)$ and $a \Leftarrow a$. For the double quiver algebra $\widetilde{\mathsf{Y}}$, we have[34]

$$\widetilde{\phi}(z) = \frac{-\zeta(\epsilon - \epsilon_1)}{\zeta(-\epsilon_1)} \frac{\zeta(z)\zeta(-z)\zeta(z + \epsilon - \epsilon_1)\zeta(z - \epsilon + \epsilon_1)}{\zeta(z + \epsilon)(-z + \epsilon)\zeta(z - \epsilon_1)\zeta(z + \epsilon_1)}. \tag{215}$$

The charge function reads

$$\widetilde{\Psi}_{\mathscr{C}}(z) = \frac{-\zeta(z - \epsilon)}{\zeta(z)} \prod_{\mathfrak{a} \in \mathscr{C}} \widetilde{\phi}(z - \epsilon_{\mathfrak{a}}). \tag{216}$$

As an illustration, consider the state with the single initial atom $\mathfrak{o}$. The actions of the currents are

$$\widetilde{\psi}_{\pm}(z)|\mathfrak{o}\rangle = \left[ \frac{-\zeta(z - \epsilon)}{\zeta(z)} \widetilde{\phi}(z) \right]_{\pm} |\mathfrak{o}\rangle, \tag{217a}$$

$$\widetilde{\omega}(z)|\mathfrak{o}\rangle = \delta(z + \epsilon_1) \left( \frac{\zeta(\epsilon - 2\epsilon_1)\zeta(\epsilon)}{\zeta(2\epsilon_1)} \right)^{1/2} |\mathfrak{o}\rangle, \tag{217b}$$

$$\widetilde{e}(z)|\mathfrak{o}\rangle = \delta(z - \epsilon_1) \left( \frac{\zeta(\epsilon_1 - \epsilon)\zeta(2\epsilon_1 - \epsilon)\zeta(\epsilon)\zeta(\epsilon)}{\zeta(\epsilon_1 + \epsilon)\zeta(2\epsilon_1)} \right)^{1/2} |\mathfrak{o} \to \mathfrak{a}\rangle, \tag{217c}$$

$$\widetilde{f}(z)|\mathfrak{o}\rangle = \delta(z)\zeta(\epsilon)^{1/2}|\varnothing\rangle, \tag{217d}$$

where the only state with two atoms is sketched as $|\mathfrak{o} \to \mathfrak{a}\rangle$. For the single quiver Yangian $\mathsf{Y}$, we have

$$\phi(z) = -\frac{\zeta(z + \epsilon_1)}{\zeta(z - \epsilon_1)}. \tag{218}$$

The charge function is

$$\Psi_{\mathscr{C}}(z) = \frac{1}{\zeta(z)} \prod_{\mathfrak{a} \in \mathscr{C}} \phi(z - \epsilon_{\mathfrak{a}}). \tag{219}$$

The actions of the currents can be written in a similar manner following the expressions in Appendix A.

**Example 2** Let us now consider the quiver (66). For the double quiver algebra $\widetilde{\mathsf{Y}}$, we have

$$\widetilde{\phi}^{1 \Leftarrow 2}(z) = \frac{-\zeta(z - \epsilon + \epsilon_1)}{\zeta(z + \epsilon_1)} \frac{-\zeta(z + \epsilon - \epsilon_2)}{\zeta(z - \epsilon_2)}, \tag{220a}$$

$$\widetilde{\phi}^{2 \Leftarrow 1}(z) = \frac{-\zeta(z - \epsilon + \epsilon_2)}{\zeta(z + \epsilon_2)} \frac{-\zeta(z + \epsilon - \epsilon_1)}{\zeta(z - \epsilon_1)}, \tag{220b}$$

with $\widetilde{\phi}^{a \Leftarrow a}(z) = \frac{\zeta(z)\zeta(-z)}{\zeta(z + \epsilon)(-z + \epsilon)}$. The charge functions are

$$\widetilde{\Psi}_{\mathscr{C}}^{(1)}(z) = \frac{-\zeta(z - \epsilon)}{\zeta(z)} \prod_{2 \in \mathscr{C}} \widetilde{\phi}^{1 \Leftarrow 2}(z - \epsilon(2)), \tag{221a}$$

$$\widetilde{\Psi}_{\mathscr{C}}^{(2)}(z) = \prod_{1 \in \mathscr{C}} \widetilde{\phi}^{2 \Leftarrow 1}(z - \epsilon(1)). \tag{221b}$$

For the single quiver algebra $\mathsf{Y}$, we have

$$\phi^{1 \Leftarrow 2}(z) = -\frac{\zeta(z + \epsilon_1)}{\zeta(z - \epsilon_2)}, \tag{222a}$$

$$\phi^{2 \Leftarrow 1}(z) = -\frac{\zeta(z + \epsilon_2)}{\zeta(z - \epsilon_1)}, \tag{222b}$$

---

[34]Notice that here, $\epsilon = \epsilon_1 + \epsilon_2$ with the refinement parameter $\epsilon_2$. Equivalently, $\epsilon$ is treated as an independent parameter.

with $\phi^{a\Leftarrow a}(z) = 1$. The charge functions read

$$\Psi_{\mathscr{C}}^{(1)}(z) = \frac{1}{\zeta(z)} \prod_{2\in\mathscr{C}} \phi^{1\Leftarrow 2}(z - \epsilon(2)), \tag{223a}$$

$$\Psi_{\mathscr{C}}^{(2)}(z) = \prod_{1\in\mathscr{C}} \phi^{2\Leftarrow 1}(z - \epsilon(1)). \tag{223b}$$

The actions of the currents can be written similarly following the expressions in Appendix A.

### 8.1.2 Trivial partition functions

For quivers with no free parameters, we have $Z_{1\text{-loop}} = 1$. Therefore,

$$\widetilde{\phi}^{a\Leftarrow b}(z) = 1, \qquad \widetilde{\Psi}^{(a)}(z) = 1, \tag{224}$$

for the double quiver algebra $\widetilde{Y}$, and

$$\phi^{a\Leftarrow b}(z) = 1, \qquad \Psi^{(a)}(z) = 1, \tag{225}$$

for the single quiver algebra $Y$. In terms of the crystal representation, there is only one state $|\varnothing\rangle$, which is the eigenstate of the $\widetilde{\psi}, \psi$ currents, and is annihilated by the $\widetilde{e}, \widetilde{f}, e, f$ currents. In the remaining part of this subsection, we shall only write the bond factors explicitly and omit the charge functions as well as the actions of the currents.

### 8.1.3 Affine $C_2$ theory

Let us consider the quiver (72). For the double quiver algebra $\widetilde{Y}$, we have

$$\widetilde{\phi}^{1\Leftarrow 1}(z) = \widetilde{\phi}^{3\Leftarrow 3}(z) = \frac{-\zeta(\epsilon - \epsilon_3)}{\zeta(-\epsilon_3)} \frac{\zeta(z)\zeta(-z)}{\zeta(z+\epsilon)\zeta(-z+\epsilon)} \frac{\zeta(z+\epsilon-\epsilon_3)\zeta(z-\epsilon+\epsilon_3)}{\zeta(z-\epsilon_3)\zeta(z+\epsilon_3)}, \tag{226a}$$

$$\widetilde{\phi}^{2\Leftarrow 2}(z) = \frac{-\zeta(\epsilon - \epsilon_3/2)}{\zeta(-\epsilon_3/2)} \frac{\zeta(z)\zeta(-z)}{\zeta(z+\epsilon)\zeta(-z+\epsilon)} \frac{\zeta(z+\epsilon-\epsilon_3/2)\zeta(z-\epsilon+\epsilon_3/2)}{\zeta(z-\epsilon_3/2)\zeta(z+\epsilon_3/2)}, \tag{226b}$$

$$\widetilde{\phi}^{a\Leftarrow a+1}(z) = \frac{-\zeta(z-\epsilon+\epsilon_1)}{\zeta(z+\epsilon_1)} \frac{-\zeta(z+\epsilon-\epsilon_2)}{\zeta(z-\epsilon_2)}, \tag{226c}$$

$$\widetilde{\phi}^{a+1\Leftarrow a}(z) = \frac{-\zeta(z-\epsilon+\epsilon_2)}{\zeta(z+\epsilon_2)} \frac{-\zeta(z+\epsilon-\epsilon_1)}{\zeta(z-\epsilon_1)}, \tag{226d}$$

with other bond factors being 1. For the single quiver algebra $Y$, we have

$$\phi^{1\Leftarrow 1}(z) = \phi^{3\Leftarrow 3}(z) = \frac{-\zeta(z-2h)}{\zeta(z+2h)}, \tag{227a}$$

$$\phi^{2\Leftarrow 2}(z) = \frac{-\zeta(z-h)}{\zeta(z+h)}, \tag{227b}$$

$$\phi^{a\Leftarrow a+1}(z) = \phi^{a+1\Leftarrow a}(z) = \frac{-\zeta(z+h)}{\zeta(z-h)}, \tag{227c}$$

with other bond factors being 1.

### 8.1.4 Affine $G_2$ theory

Let us now consider the quiver (77). For the double quiver algebra $\widetilde{\mathsf{Y}}$, we have

$$\widetilde{\phi}^{1\Leftarrow 1}(z) = \widetilde{\phi}^{2\Leftarrow 2}(z) = \widetilde{\phi}^{3\Leftarrow 3}(z) = \frac{-\zeta(\epsilon - \epsilon_3)}{\zeta(-\epsilon_3)} \frac{\zeta(z)\zeta(-z)}{\zeta(z+\epsilon)\zeta(-z+\epsilon)} \frac{\zeta(z+\epsilon-\epsilon_3)\zeta(z-\epsilon+\epsilon_3)}{\zeta(z-\epsilon_3)\zeta(z+\epsilon_3)} , \tag{228a}$$

$$\widetilde{\phi}^{3\Leftarrow 3}(z) = \frac{-\zeta(\epsilon - \epsilon_3/3)}{\zeta(-\epsilon_3/3)} \frac{\zeta(z+\epsilon-\epsilon_3/3)\zeta(z-\epsilon+\epsilon_3/3)}{\zeta(z-\epsilon_3/3)\zeta(z+\epsilon_3/3)} , \tag{228b}$$

$$\widetilde{\phi}^{a\Leftarrow a+1}(z) = \frac{-\zeta(z-\epsilon+\epsilon_1)}{\zeta(z+\epsilon_1)} \frac{-\zeta(z+\epsilon-\epsilon_2)}{\zeta(z-\epsilon_2)} , \tag{228c}$$

$$\widetilde{\phi}^{a+1\Leftarrow a}(z) = \frac{-\zeta(z-\epsilon+\epsilon_2)}{\zeta(z+\epsilon_2)} \frac{-\zeta(z+\epsilon-\epsilon_1)}{\zeta(z-\epsilon_1)} , \tag{228d}$$

with other bond factors being 1. For the single quiver algebra $\mathsf{Y}$, we have

$$\phi^{1\Leftarrow 1}(z) = \phi^{2\Leftarrow 2}(z) = \frac{-\zeta(z-2h)}{\zeta(z+2h)} , \tag{229a}$$

$$\phi^{3\Leftarrow 3}(z) = \frac{-\zeta(z-2h/3)}{\zeta(z+2h/3)} , \tag{229b}$$

$$\phi^{a\Leftarrow a+1}(z) = \phi^{a+1\Leftarrow a}(z) = \frac{-\zeta(z+h)}{\zeta(z-h)} , \tag{229c}$$

with other bond factors being 1.

### 8.1.5 Affine super-Dynkin theories

Let us briefly discuss the theories associated with the (untwisted) affine super Dynkin diagrams. We shall focus on the cases with non-isotropic odd nodes, namely the $\mathfrak{osp}(2m+1|2n)^{(1)}$ and the $G(3)^{(1)}$ cases (satisfying the no-overlap condition). For the non-isotropic odd node, the corresponding quiver node would be a fermionic node with two adjoints [84].

**Example** As an example, consider the quiver (82). For the double quiver algebra $\widetilde{\mathsf{Y}}$, we have

$$\widetilde{\phi}^{1\Leftarrow 1}(z) = \frac{-\zeta(\epsilon - \epsilon_3)}{\zeta(-\epsilon_3)} \frac{\zeta(z)\zeta(-z)}{\zeta(z+\epsilon)\zeta(-z+\epsilon)} \frac{\zeta(z+\epsilon-\epsilon_3)\zeta(z-\epsilon+\epsilon_3)}{\zeta(z-\epsilon_3)\zeta(z+\epsilon_3)} , \tag{230a}$$

$$\widetilde{\phi}^{1\Leftarrow 2}(z) = \frac{-\zeta(z-\epsilon+\epsilon_1)}{\zeta(z+\epsilon_1)} \frac{-\zeta(z+\epsilon-\epsilon_2)}{\zeta(z-\epsilon_2)} , \tag{230b}$$

$$\widetilde{\phi}^{2\Leftarrow 1}(z) = \frac{-\zeta(z-\epsilon+\epsilon_2)}{\zeta(z+\epsilon_2)} \frac{-\zeta(z+\epsilon-\epsilon_1)}{\zeta(z-\epsilon_1)} , \tag{230c}$$

$$\widetilde{\phi}^{2\Leftarrow 2}(z) = \frac{\zeta(\epsilon-\epsilon_3)\zeta(\epsilon+\epsilon_3/2)}{\zeta(-\epsilon_3)\zeta(\epsilon_3/2)} \frac{\zeta(z)\zeta(-z)}{\zeta(z+\epsilon)\zeta(-z+\epsilon)} \frac{\zeta(z+\epsilon-\epsilon_3)\zeta(z+\epsilon+\epsilon_3/2)}{\zeta(z-\epsilon_3)\zeta(z+\epsilon_3/2)} . \tag{230d}$$

For the single quiver algebra $\mathsf{Y}$, we have

$$\phi^{1\Leftarrow 1}(z) = \frac{-\zeta(z-2h)}{\zeta(z+2h)} , \tag{231a}$$

$$\phi^{2\Leftarrow 2}(z) = \frac{\zeta(z-2h)\zeta(z+h)}{\zeta(z+2h)\zeta(z-h)} , \tag{231b}$$

$$\phi^{1\Leftarrow 2}(z) = \phi^{2\Leftarrow 1}(z) = \frac{-\zeta(z+h)}{\zeta(z-h)} . \tag{231c}$$

As argued in [84], given two quivers related by Seiberg duality, one with a non-isotropic node and one without, the quiver Yangians may not be isomorphic (although the latter should

at least contain the former as a subalgebra). If the no-overlap condition is satisfied, the quiver Yangians should still play the role of the quiver BPS algebras. For the $G(3)^{(1)}$ case, all but one phase satisfy the no-overlap condition (including the one with the non-isotropic node). For $\mathfrak{osp}(2m+1|2n)^{(1)}$ case, there are two such phases (one of which has the non-isotropic node).[35] Henceforth, it is possible that the BPS algebras may not be isomorphic under Seiberg duality.

## 8.2 Overlapping atoms

Recall that the no-overlap condition says that the atoms are not allowed to occupy the same positions in the crystal. We have seen an example in §4.7 where higher order poles appear in the one-loop determinant. Let us now consider this example in terms of the algebras.

For the $D_4^{(1)}$ theory with the quiver given in (92), we would have a double pole as mentioned above. For example, consider the configuration

$$u_1^{(1)}-v_1\,, \qquad u_1^{(3)}-u_1^{(1)}-\epsilon_1\,, \qquad u_1^{(2)}-u_1^{(3)}-\epsilon_2\,, \qquad u_1^{(4)}-u_1^{(3)}-\epsilon_1\,, \qquad u_1^{(5)}-u_1^{(3)}-\epsilon_1\,. \quad (232)$$

In the crystal state $\mathscr{C}$ where we put the initial atom at the origin, one of the charge functions for $\widetilde{\mathsf{Y}}$ would become

$$\widetilde{\Psi}_{\mathscr{C}}^{(3)}(z) = \frac{1}{(z-\epsilon_1-\epsilon_2)^2} \times \dots\,, \quad (233)$$

where the ellipsis does not contain any $(z-\epsilon_1-\epsilon_2)$ factors either in the numerator or in the denominator. This was depicted in (94).

This is expected since the bond factors and the charge functions for $\widetilde{\mathsf{Y}}$ completely follow the expressions of the one-loop determinant in the JK residue formula. Let us also take a look at the single quiver algebra $\mathsf{Y}$. For the same crystal state $\mathscr{C}$, we have

$$\Psi_{\mathscr{C}}^{(3)}(z) = \phi^{3\Leftarrow 1}(z)\phi^{3\Leftarrow 3}(z-\epsilon_1)\phi^{3\Leftarrow 2}(z-2\epsilon_1)\phi^{3\Leftarrow 4}(z-2\epsilon_1)\phi^{3\Leftarrow 5}(z-2\epsilon_1)$$
$$= \frac{(z-\epsilon_1)^2}{(z-3\epsilon_1)^2}\,, \quad (234)$$

with this double pole appearing. Suppose we uplift this with an extra parameter such that

$$\epsilon_{I_{ab}} = \begin{cases} \epsilon_1\,, & a < b\,, \\ -(\epsilon_1+\epsilon_2)\,, & a = b\,, \\ \epsilon_2\,, & a > b\,. \end{cases} \quad (235)$$

The uplift would still not resolve this:

$$\Psi_{\mathscr{C}}^{(3)}(z) = \phi^{3\Leftarrow 1}(z)\phi^{3\Leftarrow 3}(z-\epsilon_1)\phi^{3\Leftarrow 2}(z-\epsilon_1-\epsilon_2)\phi^{3\Leftarrow 4}(z-2\epsilon_1)\phi^{3\Leftarrow 5}(z-2\epsilon_1)$$
$$= \frac{(z-\epsilon_1)^2}{(z-2\epsilon_1-\epsilon_2)^2}\,. \quad (236)$$

Of course, it only shows that the standard representation does not work for BPS counting here, and in fact, this does not even give us a representation. There is still a possibility that the quiver Yangians might still play the role of the BPS algebras, but we need to seek some different representations.

---

[35]For $\mathfrak{osp}(1|2n)^{(1)}$, namely $B(0,n)^{(1)}$, there is only one single phase in all. It has the non-isotropic node and satisfies the no-overlap condition.

## 8.3 Theories with two supercharges

Let us now consider some theories with two supercharges. The double quiver algebras $\widetilde{Y}$ were constructed in §7. Recall that in these cases, we only have the unrefined counting where $\epsilon = \sum_k \epsilon_k$ is always zero.

### 8.3.1 The $\mathbb{C}^4$ theory

The simplest toric $CY_4$ case would be $\mathbb{C}^4$ whose quiver reads

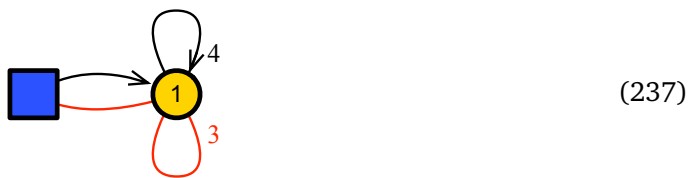

$$\text{(237)}$$

The numbers attached to the edges indicate their multiplicities. The weights of the edges are[36]

| $\chi_1$ | $\chi_2$ | $\chi_3$ | $\chi_4$ | $\Lambda_1$ | $\Lambda_2$ | $\Lambda_3$ | $\chi_{\infty 1}$ | $\Lambda_{1\infty}$ |
|---|---|---|---|---|---|---|---|---|
| $\epsilon_1$ | $\epsilon_2$ | $\epsilon_3$ | $\epsilon_4$ | $-\epsilon_2 - \epsilon_3$ | $-\epsilon_1 - \epsilon_3$ | $-\epsilon_1 - \epsilon_2$ | $v_1$ | $-v_2$ |

$$\text{(238)}$$

Here, the variables satisfy the Calabi-Yau condition $\sum_{i=1}^{4} \epsilon_4 = 0$.

The bond factor is[37]

$$\widetilde{\phi}(z) = -\frac{\zeta(z)\zeta(-z) \displaystyle\prod_{1 \le k < l \le 3} \zeta(\epsilon_k + \epsilon_l)\zeta(z + \epsilon_k + \epsilon_l)\zeta(z - \epsilon_k - \epsilon_l)}{\displaystyle\prod_{k=1}^{4} \zeta(\epsilon_k)\zeta(z + \epsilon_k)\zeta(z - \epsilon_k)} . \tag{239}$$

The charge function is

$$\widetilde{\Psi}_{\mathscr{C}}(z) = \frac{\zeta(z - v_2)}{\zeta(z - v_1)} \prod_{\mathfrak{a} \in \mathscr{C}} \widetilde{\phi}(z - \epsilon_{\mathfrak{a}}), \tag{240}$$

where we have put the initial atom at the position $v_1$.

Let us list the actions of the currents on the first two crystal states as an illustration:

- Level 0:

$$\widetilde{\psi}_{\pm}(z)|\varnothing\rangle = \left[\frac{\zeta(z - v_2)}{\zeta(z - v_1)}\right]_{\pm}|\varnothing\rangle, \tag{241a}$$

$$\widetilde{\omega}(z)|\varnothing\rangle = 0, \tag{241b}$$

$$\widetilde{e}(z)|\varnothing\rangle = \delta(z - v_1)\zeta(v_1 - v_2)^{1/2}|\mathfrak{o}\rangle, \tag{241c}$$

$$\widetilde{f}(z)|\varnothing\rangle = 0. \tag{241d}$$

---

[36]Since there is only one gauge node, we write the edges $\chi_{11,i}$ and $\Lambda_{11,i}$ as $\chi_i$ and $\Lambda_i$ for brevity.
[37]In this subsection, we will not invoke the balancing (181) for the bond factor.

- Level 1:

$$\widetilde{\psi}_\pm(z)|\mathfrak{o}\rangle = \left[ \frac{\zeta(z-v_2)}{\zeta(z-v_1)} \prod_{k=1}^4 \widetilde{\phi}(z-\epsilon_k) \right]_\pm |\mathfrak{o}\rangle, \tag{242a}$$

$$\widetilde{\omega}(z)|\mathfrak{o}\rangle = \sum_{k=1}^4 \delta(z+\epsilon_k) \left( \frac{\zeta(\epsilon_k+v_2)}{\zeta(\epsilon_k+v_1)} \zeta(x+\epsilon_k) \widetilde{\phi}(x+\epsilon_k) \Big|_{x=-\epsilon_k} \right) |\mathfrak{o}\rangle, \tag{242b}$$

$$\widetilde{e}_\pm(z)|\mathfrak{o}\rangle = \sum_{k=1}^4 \delta(z-\epsilon_k) \left( \frac{\zeta(\epsilon_k-v_2)}{\zeta(\epsilon_k-v_1)} \zeta(x-\epsilon_k) \widetilde{\phi}(x-\epsilon_k) \Big|_{x=\epsilon_k} \right)^{1/2} |\mathfrak{o}\to\mathfrak{a}_k\rangle, \tag{242c}$$

$$\widetilde{f}(z)|\mathfrak{o}\rangle = \delta(z) \left( \zeta(v_1-v_2) \widetilde{\phi}(z-v_1) \right)^{1/2} |\varnothing\rangle. \tag{242d}$$

Here, we have denoted the crystal state with two atoms as $|\mathfrak{o}\to\mathfrak{a}_k\rangle$ where the second atom $\mathfrak{a}_k$ is added along the $\epsilon_k$ direction. In the remaining part of this subsection, we shall only write the bond factors explicitly.

### 8.3.2 The conifold$\times\mathbb{C}$ theory

Our next example would be the theory for conifold$\times\mathbb{C}$ geometry, whose quiver reads

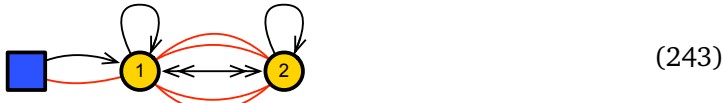

$$\tag{243}$$

The weights of the edges are

| $\chi_{11}$ | $\chi_{12,1}$ | $\chi_{12,2}$ | $\chi_{21,1}$ | $\chi_{21,2}$ | $\chi_{22}$ | $\Lambda_{12,1}$ | $\Lambda_{12,2}$ | $\Lambda_{21,1}$ | $\Lambda_{21,2}$ | $\chi_{\infty 1}$ | $\Lambda_{1\infty}$ |
|---|---|---|---|---|---|---|---|---|---|---|---|
| $\epsilon_4$ | $\epsilon_1$ | $-\epsilon_1$ | $\epsilon_2$ | $-\epsilon_2-\epsilon_4$ | $\epsilon_4$ | $-\epsilon_1+\epsilon_4$ | $\epsilon_1+\epsilon_4$ | $\epsilon_2+\epsilon_4$ | $-\epsilon_2$ | $v_1$ | $-v_2$ |

$$\tag{244}$$

The bond factors are

$$\widetilde{\phi}^{1\Leftarrow 1}(z) = \widetilde{\phi}^{2\Leftarrow 2}(z) = \frac{\zeta(z)\zeta(-z)}{\zeta(\epsilon_4)\zeta(z-\epsilon_4)\zeta(z+\epsilon_4)}, \tag{245a}$$

$$\widetilde{\phi}^{1\Leftarrow 2}(z) = \frac{\zeta(z-\epsilon_1+\epsilon_4)\zeta(z+\epsilon_1+\epsilon_4)\zeta(z-\epsilon_2-\epsilon_4)\zeta(z+\epsilon_2)}{\zeta(z+\epsilon_1)\zeta(z-\epsilon_1)\zeta(z-\epsilon_2)\zeta(z+\epsilon_2+\epsilon_4)}, \tag{245b}$$

$$\widetilde{\phi}^{2\Leftarrow 1}(z) = \frac{\zeta(z+\epsilon_1-\epsilon_4)\zeta(z-\epsilon_1-\epsilon_4)\zeta(z+\epsilon_2+\epsilon_4)\zeta(z-\epsilon_2)}{\zeta(z-\epsilon_1)\zeta(z+\epsilon_1)\zeta(z+\epsilon_2)\zeta(z-\epsilon_2-\epsilon_4)}. \tag{245c}$$

One may also consider the wall-crossing phenomenon. The double quiver algebra $\widetilde{\mathsf{Y}}$ remains the same, but the corresponding representations would change. For instance, there are cyclic chambers connected by changing the framing [38]:

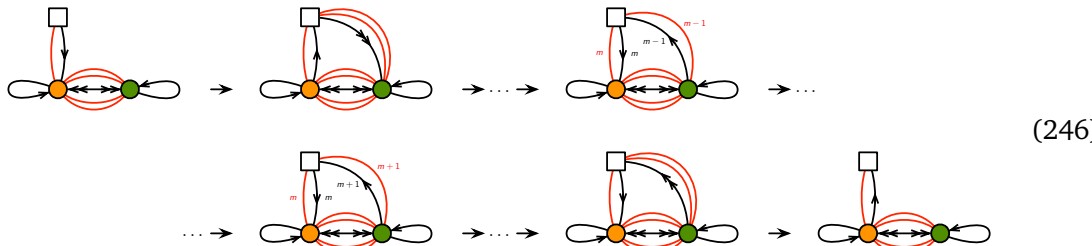

$$\tag{246}$$

In the crystal representations, this is simply a change of the factors from the framing:

$$^\#\widetilde{\psi}^{(1)}(z) = \frac{\zeta(z-v_2-K\epsilon_1-K\epsilon_2)\ldots\zeta(z-v_2+K\epsilon_1-K\epsilon_2)}{\zeta(z-v_1-K\epsilon_1-K\epsilon_2)\ldots\zeta(z-v_1+K\epsilon_1-K\epsilon_2)}, \tag{247a}$$

$$^\#\widetilde{\psi}^{(2)}(z) = \frac{\zeta(v_1+(K-1)\epsilon_1+K\epsilon_2-z)\ldots\zeta(v_1-(K-1)\epsilon_1+K\epsilon_2-z)}{\zeta(v_2+(K-1)\epsilon_1+K\epsilon_2-z)\ldots\zeta(v_2-(K-1)\epsilon_1+K\epsilon_2-z)}, \tag{247b}$$

for the first row and

$$^\#\widetilde{\psi}^{(1)}(z) = \frac{\zeta(z - v_2 - (K-1)\epsilon_1 - (K-1)\epsilon_2)\dots\zeta(z - v_2 + (K-1)\epsilon_1 - (K-1)\epsilon_2)}{\zeta(z - v_1 - (K-1)\epsilon_1 - (K-1)\epsilon_2)\dots\zeta(z - v_1 + (K-1)\epsilon_1 - (K-1)\epsilon_2)}, \quad (248a)$$

$$^\#\widetilde{\psi}^{(2)}(z) = \frac{\zeta(v_1 + K\epsilon_1 + (K-1)\epsilon_2 - z)\dots\zeta(v_1 - K\epsilon_1 + (K-1)\epsilon_2 - z)}{\zeta(v_2 + K\epsilon_1 + (K-1)\epsilon_2 - z)\dots\zeta(v_2 - K\epsilon_1 + (K-1)\epsilon_2 - z)}, \quad (248b)$$

for the second row. Here, we have always labelled the node with incoming chirals from the framing as node 1, and there are $K+1$ (resp. $K$) such arrows for the first (resp. second) row.

### 8.3.3 The $Q^{1,1,1}$ theory

We shall now consider the $Q^{1,1,1}$ theory. It has three phases (two of which are toric) [18]. Here, let us only mention one of them as an illustration:

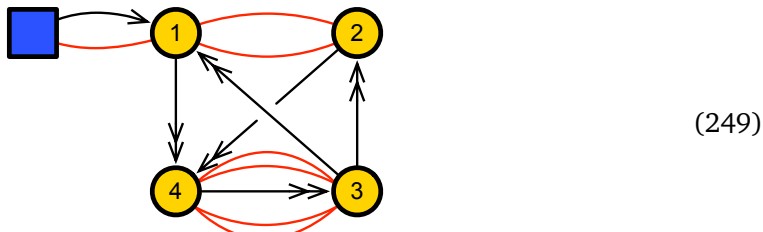

$$(249)$$

The weights of the edges are

| $\chi_{14,1}$ | $\chi_{14,2}$ | $\chi_{24,1}$ | $\chi_{24,2}$ | $\chi_{31,1}$ | $\chi_{31,2}$ | $\chi_{32,1}$ | $\chi_{32,2}$ | $\chi_{43,1}$ | $\chi_{43,2}$ |
|---|---|---|---|---|---|---|---|---|---|
| $\epsilon_1$ | $-\epsilon_1$ | $\epsilon_2$ | $-\epsilon_2$ | $\epsilon_2$ | $-\epsilon_2$ | $\epsilon_1$ | $-\epsilon_1$ | $\epsilon_3$ | $-\epsilon_3$ |
| $\Lambda_{12}$ | $\Lambda_{21}$ | $\Lambda_{34,1}$ | $\Lambda_{34,2}$ | $\Lambda_{34,3}$ | $\Lambda_{34,4}$ | $\chi_{\infty 1}$ | $\Lambda_{1\infty}$ | | |
| $-\epsilon_3$ | $-\epsilon_3$ | $\epsilon_1 + \epsilon_2$ | $\epsilon_1 + \epsilon_2$ | $\epsilon_1 - \epsilon_2$ | $\epsilon_2 - \epsilon_1$ | $v_1$ | $-v_2$ | | |

$$(250)$$

The bond factors are

$$\widetilde{\phi}^{1\Leftarrow 2}(z) = -\zeta(z - \epsilon_3)\zeta(z + \epsilon_3), \quad (251a)$$

$$\widetilde{\phi}^{1\Leftarrow 3}(z) = \frac{1}{\zeta(z - \epsilon_2)\zeta(z + \epsilon_2)}, \quad (251b)$$

$$\phi^{1\Leftarrow 4}(z) = \frac{1}{\zeta(z + \epsilon_1)\zeta(z - \epsilon_1)}, \quad (251c)$$

$$\widetilde{\phi}^{2\Leftarrow 1}(z) = -\zeta(z + \epsilon_3)\zeta(z - \epsilon_3), \quad (251d)$$

$$\widetilde{\phi}^{2\Leftarrow 3}(z) = \frac{1}{\zeta(z - \epsilon_1)\zeta(z + \epsilon_1)}, \quad (251e)$$

$$\widetilde{\phi}^{2\Leftarrow 4}(z) = \frac{1}{\zeta(z + \epsilon_2)\zeta(z - \epsilon_2)}, \quad (251f)$$

$$\widetilde{\phi}^{3\Leftarrow 1}(z) = \frac{1}{\zeta(z + \epsilon_2)\zeta(z - \epsilon_2)}, \quad (251g)$$

$$\widetilde{\phi}^{3\Leftarrow 2}(z) = \frac{1}{\zeta(z + \epsilon_1)\zeta(z - \epsilon_1)}, \quad (251h)$$

$$\widetilde{\phi}^{3\Leftarrow 4}(z) = \frac{\zeta(z + \epsilon_1 + \epsilon_2)^2\zeta(z + \epsilon_1 - \epsilon_2)\zeta(z + \epsilon_2 - \epsilon_1)}{\zeta(z - \epsilon_3)\zeta(z + \epsilon_3)}, \quad (251i)$$

$$\phi^{4\Leftarrow 1}(z) = \frac{1}{\zeta(z - \epsilon_1)\zeta(z + \epsilon_1)}, \quad (251j)$$

$$\widetilde{\phi}^{4\Leftarrow 2}(z) = \frac{1}{\zeta(z - \epsilon_2)\zeta(z + \epsilon_2)}, \quad (251k)$$

$$\widetilde{\phi}^{4\Leftarrow 3}(z) = \frac{\zeta(z-\epsilon_1-\epsilon_2)^2\zeta(z-\epsilon_1+\epsilon_2)\zeta(z-\epsilon_2+\epsilon_1)}{\zeta(z+\epsilon_3)\zeta(z-\epsilon_3)}, \tag{251l}$$

with $\widetilde{\phi}^{a\Leftarrow a}(z) = \zeta(z)\zeta(-z)$.

### 8.3.4 Non-toric examples

Let us also mention some examples that are not from toric $CY_4$. For the quiver (95), the bond factor is

$$\widetilde{\phi}(z) = \frac{-\epsilon(\epsilon_1+\epsilon_2)\zeta(z+\epsilon_1+\epsilon_2)\zeta(z-\epsilon_1-\epsilon_2)}{\zeta(\epsilon_1)\zeta(\epsilon_2)\zeta(z+\epsilon_1)\zeta(z-\epsilon_1)\zeta(z+\epsilon_2)\zeta(z-\epsilon_2)}. \tag{252}$$

For the quiver (100), the bond factors are

$$\widetilde{\phi}^{a\Leftarrow a}(z) = \zeta(z)\zeta(-z), \tag{253a}$$

$$\widetilde{\phi}^{1\Leftarrow 2}(z) = \frac{\zeta(z+\epsilon_1+\epsilon_2+\epsilon_3)}{\zeta(z+\epsilon_1)\zeta(z+\epsilon_2)\zeta(z-\epsilon_3)}, \tag{253b}$$

$$\widetilde{\phi}^{2\Leftarrow 1}(z) = \frac{\zeta(z-\epsilon_1-\epsilon_2-\epsilon_3)}{\zeta(z-\epsilon_1)\zeta(z-\epsilon_2)\zeta(z+\epsilon_3)}. \tag{253c}$$

# 9 Summary and outlook

Given a unitary quiver with polynomial relations, we can use the JK residue formula to obtain the supersymmetric indices as long as it satisfies the no-overlap condition. This extends the previous calculations for the toric CY cases by an uplift with more equivariant parameters. From the JK residue formula, we can also construct the double quiver algebras whose representations encode the BPS state counting.

For theories with four supercharges, we showed that one can actually separate the admissible poles (at least for the cyclic chambers). This then recovers the single quiver BPS algebras in [40–43]. If we keep all the necessary factors in the one-loop determinants of the JK residue formula, then we can obtain the double quiver algebras $\widetilde{Y}$. We have also discussed their relations to the single quiver algebras $Y$.

For theories with two supercharges (including those from toric $CY_4$), there is no similar notion of the single quiver Yangians so far. Nevertheless, since our construction comes from the JK residue formula, we can still construct the double quiver algebras $\widetilde{Y}$ which encode the BPS counting. The states in the representations of $\widetilde{Y}$ are in one-to-one correspondence with the fixed points whose combinatorial structure can be formulated as crystal states. As the crystals themselves are not enough in such situations, there is additional information in the coefficients of the partition functions. This can be recovered from the actions of the generating currents of $\widetilde{Y}$ on the crystal states.

As the discussions here require certain conditions on the quivers, this restricts the validity of the double quiver algebras $\widetilde{Y}$ and the single quiver algebras $Y$. In particular, we have seen that there would be higher-order poles appearing in the charge functions when the no-overlap conditions are not satisfied. One possibility is that the BPS algebras need to be modified when considering more general quivers.

A priori, we need to find a systematic procedure to compute the partition functions for cases that do not satisfy our conditions. This would probably require some additional tricks applied to the JK residue formula. Only with the counting problems in these cases solved can we find the right BPS algebras and the right representations, and understanding how the indices can be computed would be helpful to obtain the BPS algebra structure.

However, this does not rule out the quiver Yangians still being the BPS algebras in these cases. After all, the standard construction which leads to higher order poles does not give us representations of the quiver Yangians. It might be possible that the quiver Yangians are still the desired BPS algebras, but we need to find the right representations. Recently, it was shown in [97] that the Maulik-Okounkov Lie algebra of a quiver is isomorphic to the BPS Lie algebra of its tripled quiver with the canonical superpotential. In particular, we still have the Jacobi algebra as the path algebra quotiented by the $F$-term relations.[38]

Back to the quivers satisfying the conditions in this paper, there are still many investigations for future works. For quiver Yangians, they are not only useful in the context of BPS counting, but also have deep connections to a vast range of areas in physics and mathematics. On the other hand, the properties of the double quiver algebras $\widetilde{Y}$ would still require further studies.

For instance, it could be possible that the quiver Yangians may not always be isomorphic for Seiberg dual quivers, but for the examples we have, one could still get one quiver Yangian at least as a subalgebra of its dual quiver Yangian. Then it is natural to wonder whether the double quiver algebras $\widetilde{Y}$ would be isomorphic under the duality [98] or triality [99] of the quiver gauge theories.

In the case of toric CYs, the quiver theories for $CY_3 \times \mathbb{C}$ can be obtained from those for $CY_3$ under dimensional reduction. It would be interesting to understand the relation between their quiver algebras $\widetilde{Y}_{\mathcal{N}=4}$ and $\widetilde{Y}_{\mathcal{N}=2}$.

For theories with two supercharges, the quiver algebras still take the crystal states as their representations while keeping the extra information in the coefficients of the actions of the generators. It might be possible that the extra information could also be encoded in the crystal states, by suitably enriching the definitions of the crystals. If we know more about the structure of the BPS states, we might be able to construct some new algebras that admit them as representations. Either for the double quiver algebras $\widetilde{Y}$ or for the potential new algebras, we hope that our discussions here shed light on the study of BPS counting, as well as the connections to other topics such as integrability and vertex operator algebras (VOAs).

Finally, it would be a fascinating problem to study the implications of our findings for the spacetime foams [100], where classical smooth geometry emerges in the thermodynamic limit [21, 23, 81]. In this context, the crystal states discussed in this paper represent "atoms" of spacetime, and quiver algebras discussed in this paper represent the creation and annihilation of such "atoms of spacetime." In cyclic chambers, we have seen that all molecules can be constructed from the vacuum by creating atoms one by one, and this bodes well with the philosophy that spacetime can be created out of spacetime foam. However, we have seen that the situation is more complicated in general—in non-cyclic chambers we are sometimes forced to create multiple atoms, as we have seen towards the end of §5.2. This can be interpreted as a "confinement mechanisms for spacetime foams," which may be a general phenomenon in quantum gravity.

# Acknowledgments

The contents of this work were presented by MY at a conference "Quantization in Representation Theory, Derived Algebraic Geometry, and Gauge Theory" (September 2024, SwissMAP Research Station, Les Diablerets), and we thank the organizers for providing a stimulating environment.

---

[38]This includes the DE cases that violate the no-overlap condition. However, it does not contradict our discussions on the other affine (non-ADE) Dynkin cases since the superpotentials are different.

**Funding information** This work is supported in part by a JSPS fellowship [JB], by the JSPS Grant-in-Aid for Scientific Research (Grant No. 20H05860, 23K17689, 23K25865) [MY], and by JST, Japan (PRESTO Grant No. JPMJPR225A, Moonshot R&D Grant No. JPMJMS2061) [MY].

# A Quiver BPS algebras and crystal representations

The relations of the quiver Yangian/BPS algebra are

$$\psi_\pm^{(a)}(z)\psi_\pm^{(b)}(w) \simeq C^{\pm\chi_{ab}}\psi_\pm^{(b)}(w)\psi_\pm^{(a)}(z), \tag{A.1a}$$

$$\psi_+^{(a)}(z)\psi_-^{(b)}(w) \simeq \frac{\varphi^{a\Leftarrow b}(z+c/2, w-c/2)}{\varphi^{a\Leftarrow b}(z-c/2, w+c/2)}\psi_-^{(b)}(w)\psi_+^{(a)}(z), \tag{A.1b}$$

$$\psi_\pm^{(a)}(z)e^{(b)}(w) \simeq \varphi^{a\Leftarrow b}(z\pm c/2, w)e^{(b)}(w)\psi_\pm^{(a)}(z), \tag{A.1c}$$

$$\psi_\pm^{(a)}(z)f^{(b)}(w) \simeq \varphi^{a\Leftarrow b}(z\mp c/2, w)^{-1}f^{(b)}(w)\psi_\pm^{(a)}(z), \tag{A.1d}$$

$$e^{(a)}(z)e^{(b)}(w) \simeq (-1)^{|a||b|}\varphi^{a\Leftarrow b}(z, w)e^{(b)}(w)e^{(a)}(z), \tag{A.1e}$$

$$f^{(a)}(z)f^{(b)}(w) \simeq (-1)^{|a||b|}\varphi^{a\Leftarrow b}(z, w)^{-1}f^{(b)}(w)f^{(a)}(z), \tag{A.1f}$$

$$\left[e^{(a)}(z), f^{(b)}(w)\right\} \simeq \delta_{ab}\left(\delta(z-w-c)\psi_+^{(a)}(z-c/2) - \delta(z-w+c)\psi_-^{(a)}(w-c/2)\right). \tag{A.1g}$$

The bond factor reads

$$\phi^{a\Leftarrow b}(z) = (-1)^{|b\to a|}\frac{\prod\limits_{I\in\{a\to b\}}\zeta(z+\epsilon_I)}{\prod\limits_{I\in\{b\to a\}}\zeta(z-\epsilon_I)}, \tag{A.2}$$

and the balanced bond factor

$$\varphi^{a\Leftarrow b}(z, w) = (ZW)^{\frac{\mathfrak{t}}{2}\chi_{ab}}\phi^{a\Leftarrow b}(z-w), \tag{A.3}$$

with

$$\mathfrak{t} = \begin{cases} 0, & \text{rational}, \\ 1, & \text{trigonometric/elliptic}, \end{cases} \tag{A.4}$$

avoids the fractional powers in the Laurent expansions for chiral quivers. We have $\psi^{(a)}(z) := \psi_+^{(a)}(z) = \psi_-^{(a)}(z)$ and $c$ is trivial for the rational case. The mode expansions of the currents can be found in [43, (2.17)∼(2.19)].

When $c = 0$, we have the crystal representation:

$$\psi_\pm^{(a)}(z)|\mathscr{C}\rangle = \left[\Psi_\mathscr{C}^{(a)}(z)\right]_\pm|\mathscr{C}\rangle, \tag{A.5a}$$

$$e^{(a)}(z)|\mathscr{C}\rangle = \sum_{\mathfrak{a}\in\mathrm{Add}(\mathscr{C})}\left(\pm\delta(z-\epsilon_\mathfrak{a})\left(\pm\lim_{x\to\epsilon_\mathfrak{a}}\Psi_\mathscr{C}^{(a)}(x)\right)^{1/2}\right)|\mathscr{C}+\mathfrak{a}\rangle, \tag{A.5b}$$

$$f^{(a)}(z)|\mathscr{C}\rangle = \sum_{\mathfrak{a}\in\mathrm{Rem}(\mathscr{C})}\left(\pm\delta(z-\epsilon_\mathfrak{a})\left(\pm\lim_{x\to\epsilon_\mathfrak{a}}\Psi_\mathscr{C}^{(a)}(x)\right)^{1/2}\right)|\mathscr{C}-\mathfrak{a}\rangle. \tag{A.5c}$$

The charge function is

$$\Psi_\mathscr{C}^{(a)}(z) = {}^\#\psi^{(a)}(z)\prod_{b\in Q_0}\prod_{\mathfrak{b}\in\mathscr{C}}\varphi^{a\Leftarrow b}(z, \epsilon_\mathfrak{b}). \tag{A.6}$$

In the main text, we have introduced a central element for the double quiver algebra $\widetilde{\mathsf{Y}}$ such that the $\widetilde{\psi}_+$ currents would always commute with themselves. Alternatively, we may consider a central element that looks more symmetric between $\widetilde{\psi}_+$ and $\widetilde{\psi}_-$:

$$\widetilde{\omega}^{(a)}(z)\widetilde{\omega}^{(b)}(w) \simeq \widetilde{\omega}^{(b)}(w)\widetilde{\omega}^{(a)}(z), \tag{A.7a}$$

$$\widetilde{\psi}_\pm^{(a)}(z)\widetilde{\psi}_\pm^{(b)}(w) \simeq \widetilde{\phi}^{a\Leftarrow b}(z-w\pm c)\widetilde{\phi}^{a\Leftarrow b}(z-w\mp c)^{-1}\widetilde{\psi}_\pm^{(b)}(w)\widetilde{\psi}_\pm^{(a)}(z), \tag{A.7b}$$

$$\widetilde{\psi}_+^{(a)}(z)\widetilde{\psi}_-^{(b)}(w) \simeq \widetilde{\psi}_-^{(b)}(w)\widetilde{\psi}_+^{(a)}(z), \tag{A.7c}$$

$$\widetilde{\psi}_\pm^{(a)}(z)\widetilde{\omega}^{(b)}(w) \simeq \widetilde{\phi}^{a\Leftarrow b}(z-w\mp c/2)^{-1}\widetilde{\phi}^{a\Leftarrow b}(z-w\pm c/2)\widetilde{\omega}^{(b)}(w)\widetilde{\psi}_\pm^{(a)}(z), \tag{A.7d}$$

$$\widetilde{\psi}_\pm^{(a)}(z)\widetilde{e}^{(b)}(w) \simeq \widetilde{\phi}^{a\Leftarrow b}(z-w\mp c/2)\widetilde{e}^{(b)}(w)\widetilde{\psi}_\pm^{(a)}(z), \tag{A.7e}$$

$$\widetilde{\psi}_\pm^{(a)}(z)\widetilde{f}^{(b)}(w) \simeq \widetilde{\phi}^{a\Leftarrow b}(z-w\pm c/2)^{-1}\widetilde{f}^{(b)}(w)\widetilde{\psi}_\pm^{(a)}(z), \tag{A.7f}$$

$$\delta(z-w)\widetilde{\phi}^{d\Leftarrow a}(u-z)\widetilde{e}^{(d)}(u)\widetilde{\omega}^{(a)}(z) + \delta(u-w)\widetilde{\phi}^{a\Leftarrow d}(z-w)\widetilde{e}^{(a)}(z)\widetilde{\omega}^{(d)}(w)$$
$$\simeq \delta(z-w)\widetilde{\omega}^{(a)}(z)\widetilde{e}^{(d)}(u) + \delta(u-w)\widetilde{\omega}^{(d)}(u)\widetilde{e}^{(a)}(z), \tag{A.7g}$$

$$\delta(z-w)\widetilde{\phi}^{d\Leftarrow a}(u-z)^{-1}\widetilde{f}^{(d)}(u)\widetilde{\omega}^{(a)}(z) + \delta(u-w)\widetilde{\phi}^{a\Leftarrow d}(z-w)^{-1}\widetilde{f}^{(a)}(z)\widetilde{\omega}^{(d)}(w)$$
$$\simeq \delta(z-w)\widetilde{\omega}^{(a)}(z)\widetilde{f}^{(d)}(u) + \delta(u-w)\widetilde{\omega}^{(d)}(u)\widetilde{f}^{(a)}(z), \tag{A.7h}$$

$$\widetilde{e}^{(a)}(z)\widetilde{e}^{(b)}(w) \simeq (-1)^{|a||b|}\widetilde{e}^{(b)}(w)\widetilde{e}^{(a)}(z), \tag{A.7i}$$

$$\widetilde{f}^{(a)}(z)\widetilde{f}^{(b)}(w) \simeq (-1)^{|a||b|}\widetilde{f}^{(b)}(w)\widetilde{f}^{(a)}(z), \tag{A.7j}$$

$$\widetilde{\phi}^{a\Leftarrow b}(z-w)\widetilde{e}^{(a)}(z)\widetilde{f}^{(b)}(w) - (-1)^{|a||b|}\widetilde{f}^{(b)}(w)\widetilde{e}^{(a)}(z)$$
$$\simeq \delta_{ab}\Big(\delta(z-w-c)\widetilde{\psi}_+^{(a)}(w+c/2) - \delta(z-w+c)\widetilde{\psi}_-^{(a)}(z+c/2) - \delta(z-w)\widetilde{\omega}^{(a)}(z)\Big). \tag{A.7k}$$

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
