# Peer review of "Crystals and Double Quiver Algebras from Jeffrey-Kirwan Residues"

_SciPost Physics, doi:SciPost Phys. 18, 143 (2025)_

## Round 1 · Referee Report · Anonymous (Referee 1) · 2025-2-22

Report

I am familiar with this paper to some extent.

It is an interesting attempt to interpret the emergence of Young diagrams (and their generalizations)
in representation theory of quiver algebras without mentioning them explicitly.
Also it considers extension from CY3 to CY4, which is not quite possible (yet?),
and there is an ongoing dispute of which of the properties to preserve.

In my opinion, this is a nice contribution to the discussion of a hot subject,
and it deserves publication in SciPost .

Recommendation

Publish (meets expectations and criteria for this Journal)

---

## Round 1 · Referee Report · Taro Kimura (Referee 2) · 2025-3-23

Strengths

  1. A systematic review of the Jeffrey-Kirwan residue prescription

  2. A new algebraic formulation of the BPS crystals for both two and four supercharge systems

  3. Detailed study of various examples

Weaknesses

I don't find any specific weak points.

Report

The authors have studied a new algebraic formulation of the BPS crystal through the correspondence to the Jeffrey-Kirwan (JK) residue prescription. In particular, they have examined the role of the inadmissible poles in the JK formulation, and introduced additional "Cartan part" of the quiver algebra, which yields a new algebra that they call the double quiver algebra. From my point of view, the manuscript is clearly written, and it contains useful results
for future reference in the community. I would recommend the current manuscript for publication in SciPost Physics after addressing the points raised in the following.

Requested changes

  1. Page 4: In the discription of $\mathfrak{M}_\text{sing} = \bigcup_i H_i$, what is the range of the index $i$? Is it like $i = 1,...,N$?

  2. Page 5: In eq. (2.8), is the summation of $\mathcal{F}$'s for $\mathcal{FL}^+$?

  3. Page 6: In the 3rd point, concentra $\to$ concentration

  4. Page 6: In "the root system $\Phi$ coincides with the set of vectors given by the adjoint representation," $\to$ "... the set of non-zero weights of the adjoint representation."

  5. Page 6: It'd be instructive to mention that the choice of $\zeta$ in eq. (2.11) corresponds to the elliptic genera, Witten indices, and matrix model partition functions, mentioned in the Introduction.

  6. Page 8: Provide a definition of $\epsilon_\Lambda$ (the charge of the Fermi multiplet).

  7. Page 9: The authors mention the cyclic chamber in the beginning of \S3 and other places. It'd be instructive to give the definition of it somewhere.

  8. Page 11: In eq. (3.8), is it $\subseteq$ rather than $\subset$?

  9. Page 13: In the 1st line, $\sum_{i=1}^F$ $\to$ $\bigoplus_{k=1}^F$

  10. Page 13: In eq. (3.13), should "Suppose that $\mathfrak{a}, \mathfrak{b} \in \mathscr{M}$..." be "... $\mathfrak{a}, \mathfrak{b} \in \mathscr{A}$"?

  11. Page 15: In the last paragraph, for "...no poles originating from the vector multiplets would contribute...," I think it is automatic for two-SUSY cases since there is no pole in the one-loop determinant apriori. How this choice of $\eta$ ensures no contribution of the vector multiplets for four-SUSY cases, in particular for the refined situation and for the unrefined situation?

  12. Page 16: In the last line, Jacobian $\to$ Jacobi (it still makes sense, but it'd be better to unify the notation)

  13. Page 21: Provide a definition of the partition function computed in eq. (3.42), which would be like $\mathcal{Z} = \sum_{{N_a}} (\prod_{a} p_a^{N_a}) \mathcal{Z}_{{N_a}}({\epsilon})$.

  14. Page 21: In eq. (4.2), it seems that the partition function can be written as $\mathcal{Z} = \frac{1-p_1}{1-p_2}$. Does it allow any interpretation compared with eq. (3.42)?

  15. Page 22: It is mentioned that Example 2 gives rise to a four-dimensional crystal. Does it mean that this quiver is related to a CY4 geometry? In that case, is it necessary to impose the CY4 condition, $\sum_{i=1}^4 \epsilon_i = 0$?

  16. Page 23: A related question to above, does the replacement $\epsilon_3 \to - \epsilon_1 - \epsilon_2$ has an interpretation as the CY3 condition?

  17. Page 31: they write $a \Rightarrow b$ in the bond factor, while they also use the notation $a \to b$ to describe the edge. Any distinction?

  18. Page 34: In eq. (5.13), $q$ $\to$ $\mathfrak{q}$ (Eq. (7.8) as well).

  19. Page 34: Below eq. (5.13), $\tilde{\Psi}^{(a)}(z)$ is not (yet) defined.

  20. Page 35: For "The $\tilde{\omega}$ currents collect all the inadmissible poles," does it depend on the choice of $\eta$?

  21. Page 37: In eq. (5.24), any difference between the notations, $\epsilon_{\mathfrak{c}}$ and $\epsilon(\mathfrak{c})$?

  22. Page 44: Below eq. (6.15), it is explained how to take the limit from $\tilde{\mathsf{Y}}$ to $\mathsf{Y}$, which gives rise to some overall factors for the trigonometric and elliptic cases. In the trigonometric case, it would be interpreted as a contribution of the Chern-Simons term. On the other hand, in the elliptic case, is such a factor still consistent with the modular property?

  23. In the double quiver algebra, an interpretation is that $\psi$ and $\omega$ correspond to the BPS particles and anti-particles. Is there any particle-hole symmetry between them? Is there any involution which exchanges $\psi$ and $\omega$ in the double quiver algebra?

  24. Page 48: Does the shift $\mathfrak{s}^{(a)}$ implies the shifted algebra (like shifted Yangian) specific to the two-SUSY case?

  25. Page 60: I suppose the partition function associated with the quiver shown in eq. (8.23) gives rise to that for the Magnificent Four theory of Nekrasov. From this point of view, is it necessary to impose the CY4 condition, $\sum_{i=1}^4 \epsilon_i = 0$?

  26. In the whole manuscript, the authors consider the no-overlap condition, which is equivalent to the condition that no higher poles, as a guiding principle. Meanwhile, in the context of the $qq$-character (see, e.g., https://arxiv.org/abs/1512.05388), one may have such a higher pole, which can be evaluated with the derivatives. In fact, it naturally appears in the $qq$-character of the adjoint representation of the $D_4$ quiver (see Section 7.3 of the aforementioned paper), while the authors discuss the overlapping atoms for the $D_4^{(1)}$ theory in Section 4.7.

Another comment: In the context of the Bethe ansatz, although usually only simple zeros and simple poles appear in the Bethe ansatz equation, one may have higher zeros and poles in a peculiar limit associated with non-simply-laced algebras, https://arxiv.org/abs/1612.00810; https://arxiv.org/abs/1805.01308; https://arxiv.org/abs/2110.14600

Recommendation

Ask for minor revision

---

## Round 1 · Referee Report · Anonymous (Referee 3) · 2025-3-30

Report

This paper investigates crystal melting in quiver gauge theories with two or more supercharges. While this problem has been extensively studied in the context of quivers associated with toric Calabi–Yau 3-folds and 4-folds—where crystal structures are elegantly captured by brane tilings and brane brick models, respectively—this work takes an interesting step forward. It extends the notion of crystals to a broader class of quivers, provided they satisfy a condition termed no-overlap.

Crystals are constructed from Jeffrey–Kirwan (JK) residues in the computation of partition functions. The no-overlap condition ensures that only simple poles appear in the one-loop determinants, guaranteeing the applicability of the JK residue formula.

The authors also introduce a novel class of quiver algebras, which they call double quiver Yangians. Unlike ordinary quiver Yangians, these new algebras include generators for both BPS particles and BPS anti-particles. In theories with two supercharges, partition functions encode more information than the crystal configurations alone, and the double quiver Yangians capture the nontrivial weights associated with these configurations.

The paper illustrates its ideas with several explicit examples of both general crystals and double quiver Yangians. The results are original and interesting, and the paper is well written. I therefore recommend its publication in SciPost.

Recommendation

Publish (easily meets expectations and criteria for this Journal; among top 50%)

---

## Round 2 · Referee Report · Taro Kimura (Referee 2) · 2025-4-11

Report

In the revised version, the authors appropriately addressed all the points that I raised before. I now recommend the current manuscript for publication in SciPost Physics.

Recommendation

Publish (easily meets expectations and criteria for this Journal; among top 50%)

---

## Round 2 · List of Changes

We are grateful to the referees for their careful reading and helpful comments on the paper. We have made the following changes following the questions/suggestions of Report 2. We hope that the changes would implement the referee's comments. We now believe that the paper is ready for publication.

  1. In the relations of the algebras, we noticed that we missed some factors which are called $\widetilde{d}^{(ab)}$ in for example eq.~(5.1j) and (5.1k). The definition of $\widetilde{d}^{(ab)}$ can be found in eq.~(5.4). This factor is necessary when the two atoms to be added/removed are not at generic positions. For instance, $\mathscr{C}-\mathfrak{a}$ and $\mathscr{C}-\mathfrak{a}-\mathfrak{b}$ can be valid configurations while $\mathscr{C}-\mathfrak{b}$ is not if $\mathfrak{b}$ can only be removed after $\mathfrak{a}$ is removed. In this case, $\widetilde{f}^{(a)}(z)\widetilde{f}^{(b)}(w)$ would be zero while $\widetilde{f}^{(b)}(w)\widetilde{f}^{(a)}(z)$ is not. However, $\widetilde{d}^{(ab)}(z-w)\widetilde{f}^{(b)}(w)\widetilde{f}^{(a)}(z)$ would be zero since the relative position of $\mathfrak{a}$ and $\mathfrak{b}$ would make the factor $\widetilde{d}^{(ab)}$ zero. Including this factor on both the left and right hand sides does not mean that we can simply cancel it. This is because the relations of the modes would be different from simply commuting with each other as illustrated in eq.~(5.16). We have made the corresponding changes for the relations for both the $\mathcal{N}=4$ and the $\mathcal{N}=2$ cases.

  2. Page 4: In the description of $\mathfrak{M}_\text{sing}=\bigcup_iH_i$, what is the range of the index $i$? Is it like $i=1,\dots,N$?

The range of $i$ is over all hyperplanes arising from singularities of the one-loop determinant, which could in general be larger than $N$.

  1. Page 5: In eq.~(2.8), is the summation of $\mathcal{F}$'s for $\mathcal{FL}^+$?

Yes. We have added this to the sum.

  1. Page 6: In the 3rd point, concentra $\xrightarrow{}$ concentrate

We have corrected the typo.

  1. Page 6: In the root system $\Phi$ coincides with the set of vectors given by the adjoint representation,'' $\rightarrow$$\dots$ the set of non-zero weights of the adjoint representation.''

We have changed the vectors to non-zero weights.

  1. Page 6: It'd be instructive to mention that the choice of $\zeta$ in eq.~(2.11) corresponds to the elliptic genera, Witten indices, and matrix model partition functions, mentioned in the Introduction.

We have added a sentence to mention this at the end of the page.

  1. Page 8: Provide a definition of $\epsilon_\Lambda$ (the charge of the Fermi multiplet).

We have included the definition below eq.~(2.21): ``$\dots$ whose charges are denoted by $\epsilon_\Lambda$ $\dots$''.

  1. Page 9: The authors mention the cyclic chamber in the beginning of \S3 and other places. It'd be instructive to give the definition of it somewhere.

We have defined it below eq.~(3.16) as a condition following the discussions above.

  1. Page 11: In eq.~(3.8), is it $\subseteq$ rather than $\subset$?

Yes. We have made the correction.

  1. Page 13: In the 1st line, $\bigoplus_{i=1}^F\rightarrow\bigoplus_{k=1}^F$.

We have corrected this.

  1. Page 13: In eq.~(3.13), should Suppose that $\mathfrak{a},\mathfrak{b}\in\mathscr{M}\dots$'' be$\mathfrak{a},\mathfrak{b}\in\mathscr{A}$''.

We have corrected the notation.

  1. Page 15: In the last paragraph, for ``$\dots$ no poles originating from the vector multiplets would contribute $\dots$'', I think it is automatic for two-SUSY cases since there is no pole in the one-loop determinant apriori. How this choice of $\eta$ ensures no contribution of the vector multiplets for four-SUSY cases, in particular for the refined situation and for the unrefined situation?

We gave the reference where this was discussed for the four-SUSY cases, which was in the appendix of the paper.

  1. Page 16: In the last line, Jacobian $\rightarrow$ Jacobi (it still makes sense, but it'd be better to unify the notation)

We have unified the notation.

  1. Page 21: Provide a definition of the partition function computed in eq.~(3.42).

We have added such an expression below eq.~(3.43).

  1. Page 21: In eq.~(4.2), it seems that the partition function can be written as $\mathcal{Z}=\frac{1-p_1}{1-p_2}$. Does it allow any interpretation compared with eq.~(3.42)?

The partition function does not seem to have exactly this closed form, but yes, it does look very similar to the one in eq.~(3.42). If we unrefine it by taking $p_1=p_2=p$, the two partition functions should agree up to the signs of the terms.

  1. Page 22: It is mentioned that Example 2 gives rise to a four-dimensional crystal. Does it mean that this quiver is related to a CY4 geometry? In that case, is it necessary to impose the CY4 condition, $\sum_{i=1}^4\epsilon_i=0$?

In this case, there should be no condition as the Calabi-Yau condition. We happen to have this crystal since we have four independent parameters without any constraints as the superpotential is trivial.

  1. Page 23: A related question to above, does the replacement $\epsilon_3\rightarrow-\epsilon_1-\epsilon_2$ has an interpretation as the CY3 condition?

Yes. Here, we should have the constraint as the Calabi-Yau condition.

  1. Page 31: they write $a\Rightarrow b$ in the bond factor, while they also use the notation $a\rightarrow b$ to describe the edge. Any distinction?

The double arrow is the notation that also appears in the quiver Yangian papers, and here we just use the similar notation. In the expressions with this label, there could be factors from either $a\rightarrow b$ or $b\rightarrow a$ (such as in eq.~(5.2)).

  1. Page 34: In eq.~(5.15), $q\rightarrow\mathfrak{q}$ (Eq.~(7.10) as well).

We have corrected them.

  1. Page 34: Below eq.~(5.13), $\widetilde{\Psi}^{(a)}(z)$ is not (yet) defined.

Indeed, it is defined below. We have now referred to eq.~(5.22) for the definition there.

  1. Page 35: For ``The $\widetilde{\omega}$ currents collect all the inadmissible poles,'' does it depend on the choice of $\eta$?

In general, this could be the case, but the cases with general $\eta$ would be more involved and we only focus on the choice of $\eta=(1,1,\dots,1)$ here. We added a footnote on this on the page.

  1. Page 37: Any difference between the notations, $\epsilon_\mathfrak{c}$ and $\epsilon(\mathfrak{c})$?

There are no differences. We have also slightly rephrased the discussions there to emphasize the case with the non-generic atoms being removed.

  1. Page 44: Below eq.~(6.15), it is explained how to take the limit from $\widetilde{\mathtt{Y}}$ to $\mathtt{Y}$, which gives rise to some overall factors for the trigonometric and elliptic cases. In the trigonometric case, it would be interpreted as a contribution of the Chern-Simons term. On the other hand, in the elliptic case, is such a factor still consistent with the modular property?

This is a very interesting question. In general, each individual factor in the numerator or the denominator does not need to satisfy the property (due to the anomaly cancellation as discussed for example in [Benini-Eager-Hori-Tachikawa '13]). However, when they are multiplied together to recover the full elliptic genus, it would have this modular property.

  1. In the double quiver algebra, an interpretation is that $\psi$ and $\omega$ correspond to the BPS particles and anti-particles. Is there any particle-hole symmetry between them? Is there any involution which exchanges $\psi$ and $\omega$ in the double quiver algebra?

This is a very intriguing point. So far, we haven't found any involution that would exchange the generators/currents. In the actions of $\psi$ and $\omega$, one contains the factors of all poles while the other only involves inadmissible poles. Therefore, if there exists any involution that could be interpreted as BPS particles and anti-particles, we expect that the explicit map would be quite complicated.

  1. Page 48: Does the shift $\mathfrak{s}^{(a)}$ implies the shifted algebra (like shifted Yangian) specific to the two-SUSY case?

Yes. The shift is only for the two-SUSY case. We also have a comment on this when discussing the four-SUSY case below eq.~(5.15).

  1. Page 60: I suppose the partition function associated with the quiver shown in eq.~(8.23) gives rise to that for the Magnificent Four theory of Nekrasov. From this point of view, is it necessary to impose the CY4 condition, $\sum_{i=1}^4\epsilon_i=0$?

Yes. This is the Magnificent Four theory, and we have the Calabi-Yau condition. We have now mentioned this condition below eq.~(8.24).

  1. In the whole manuscript, the authors consider the no-overlap condition, which is equivalent to the condition that no higher poles, as a guiding principle. Meanwhile, in the context of the $qq$-character (see, e.g., https://arxiv.org/abs/1512.05388), one may have such a higher pole, which can be evaluated with the derivatives. In fact, it naturally appears in the $qq$-character of the adjoint representation of the $D_4$ quiver (see Section 7.3 of the aforementioned paper), while the authors discuss the overlapping atoms for the $D_4^{(1)}$ theory in Section 4.7.

Another comment: In the context of the Bethe ansatz, although usually only simple zeros and simple poles appear in the Bethe ansatz equation, one may have higher zeros and poles in a peculiar limit associated with non-simply-laced algebras, https://arxiv.org/abs/1612.00810; https://arxiv.org/abs/1805.01308; https://arxiv.org/abs/2110.14600.

Thank you for pointing this out. Indeed, in that paper, the discussions on the $qq$-character for the $D_4^{(1)}$ case have some extra contribution from the non-isolated fixed point set. Right now, we do not have a concrete solution to remove the no-overlap condition in our discussions. The suggestion on the Bethe ansatz is also a very good point. As we are planning to explore these points in future, let us not further mention these in the paper.

---

## Editorial Decision

published